# A nuclease-mimetic platinum nanozyme induces concurrent DNA platination and oxidative cleavage to overcome cancer drug resistance

Fangyuan Li[1,2,3,6], Heng Sun[1,6], Jiafeng Ren[1,6], Bo Zhang[2,4,6], Xi Hu[1,4,5], Chunyan Fang[4], Jiyoung Lee[1], Hongzhou Gu[4] & Daishun Ling[1,2,4] ✉

Platinum (Pt) resistance in cancer almost inevitably occurs during clinical Pt-based chemotherapy. The spontaneous nucleotide-excision repair of cancer cells is a representative process that leads to Pt resistance, which involves the local DNA bending to facilitate the recruitment of nucleotide-excision repair proteins and subsequent elimination of Pt-DNA adducts. By exploiting the structural vulnerability of this process, we herein report a nuclease-mimetic Pt nanozyme that can target cancer cell nuclei and induce concurrent DNA platination and oxidative cleavage to overcome Pt drug resistance. We show that the Pt nanozyme, unlike cisplatin and conventional Pt nanoparticles, specifically induces the nanozyme-catalyzed cleavage of the formed Pt-DNA adducts by generating in situ reactive oxygen species, which impairs the damage recognition factors-induced DNA bending prerequisite for nucleotide-excision repair. The recruitment of downstream effectors of nucleotide-excision repair to DNA lesion sites, including xeroderma pigmentosum groups A and F, is disrupted by the Pt nanozyme in cisplatin-resistant cancer cells, allowing excessive accumulation of the Pt-DNA adducts for highly efficient cancer therapy. Our study highlights the potential benefits of applying enzymatic activities to the use of the Pt nanomedicines, providing a paradigm shift in DNA damaging chemotherapy.

Cancer remains a colossal threat to public health, with ever-increasing global incidence and mortality[1,2]. Over the past decades, chemotherapy has been a mainstay of the frontline treatment for various malignancies. Particularly, platinum (Pt) compounds have been broadly used as the crucial component of the standard of care for cancers due to their capability to induce cytotoxic Pt-DNA adducts[3,4]. However, intrinsic and/or acquired Pt drug resistance frequently occurs in cancer patients, compromising the therapeutic effectiveness of Pt compounds[5–7]. Recently, numerous Pt-based nanomedicines, including Pt compound-loaded nanocarrier and metallic Pt nanoparticles,

[1]Institute of Pharmaceutics, Hangzhou Institute of Innovative Medicine, College of Pharmaceutical Sciences, Zhejiang University, Hangzhou 310058, China. [2]Frontiers Science Center for Transformative Molecules, School of Chemistry and Chemical Engineering, National Center for Translational Medicine, Shanghai Jiao Tong University, Shanghai 200240, China. [3]WLA Laboratories, Shanghai 201203, China. [4]Key Laboratory of Precision Diagnosis and Treatment for Hepatobiliary and Pancreatic Tumor of Zhejiang Province, Hangzhou 310003, China. [5]Department of Clinical Pharmacy, the First Affiliated Hospital, Zhejiang University School of Medicine, Hangzhou 310003, China. [6]These authors contributed equally: Fangyuan Li, Heng Sun, Jiafeng Ren, Bo Zhang. ✉e-mail: dsling@sjtu.edu.cn

have been developed for targeted drug delivery and/or combinational therapy to overcome Pt resistance of tumours[8–13]. For instance, a cisplatin and sulforaphane co-loaded polymeric nanoparticle was reported to inhibit the deactivation of cisplatin by glutathione (GSH) depletion, thereby facilitating the accumulation of cisplatin in cancer cells and the production of Pt-DNA adducts[14]. Our group reported a pH-sensitive Pt nanoassembly that can disassemble into small-sized Pt nanoparticles in the tumour intracellular acidic endo/lysosomal microenvironment for burst leaching of Pt ions, thus prompting Pt-DNA adduct formation to combat Pt-resistant tumour cells[15]. Nevertheless, the adaptive drug resistance is still inevitable, as the majority of Pt-based anticancer compounds and nanomedicines rely on inducing Pt-DNA adducts to exert anticancer effects, while the long-term treatment with these DNA-damaging agents eventually activates the DNA damage response (DDR) system of cancer cells to promote the removal of Pt-DNA adducts[16,17]. The DDR system encompasses several distinctive DNA repair pathways, such as nucleotide excision repair (NER), base excision repair (BER) and so on, to address different types of DNA damage[18,19]. Notably, the NER pathway plays a predominant role in the elimination of the Pt-DNA adducts, significantly impairing the therapeutic efficacy of Pt-based agents[20,21].

The NER process is commonly known to entail the engagement of a sequence of proteins involved in damage recognition, pre-excision complex formation, excision of DNA lesion, repair synthesis and ligation[22]. Accordingly, small molecular inhibitors that can occupy the active sites of NER proteins have been developed to suppress the DNA repair process. These agents, however, suffer from poor inhibitory efficacy because the active sites of NER proteins are generally small and deeply embedded in tertiary structure of the proteins, conferring minimal benefits to overall survival[23,24]. During the NER process, evidence indicates that the binding between damage recognition proteins and DNA lesion sites can result in local DNA bending[25,26]. Importantly, the bending structure is indispensable for the recruitment of downstream effectors to complete DNA repair[27,28]. Consequently, impairing the conformation of Pt-DNA adducts to prevent the recruitment of the downstream proteins represents a practical approach to suppress NER. Recent advances in the development of artificial nuclease mimics, including metal complexes and nanomaterials, have enabled the induction of hydrolytic or oxidative DNA cleavage and the conformational changes of DNA[29–31]. In particular, nanomaterials with enzyme-like activities and relatively low cost, such as quantum dots[32], carbon-based nanoparticles[33], and metal nanoclusters[34], have been exploited as robust biomimetic nucleases to induce oxidative DNA single- or double-stranded breaks (SSBs/DSBs) by generating reactive oxygen species (ROS). Notably, Pt-DNA lesion near a DSB would impair the non-homologous end-joining (NHEJ) that is responsible for the DSB repairing, permanently altering the DNA conformation essential for NER[35]. In light of these findings, we propose that concurrent DNA platination and oxidative cleavage mediated by enzymatic Pt-based nanomedicine may irreversibly disrupt the DNA conformation prerequisite for NER and overcome the Pt resistance of tumours.

In this work, we report the designed synthesis of nuclease-mimetic Pt nanozymes (NMPNs) that can readily target cancer cell nuclei to efficiently induce Pt-DNA adducts, as well as change the NER-required DNA conformation for overcoming Pt-resistant tumours (Fig. 1). Specifically, NMPNs can target cancer cell nuclei and efficiently release Pt ions to induce the formation of Pt-DNA adducts. Moreover, NMPNs exert highly potent oxidase (OXD)- and peroxidase (POD)-like activities to generate ROS (•OH and •O$_2^-$) that can coordinate with Pt ions of the formed Pt-DNA adducts, to initiate the oxidative cleavage of double-stranded DNA around the Pt-DNA binding sites, thus hindering the DNA bending required for NER. These lead to the suppression of the recruitment of NER-associated factors (e.g., xeroderma pigmentosum groups A (XPA) and xeroderma pigmentosum groups F (XPF)), dramatically inducing excessive accumulation of Pt-DNA adducts and

the apoptosis of Pt-resistant tumour cells. Our results suggest that the nuclease-mimetic NMPNs represent next generation Pt-based anticancer agents, which not only induce DNA damage but also change the DNA conformation required for NER by priming oxidative DNA cleavage, providing a promising strategy for the clinical DNA-damaging chemotherapy.

## Results

### Synthesis and characterization of NMPNs

The NMPNs were prepared as illustrated in Fig. 2a. First, ultrafine Pt nanoclusters (PtNCs) with a diameter of ~2.5 nm were synthesized via a modified heat-up method (Fig. 2b)[15]. High-resolution transmission electron microscope (HRTEM) and X-ray diffraction (XRD) analysis showed the face-centred cubic crystal lattice structure of PtNCs (JCPDS No. 04-0802; Fig. 2b and Supplementary Fig. 1). To afford intracellular nucleus-targeting capacity[36] and tumour pH sensitivity[37], PtNCs were further grafted with the heterobifunctional SH-PEG$_{2K}$-COOH to provide the carboxyl groups for 1-ethyl-3-(3-dimethylaminopropyl)-carbodiimide (EDC)- mediated coupling with the amino groups of the trans-acting activator of transcription (TAT) peptides (GRKKRRQRRR) and then shielded with pH-responsible poly(ethylene glycol) methyl ether-acryloyl-cysteamine (mPEG$_{5K}$-AC-CA) (Supplementary Figs. 2, 3)[38,39], imparting the tumour pH-dependent exposure of TAT peptides of NMPNs. The successful modification of TAT peptide is verified by Fourier transform infra-red (FT-IR) spectra (Supplementary Fig. 4), corresponding with the increased zeta potential and size relative to PtNC@PEG (Supplementary Figs. 5, 6)[40]. NMPNs are well-dispersed in water (Fig. 2c) with the high colloidal stability (Supplementary Fig. 7), and exhibit a more negative charge and a larger hydrodynamic diameter than these of TAT peptides modified PtNCs (PtNC@TAT) (Supplementary Figs. 5, 6), indicating the successful coating of pH-responsive mPEG$_{5K}$-AC-CA. NMPNs can discharge the mPEG$_{5K}$-AC-CA under acidic conditions (pH 6.5) as demonstrated by the shedding behaviour of surface modified mPEG$_{5K}$-AC-CA (Fig. 2d), which is also confirmed by the pH-dependent hydrodynamic sizes and zeta potential changes of NMPNs (Supplementary Figs. 5, 6).

The catalytic activities of NMPNs were further studied. X-ray photoelectron spectroscopy (XPS) analysis shows the co-existence of Pt$^0$, Pt$^{2+}$ and Pt$^{4+}$ on the surface of NMPNs (Fig. 2e)[41], laying the chemical basis for catalytic activities. Indeed, following Michaelis-Menten kinetics (Supplementary Table 1), NMPNs can effectively catalyze tetramethylbenzidine (TMB) oxidation to produce a blue product with maximum absorbance at 652 nm (Supplementary Fig. 8a)[42], which can be inhibited by superoxide dismutase (SOD, a •O$_2^-$ quencher) (Supplementary Fig. 9)[43]. Consistently, the characteristic electron spin resonance (ESR) signals of BMPO-OOH appear upon the addition of NMPNs, indicating the generation of •O$_2^-$ (Fig. 2f)[44]. In the presence of H$_2$O$_2$, NMPNs also exhibit Michaelis-Menten kinetics in the TMB colorimetric reaction (Supplementary Fig. 8b and Supplementary Table 2), and generate the free radical species •OH as indicated by a remarkable DMPO-OH signal in the ESR spectra (Fig. 2g)[44]. Notably, the TMB oxidation can be inhibited by mannite (a •OH quencher) (Supplementary Fig. 10)[43]. These results demonstrate the oxidase (OXD)- and peroxidase (POD)-like activities of NMPNs.

Then, density-functional theory (DFT) calculations were performed to elucidate the catalytic reaction mechanism. As shown in Fig. 2h,i, oxidized Pt surface occupies a high catalytic capacity to decompose H$_2$O$_2$ into O$_2$ and capture H atoms, as indicated by the lower free energy of oxidized Pt than that of metallic Pt (−6.70 eV vs. −0.69 eV). Specifically, oxidized Pt surface easily binds and splits H$_2$O$_2$ into OH groups, and the free energy of H$_2$O$_2$ molecules decomposed to •OH when binding to oxidized Pt is lower than that of metallic Pt (−6.17 eV vs. −2.66 eV). To illustrate the difference between the catalytic capacity of metallic Pt and oxidized Pt, the density-of-state (DOS) and the energy gap between the highest occupied molecular orbital

(HOMO) and the lowest unoccupied molecular orbital (LUMO) of the two surfaces were calculated. The HOMO and LUMO of oxidized Pt are closer to the Fermi level than metallic Pt (HOMO: −0.014 eV vs. −0.050 eV; LUMO: 0.005 eV vs. 0.116 eV, Supplementary Fig. 11), thereby oxidized Pt surface shows high oxidation and reduction reactivity for ROS generation. Metallic Pt as one of the reactants can assist in the formation of reactive radicals and release $Pt^{2+}$. Meanwhile, the DOS of metallic Pt and oxidized Pt are close to each other, showing the strong interaction between metallic and oxidized Pt. Moreover, the electrons near the Fermi level are non-local as the DOS of metallic Pt and oxidized Pt is wide and across the Fermi level ($E_{Fermi}$) (Supplementary Fig. 12), which is conducive to the electron conduction at the interfaces and initiates the oxidation-reduction reaction cycle. These results indicate that the co-existence of oxidized Pt and metallic Pt contribute to the highly OXD and POD activities of PtNCs (Fig. 2j, k).

### pH-dependent cell nucleus-targeting of NMPNs

The nucleus-targeting capability of NMPNs was then evaluated in vitro. Fluorescein isothiocyanate (FITC) labelled NMPNs or Pt NC@mPEG_{5K}-AC-CA (PNPs, without TAT peptide modification) were incubated with human hepatoma (Huh7) cells under different pH conditions (pH 6.5 or 7.4) to examine their cellular uptake and subcellular localization. Under acidic conditions, the FITC signals diffuse across the cytoplasm of the cells, indicating the efficient cellular uptake and endosomal escape of NMPNs. In stark contrast, FITC signals of NMPNs are mostly confined in endosomes in a neutral environment. Moreover, FITC-labelled PNPs are trapped in endosomes both in acidic and neutral environments (Fig. 3a, c). These results indicate that the acid-induced exposure of TAT peptides can facilitate the endosomal escape of NMPNs[45], contributing to the efficient nucleus targeting. Indeed, in an acidic microenvironment, NMPNs can efficiently accumulate in the cell nucleus (Fig. 3b, d), which is beneficial from the surface TAT peptides that can promote the entry into the nucleus via the importin α/β pathway[46]. In contrast, PNPs can barely reach the nucleus, regardless of the pH conditions (Fig. 3b, d). The Bio-TEM images as well as inductively coupled plasma mass spectrometry (ICP-MS) analysis of Pt ions in the nucleus also show the accumulation of NMPNs in the nucleus of Huh7 cells under acidic conditions (Fig. 3e and Supplementary Figs. 13, 14). Moreover, we further studied the cellular endocytosis mechanism of NMPNs in Huh7 cells. In accordance with previously reported TAT

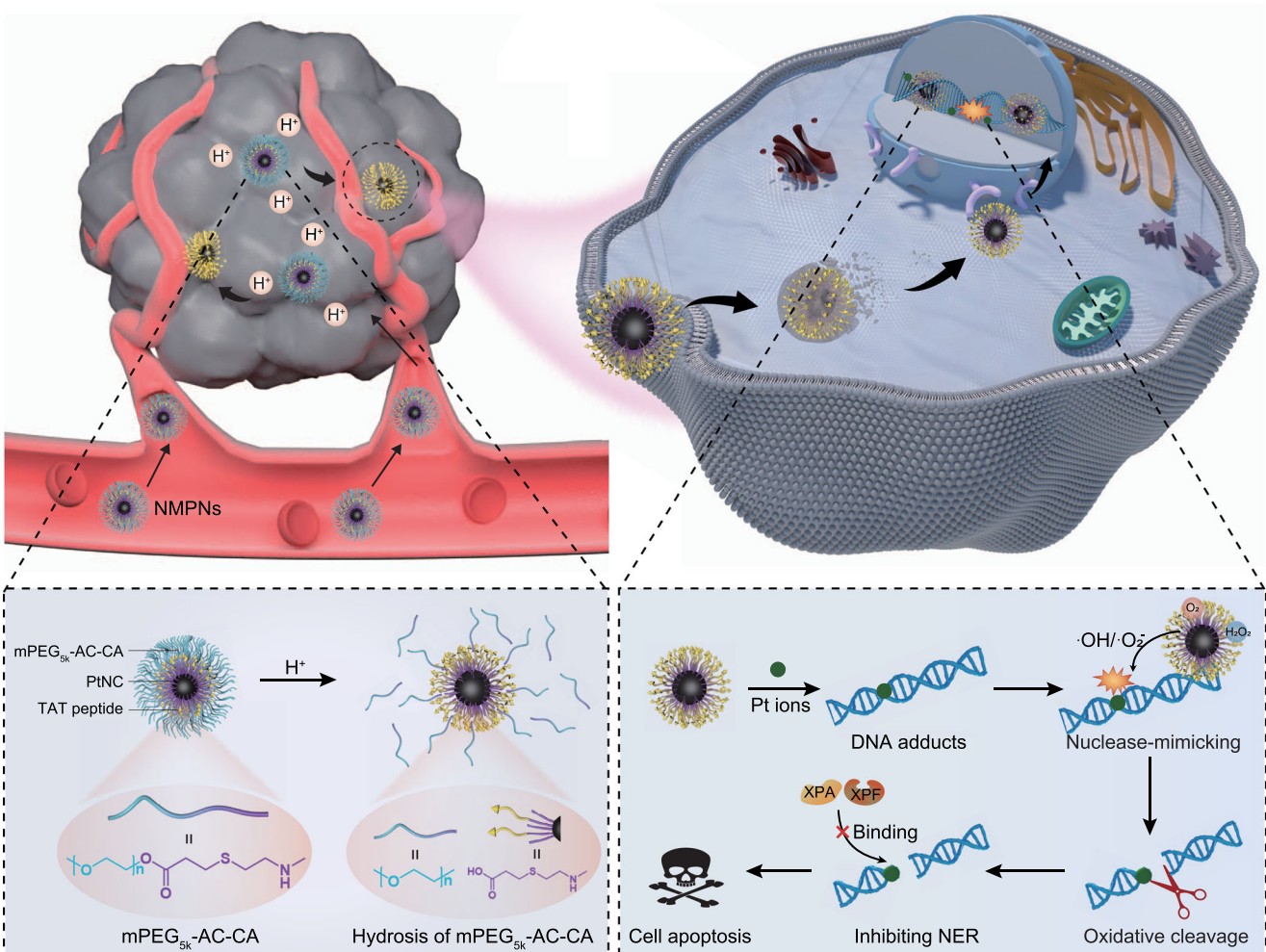

**Fig. 1 | Schematic illustration of NMPNs-mediated concurrent DNA platination and oxidative cleavage to overcome Pt resistance of cancer.** The outermost mPEG_{5K}-AC-CA of NMPNs can be selectively dissociated in tumour acidic microenvironment due to the hydrolysis of the β-thiopropionate group, thus exposing TAT peptides to promote the endocytosis, endosomal escape, and nuclear transportation. NMPNs can efficiently release Pt ions to induce Pt-DNA adducts in the nucleus of cisplatin-resistant tumour cells. Meanwhile, due to the highly active OXD and POD-like activities, NMPNs generate in situ reactive oxygen species (ROS) to induce the oxidative DNA cleavage near the Pt-DNA binding sites, which prohibits further DNA bending and thus destroying the DNA conformation required for NER. Intriguingly, the recruitment of NER effectors (XPA and XPF) to the DNA lesion sites has been disrupted. Consequently, the NMPNs induced and tailored Pt-DNA adducts can efficiently accumulate in the tumour cells without NER-mediated repairing, thus inducing the apoptosis of cisplatin-resistant tumour cells, as well as exerting a potent anti-tumour effect in vivo.

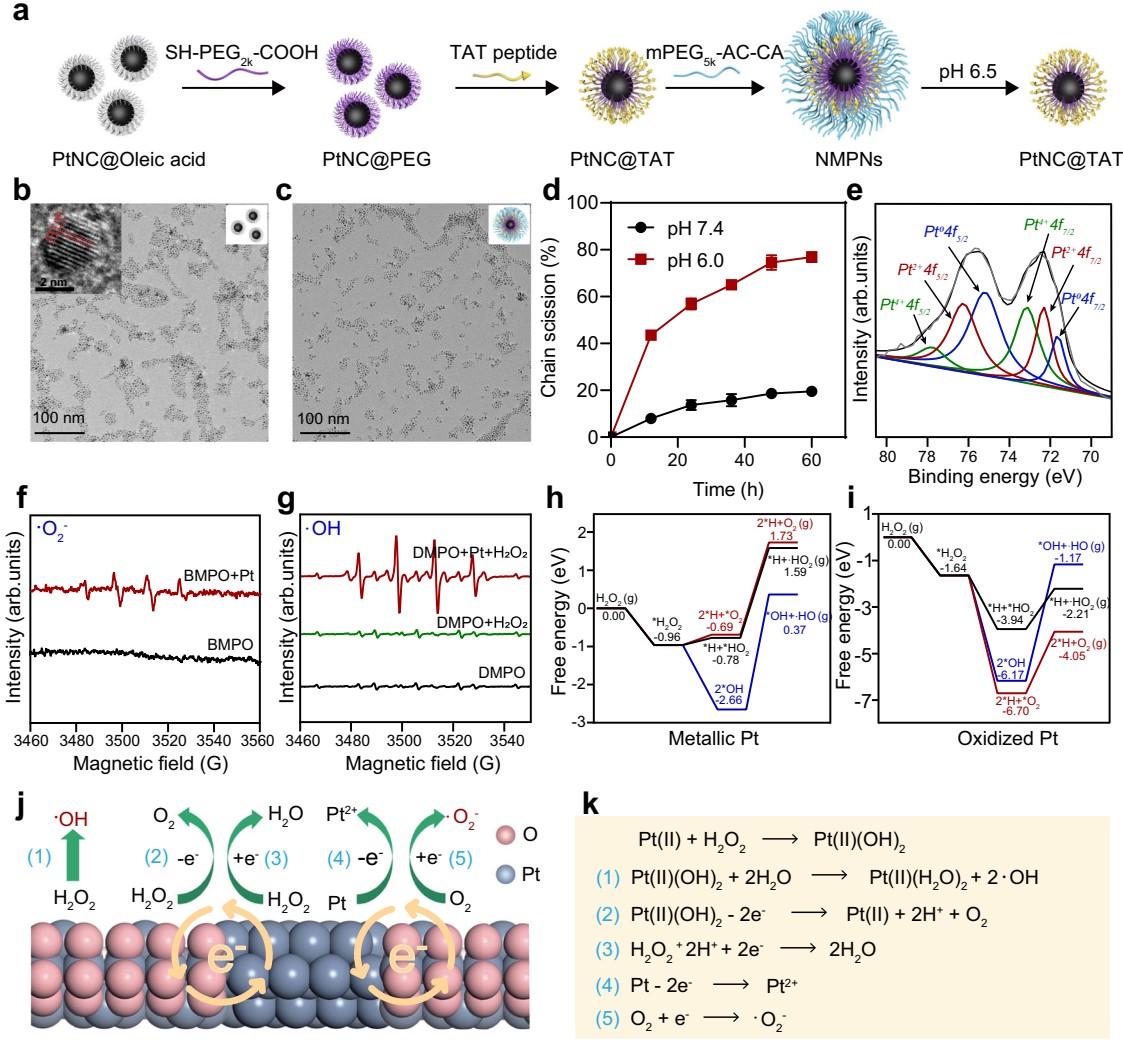

**Fig. 2 | Designed synthesis and characterization of NMPNs with high performance catalytic activity. a** Schematic diagram of the synthesis of NMPNs. **b** Representative TEM image of PtNCs in chloroform. Scale bar: 100 nm; Insert: high-resolution TEM image of PtNCs. Scale bar: 2 nm. $n = 3$ independent experiments. **c** TEM image of NMPNs in water. Scale bar: 100 nm. $n = 3$ independent experiments. **d** The analysis of shedding behaviour of mPEG$_{5K}$-AC-CA from the surface of NMPNs at pH 6.5 and 7.4. Rhodamine B isothiocyanate (RITC) was conjugated to the terminal of mPEG$_{5K}$-AC-CA to obtain RITC-mPEG$_{5K}$-AC-CA, and the fluorescence intensity of RITC-mPEG$_{5K}$-AC-CA fragments released from NTNPs

surface was analyzed. $n = 3$ independent experiments. Statistical significance was analyzed by two-tailed Student's $t$-test. **e** XPS spectra of NMPNs, confirming the co-existence of Pt$^0$ (B.E. at 71.7 and 75.3 eV), Pt$^{2+}$ (B.E. at 72.3 and 76.3 eV), and Pt$^{4+}$ (B.E. at 73.1 and 77.9 eV) on the surface of NMPNs. **f** ESR spectra of NMPNs by using BMPO as spin-trapping agents for detecting ·O$_2^-$. **g** ESR spectra of NMPNs by using DMPO as spin-trapping agents for detecting ·OH. **h** DFT calculation of catalytic reaction on metallic Pt of PtNCs. **i** DFT calculation of catalytic reaction on oxidized Pt of PtNCs. **j, k** The proposed mechanism of NMPNs to catalyze H$_2$O$_2$ and O$_2$ into ·OH and ·O$_2^-$. Source data are provided as a Source Data file.

peptide-modified nanoparticles[47], the cellular uptake of NMPNs can be inhibited when Huh7 cells were pre-incubated with amiloride, chlorpromazine, and MβCD, respectively, indicating that NMPNs can be internalized into Huh7 cells through multiple pathways, among which the MβCD-mediated endocytosis plays a leading role in the cellular endocytosis of NMPNs (Supplementary Fig. 15). These results demonstrate that, different from PNPs, NMPNs implement the pH-responsive discharge of the "protective shield" under acidic conditions to exposure TAT peptides for facilitating the endosomal escape and subsequent nucleus targeting (Fig. 3f and Supplementary Fig. 16), which can readily initiate the formation of Pt-DNA adducts (Supplementary Fig. 17a, b).

### NMPNs induce the formation of Pt-DNA adducts irreparable by NER

Encouraged by the nucleus-targeting capability of NMPNs, we next investigated the detailed effect of NMPNs on DNA strands. NMPNs can

quickly and persistently release Pt ions (Fig. 4a), which would enable NMPNs to induce the formation of Pt-DNA adducts[48]. Additionally, NMPNs can cause DNA fragmentation (Fig. 4b), probably resulting from the excessive NMPNs-catalyzed ROS generation[34]. Moreover, the NMPNs-induced DNA cleavage follows pseudo Michaelis-Menten kinetics (Supplementary Fig. 18). To illustrate the role of NMPNs in DNA damage induction, the configuration of Pt ions binding with DNA as well as the mechanism of DNA cleavage by NMPNs-generated ROS were calculated using the DFT method. Previous studies have indicated that Pt ions can bind with DNA in a similar way with cisplatin[49,50]. Consistently, the results show that the released Pt ions can readily bind to the single-stranded DNA by coordinating with the O atoms of the DNA phosphate and deoxyribose groups. Then, Pt ions can capture H4' hydrogen atoms from adjacent deoxyribose to form a rigid planar tetra-coordination structure with a C4'-sugar-radical (Fig. 4c). The formed rigid structure can keep C4'-sugar-radical stable through stereo hindrance effect. Intriguingly, ROS generated by NMPNs can

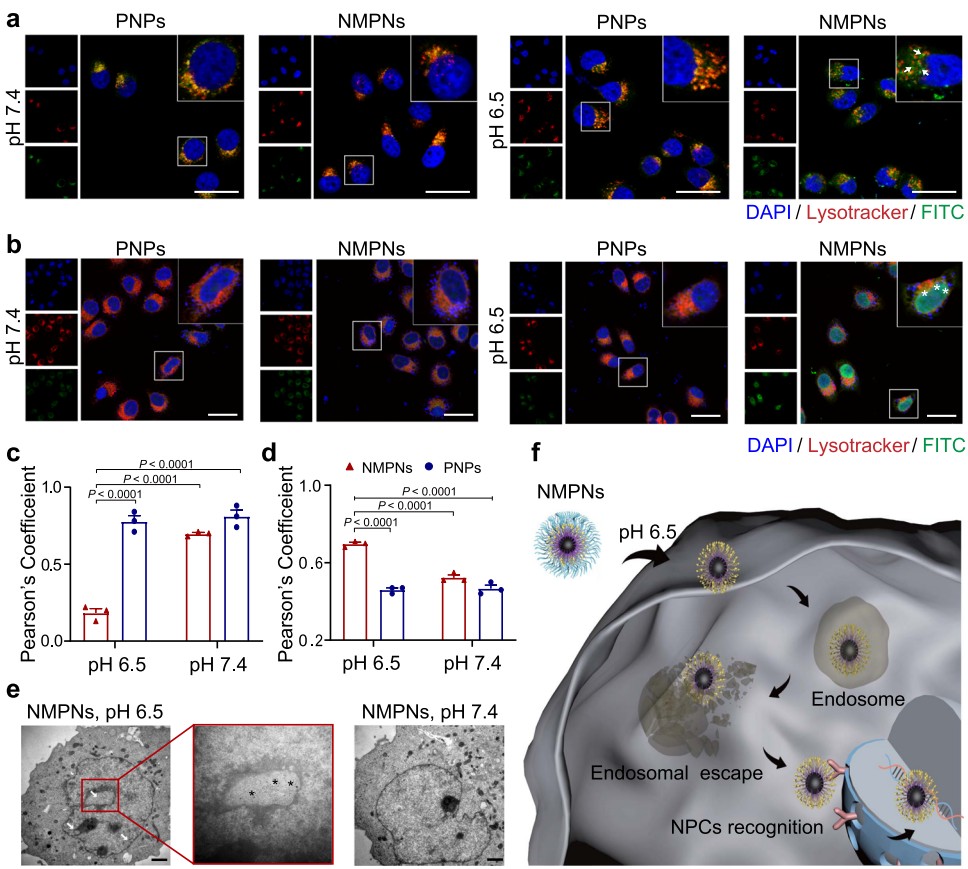

**Fig. 3 | pH-dependent cell nucleus-targeting of NMPNs. a** CLSM images of intracellular localization of FITC-labelled NMPNs and FITC-labelled PNPs after incubated for 6 h under different pH conditions. Scale bar: 20 μm. Arrows indicate NMPNs in the cytoplasm after the escape from endosomes in cisplatin-resistant Huh7 cells. **b** CLSM images of intracellular localization of FITC-labelled NMPNs and FITC-labelled PNPs after incubated for 12 h under different pH conditions. Scale bar: 40 μm. Asterisks indicate NMPNs in the nucleus of cisplatin-resistant Huh7 cells. **c** Quantitative analysis of the colocalization between lysotracker and FITC-labelled NMPNs or PNPs. **d** Quantitative analysis of the colocalization between DAPI and FITC-labelled NMPNs or PNPs. **e** Bio-TEM images of cells after incubation for 12 h with NMPNs under different pH conditions. Arrowheads indicate NMPNs accumulated in the nucleus of Huh7 cells. Black asterisks indicate NMPNs in nucleus. Scale bar: 1 μm. **f** Schematic diagram of the NMPNs to target the tumour cell nucleus. All data are presented as means ± SEM, $n = 3$ independent experiments. Statistical significance was analyzed by one-way ANOVA with multiple comparisons test (**c**, **d**). Source data are provided as a Source Data file.

readily destroy the binding of Pt ion to oxygen atom of the adjacent deoxyribose in single-stranded DNA and simultaneously coordinate with Pt ions to result in a flexible Pt-coordinate structure. The relaxation of Pt-coordinate structure promotes the activation of C4′-sugar-radical, which can bind with adjacent O atom on the phosphate group and form an epoxy group. The C−O binding promotes the single-stranded DNA cleavage near the Pt-DNA binding sites and inhibits the condensation of deoxyribonucleotides at the fracture site (Fig. 4d). Consequently, the remaining single-stranded DNA is much more vulnerable to hydrolysis compared with intact double-stranded DNA free from ROS attack (Fig. 4e), which eventually results in a double-stranded break locating near the Pt-DNA binding sites, thus prohibiting NER by destroying its structural prerequisites[28]. In line with the DFT results, we found that NMPNs can induce the formation of Pt-DNA adducts and DNA breakage in the cisplatin-resistant Huh7 cells (Fig. 5a, c). Moreover, we extracted the DNA fragmentations from NMPNs-treated cisplatin-resistant Huh7 cells and analyzed the location of Pt-DNA binding in the DNA fragmentations. Intriguingly, the result shows that the Pt-DNA binding sites mainly localize at the end of DNA fragmentation (Supplementary Fig. 19).

Considering the DFT results, we further studied the effect of NMPNs on NER to eliminate Pt-DNA adducts in cisplatin-resistant Huh7 cells that are proficient in NER. As mentioned above, NMPNs accumulated in the nucleus can effectively induce the formation of Pt-DNA adducts by releasing Pt ions (Fig. 5a), and facilitate ROS generation in

the nucleus (Fig. 5b) for inducing intensive oxidative cleavage of DNA strands (Fig. 5c), mimicking a biomimetic nuclease. The cotreatment with NAC can compromise the effect of NMPNs and PNPs on the induction of oxidative cleavage of DNA (Fig. 5c and Supplementary Fig. 20). It has been well documented that, after damage recognition by xeroderma pigmentosum groups C (XPC), the bending structure of DNA can form, which promotes the recruitment of XPA to the lesion sites to act as a scaffold for NER proceeding[27,51]. In this regard, we examined the colocalization of XPA with Pt-DNA adduct lesion sites. Intriguingly, as compared to PNPs, nearly no XPA colocalizes with Pt-DNA adducts after NMPNs treatment, indicating the recruitment of XPA is suppressed (Fig. 5d, e and Supplementary Fig. 21a, b). XPF, a pivotal NRE factor for making incisions near DNA lesion sites[52], is also significantly downregulated after the treatment with NMPNs as indicated by the decreased red fluorescence signals corresponding to XPF (Fig. 5f, g) and XPF protein expression level in the nucleus of cisplatin-resistant Huh7 cells (Fig. 5h, i). In comparison, PNPs and cisplatin show no impact on the NER process (Fig. 5h, i and Supplementary Fig. 21c, d). The cotreatment with ROS scavenger NAC shows no interference with the cellular uptake of NMPNs (Supplementary Fig. 22). However, the NMPNs-mediated inhibition of XPA and XPF recruitment in cisplatin-resistant Huh7 cells can be reversed in the presence of NAC, demonstrating that NMPN-generated ROS play a crucial role in the inhibition of NER process by destroying the DNA conformation required for NER (Fig. 5d–g). Altogether, these results indicate that NMPNs can not only

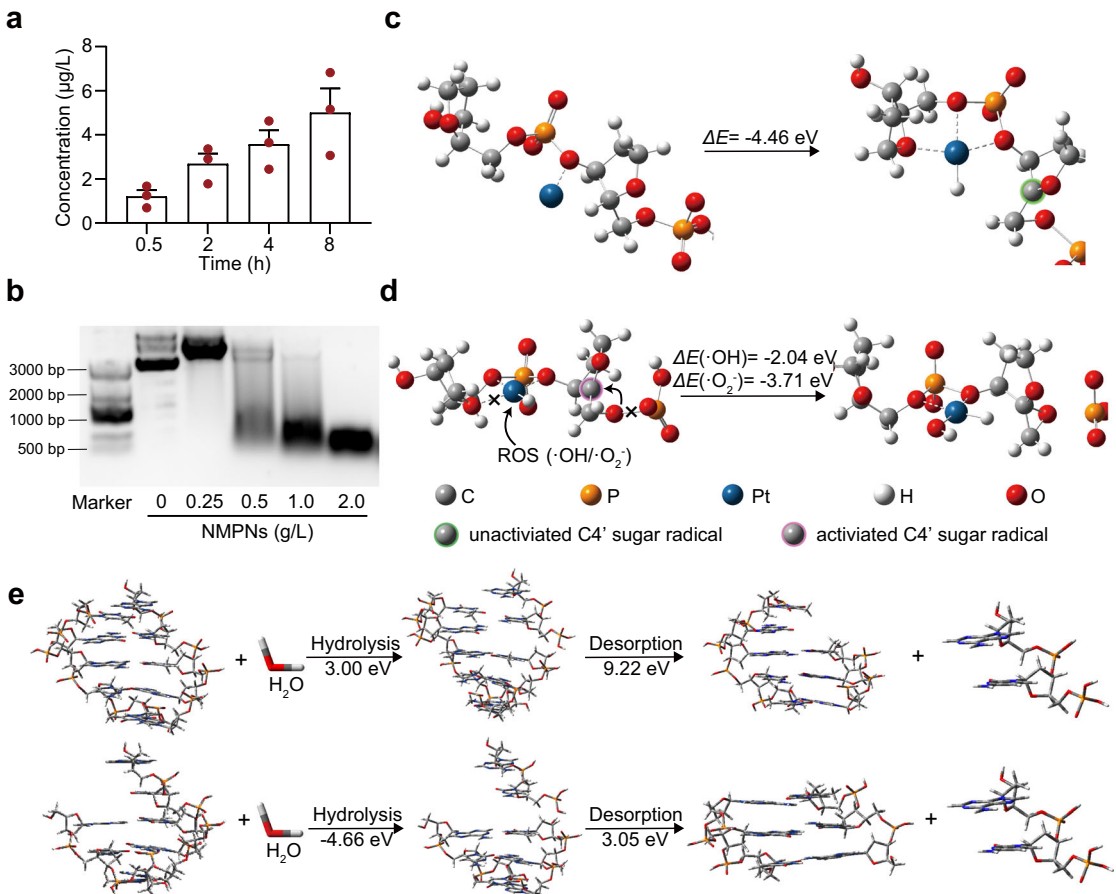

**Fig. 4 | NMPNs induce Pt-DNA adducts formation and oxidative cleavage of DNA strand. a** The release of Pt ions from NMPNs at different time points under neutral conditions after the discharge of mPEG$_{5K}$-AC-CA. **b** The agarose gel electrophoresis of DNA treated with different concentrations of NMPNs. **c** Schematic illustration of Pt ion binding to DNA backbone. **d** Schematic illustration of ROS coordinating with Pt ion and causing oxidative DNA cleavage. **e** Comparison of hydrolysis and desorption process of complete DNA double-stranded fragments (upper) and DNA double-stranded fragments attacked by ROS (lower). The hydrolysis energy of DNA strand fragments attacked by ROS is −4.66 eV, lower than that of the complete double-stranded DNA (3.00 eV). Besides, the desorption energy of the unpaired base strand in ROS-attacked DNA is 3.05 eV, which is also lower than that of the bases with pairs in complete double-stranded DNA (9.22 eV). All data are presented as means ± SEM, $n = 3$ independent experiments. Source data are provided as a Source Data file.

release Pt ions to induce the formation of Pt-DNA adducts, but also effectively generate in situ ROS to destroy the DNA conformation required for NER, inhibiting the recruitment of XPA and XPF for DNA repairing (Fig. 5j).

## NMPNs efficiently eradicate tumour cells in vitro and in vivo by inducing DNA damage irreparable by NER

The anti-tumour effect of NMPNs was further studied in cisplatin-resistant Huh7 cells in which the Pt-DNA adducts generated by cisplatin would be removed by NER[53]. Intriguingly, in contrast to cisplatin and PNPs, NMPNs induce plenty of persistent Pt-DNA adducts without being removed (Fig. 6a), which is likely ascribed to the fact that the NMPNs can behave as a biomimetic nuclease to rupture the very DNA structure required for NER by generating in situ ROS[26]. To fully understand the capability of NMPNs to facilitate the buildup of Pt-DNA adducts, we further investigated the Pt-DNA adducts formation and accumulation in cisplatin-resistant Huh7 cells after incubation with NMPNs in the presence or absence of NAC. After 6-hour or 12-hour incubation, we found that both NMPNs and NMPNs + NAC treatment can initiate the formation of Pt-DNA adducts by releasing Pt ions in the cisplatin-resistant Huh7 cells. However, after 18-h or 24-h incubation, only NMPNs can facilitate the accumulation of Pt-DNA adducts, demonstrating that NMPNs can not only release Pt ions to induce the formation of Pt-DNA adducts, but also effectively generate in situ ROS

to destroy the DNA conformation required for NER[26], promoting the accumulation of cytotoxic Pt-DNA adducts (Fig. 6b, c). Compared with cisplatin, NMPNs cause more severe DNA damage (Fig. 6d), effectively inducing apoptosis to inhibit the proliferation of cisplatin-resistant Huh7 cells (Fig. 6e–g). However, the inhibitory effect of NMPNs on cisplatin-resistant Huh7 cell growth is significantly compromised in the presence of NAC (Supplementary Fig. 23). To further investigate the role of NER in NMPNs-mediated therapeutic effect on cisplatin-resistant Huh7 cells, we checked the cytotoxicity of NMPNs, PNPs, or cisplatin to cisplatin-resistant Huh7 cells (which are proficient in NER) and si*XPA*-transfected cisplatin-resistant Huh7 cells (which are deficient in NER) (Supplementary Fig. 24). The results show that both PNPs and cisplatin exert enhanced cytotoxicity against NER-deficient Huh7 cells as compared to that of NER-proficient cisplatin-resistant Huh7 cells, indicating that NER deficiency can sensitize cisplatin-resistant Huh7 cells to PNPs and cisplatin. However, there is no obvious difference between the cytotoxicity of NMPNs against NER-proficient and NER-deficient Huh7 cells. In NER-deficient Huh7 cells, the XPA expression level is too low to be recruited to the Pt-DNA adducts for repairing[54]. While in NER-proficient Huh7 cells that express a high level of XPA, NMPNs can inhibit the recruitment of XPA to the Pt-DNA adducts by destroying the NER-required DNA bending structure (Fig. 6h). Moreover, NMPMs show no obvious inhibitory effect on the normal cell growth, such as L02 cell, under neutral conditions

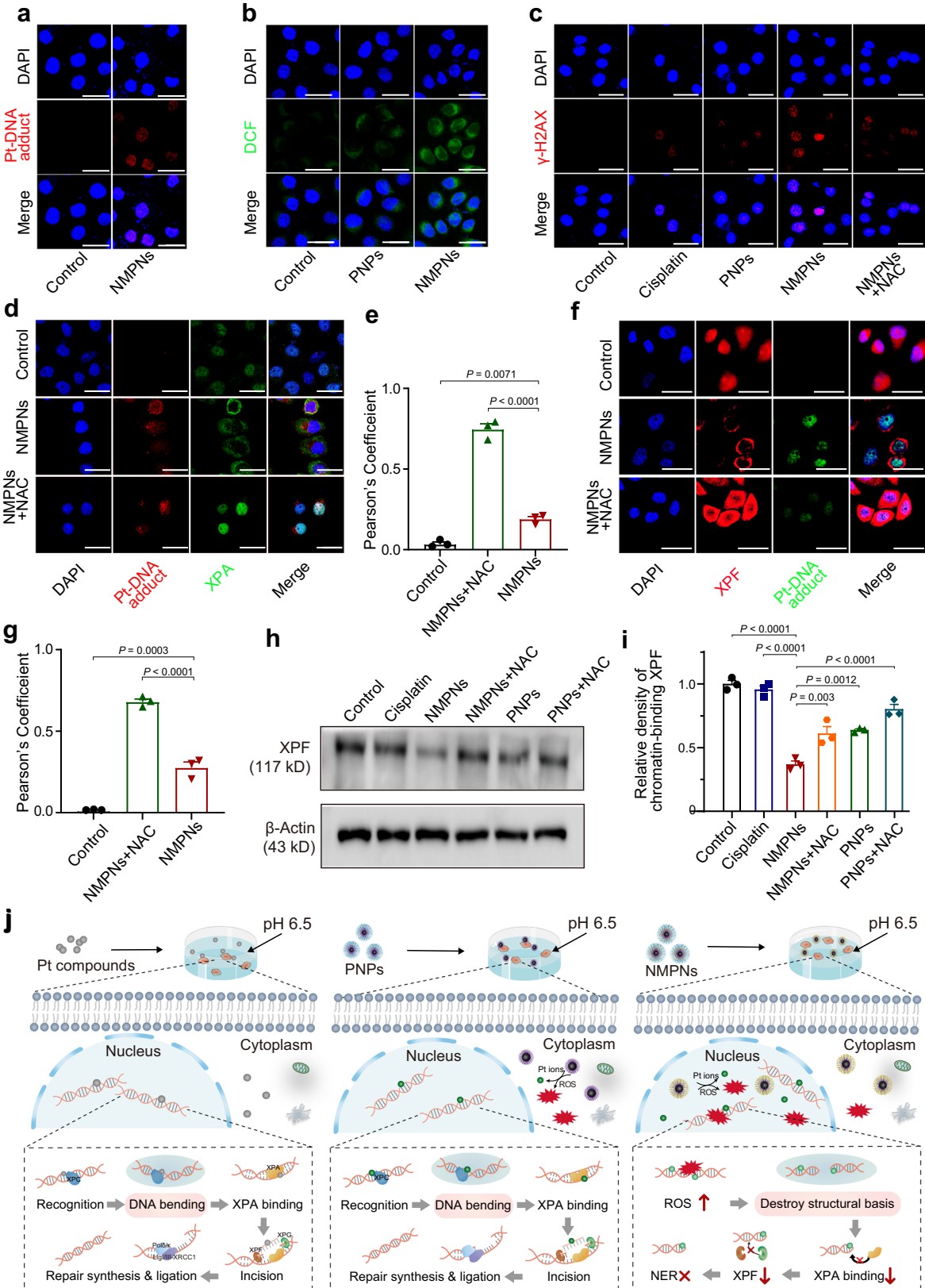

(Supplementary Fig. 25). Altogether, other than cisplatin or PNPs that cannot prominently generate ROS in the nucleus, NMPNs act as a potent anti-tumour agent to induce NER-eluding DNA damage via simultaneous DNA platination and nanozyme-catalyzed cleavage, demonstrating the feasibility of applying nuclease-mimetic activity to Pt-based nanomedicines for combating Pt resistance in cancer.

Finally, the in vivo therapeutic effect of NMPNs was examined in an orthotopic liver cancer mouse model that has been reported to be insensitive to Pt compounds (Fig. 7a)[55]. NMPNs can efficiently accumulate in tumour tissues, likely due to their capability to expose TAT peptides under acidic tumour microenvironment (Supplementary Fig. 26). As shown in Fig. 7b, c, NMPNs significantly suppress the

**Fig. 5 | NMPNs circumvent NER pathway by concurrent of DNA platination and oxidative cleavage. a** The level of Pt-DNA adducts in the nucleus of cisplatin-resistant Huh7 cells after treatment with NMPNs at pH 6.5. Scale bar: 40 μm. Scale bar: 40 μm. **b** ROS levels in the nucleus of cisplatin-resistant Huh7 cells after treatment with PNPs or NMPNs at pH 6.5. Scale bar: 40 μm. **c** Immunofluorescence of the γ-H2AX in cisplatin-resistant Huh7 cells after different treatments at pH 6.5. Scale bar: 40 μm. **d** Immunofluorescence of the XPA and Pt-DNA adducts in cisplatin-resistant Huh7 cells after different treatments at pH 6.5. Scale bar: 40 μm. **e** Quantitative analysis of the colocalization between XPA and Pt-DNA adducts. **f** Immunofluorescence of the XPF and Pt-DNA adducts in cisplatin-resistant Huh7 cells after different treatments at pH 6.5. Scale bar: 40 μm. **g** Quantitative analysis of the colocalization between XPF and Pt-DNA adducts. **h** Western blot analysis of XPF

expression in the nucleus of cisplatin-resistant Huh7 cells after different treatments. **i** Quantitative analysis of XPF expression in cisplatin-resistant Huh7 cells after different treatments at pH 6.5. **j** The schematic illustration of the mechanism underlying NMPNs to induce DNA platination and oxidative cleavage to combat Pt resistance in tumour cells. In comparison to Pt compounds and PNPs that cannot effectively generate ROS in the nucleus, NMPNs can readily accumulate in the nucleus by acidity-induced exposure of TAT peptides, and induce in situ ROS generation to induce DNA oxidative cleavage, thus destroying the DNA conformation required for NER. It hampers the recruitment of XPA and XPF and thus inhibiting NER pathway. All data are presented as means ± SEM, $n = 3$ independent experiments. Statistical significance was analyzed by one-way ANOVA with multiple comparisons test. Source data are provided as a Source Data file.

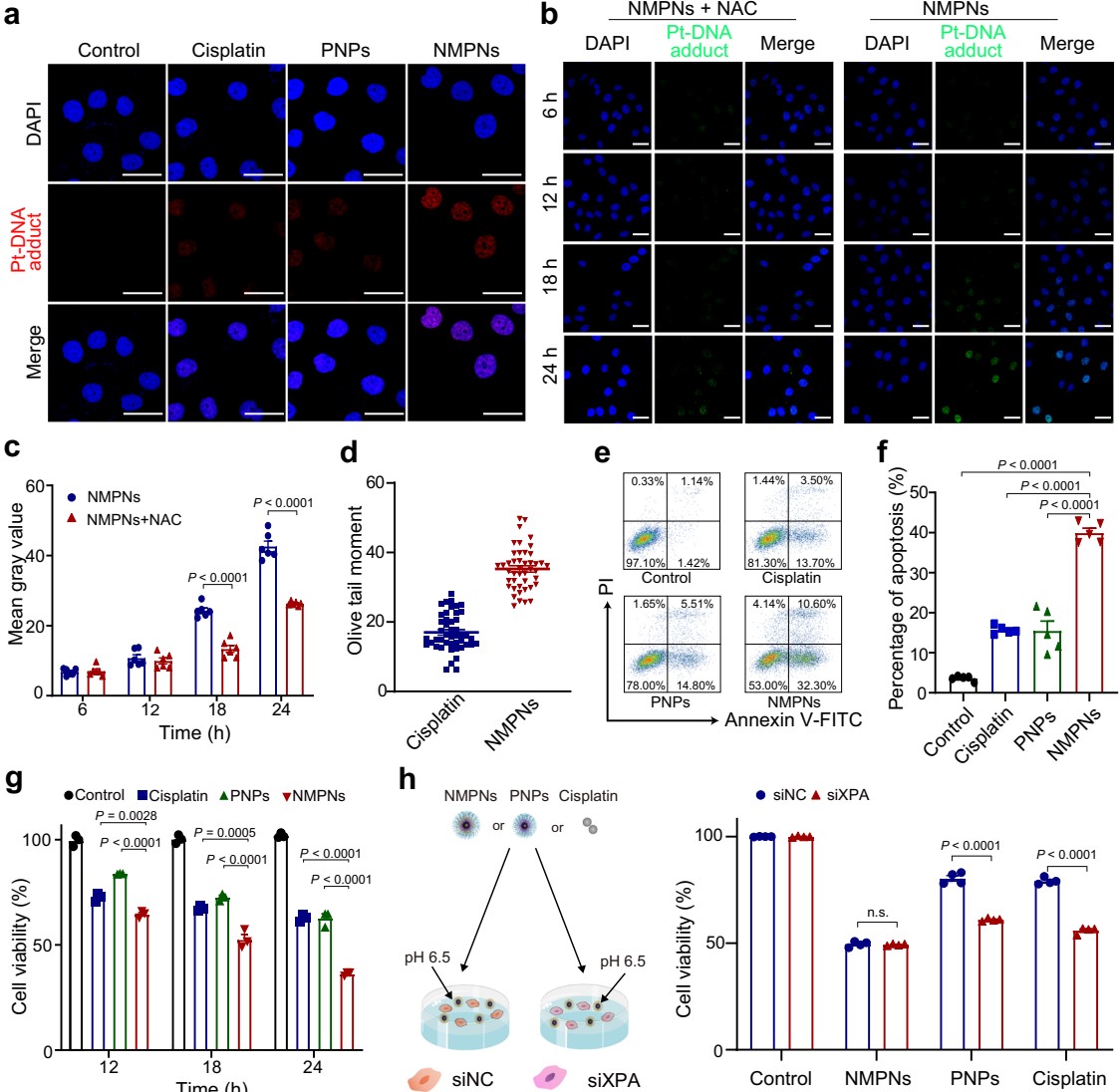

**Fig. 6 | NMPNs promote cisplatin-resistant tumour cell apoptosis by inducing Pt-DNA adducts without NER repairing. a** Immunofluorescence of the Pt-DNA adducts in cisplatin-resistant Huh7 cells after different treatments at pH 6.5. Scale bar: 40 μm. **b** Immunofluorescence of the Pt-DNA adducts in cisplatin-resistant Huh7 cells after treatments with NMPNs or NMPNs+NAC at different time points. Scale bar: 40 μm. **c** Quantitative analysis of Pt-DNA adducts in cisplatin-resistant Huh7 cells after treatments with NMPNs or NMPNs+NAC at different time points. $n = 6$ independent experiments. **d** Quantitative analysis of DNA damage of cisplatin-resistant Huh7 cells after treatment with cisplatin or NMPNs. Cisplatin: $n = 46$;

NMPNs: $n = 44$. **e** Flow cytometry analysis of cell apoptosis after different treatments at pH 6.5. **f** and corresponding quantitative results. $n = 5$ independent experiments. **g** The inhibition effect of cisplatin and NMPNs on cisplatin-resistant Huh7 cells growth at different incubation times. **h** The cell viabilities of cisplatin-resistant Huh7 cells or si*XPA*-transfected cisplatin-resistant Huh7 cells after treatment with NMPNs, PNPs or cisplatin. $n = 4$ independent experiments. All the data are presented as means ± SEM, Statistical significance was analyzed by one-way ANOVA with multiple comparisons test. Source data are provided as a Source Data file.

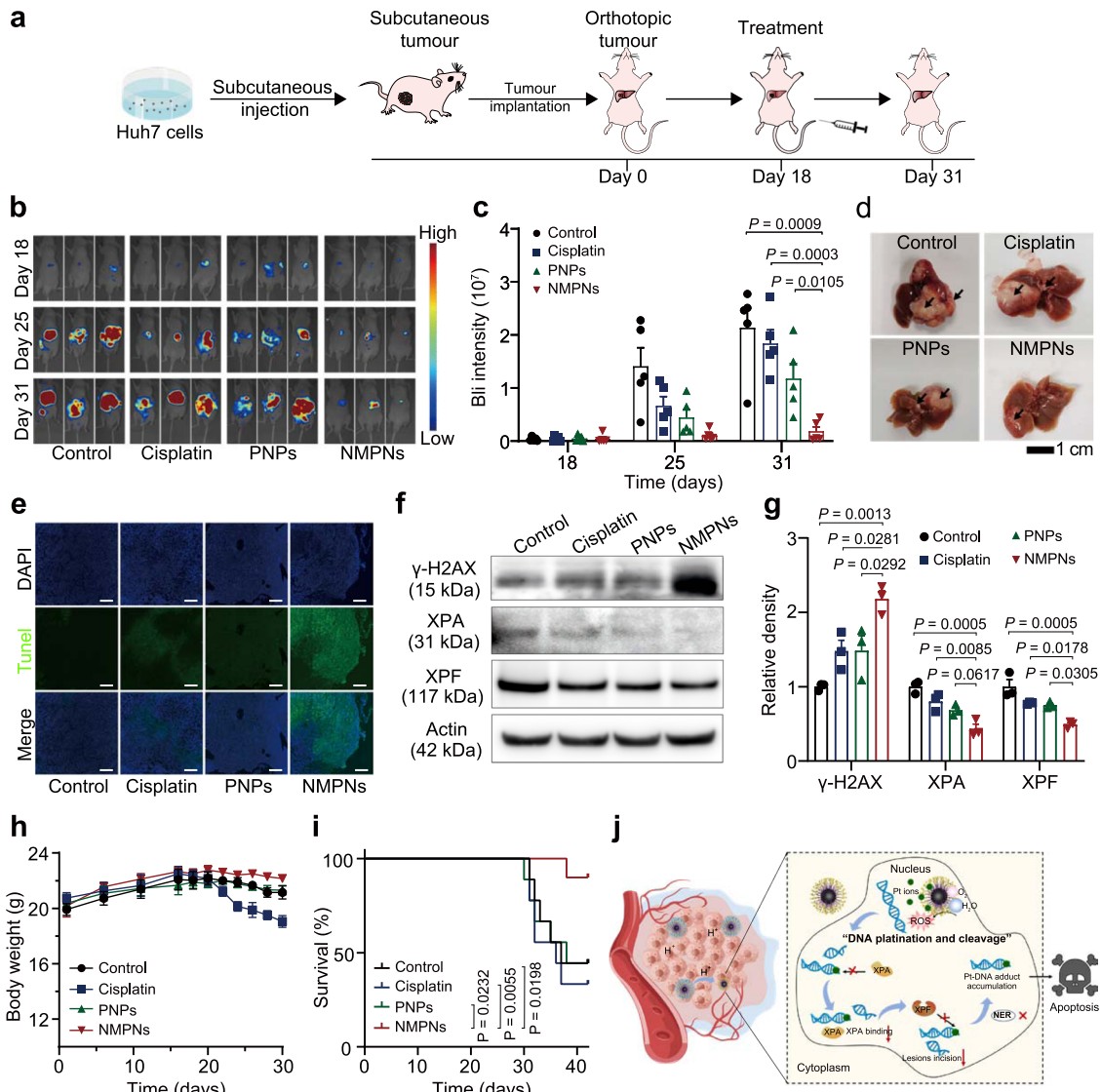

**Fig. 7 | NMPNs suppress tumour in vivo and improve overall therapeutic outcome. a** Schematic illustration of the establishment of orthotopic liver tumour mice model and treatment process. **b** Representative bioluminescence (BLI) images of each group after different treatments. **c** Quantitative BLI signals of tumours after different treatments. $n = 5$ mice. Statistical significances were analyzed by two-tailed unpaired Student's $t$-test. **d** Representative images of liver collected from each treatment group on day 31. Black arrows indicate the tumour tissues. **e** Representative TUNEL and DAPI staining images of tumour tissues. Scale bar: 200 μm. **f, g** Western blot analysis and quantitative analysis of γ-H2AX, XPA and XPF in

the nucleus of tumour cells after different treatments. $n = 3$ independent experiments. **h** The body weight changes of mice with different treatments. $n = 5$ mice. **i** The survival curves of orthotopic liver tumour mice after different treatments. $n = 9$ mice. **j** The schematic illustration of NMPNs to induce Pt-DNA adduct formation and oxidative cleavage for inhibiting the recruitment of XPA and XPF, which nullifies NER pathway and thus suppresses the tumour growth in vivo. All data are presented as means ± SEM. Statistical significances were analyzed by one-way ANOVA with multiple comparisons test (**g, h**) and the log-rank (Mantel−Cox) test (**i**). Source data are provided as a Source Data file.

tumour growth, outperforming cisplatin and PNPs. It is also validated using ex vivo fluorescence imaging of excised liver (Fig. 7d). TdT mediated dUTP Nick-End Labelling (TUNEL) assay shows that, compared to the treatments with cisplatin or PNPs, an increasing number of TUNEL-positive tumour cells can be found after the treatment with NMPNs (Fig. 7e). Western blot analysis of tumour tissue reveals that NMPNs can decrease the expression of XPA and XPF in the nucleus to inhibit the DNA repair mediated by NER, thus inducing more severe DNA damage (Fig. 7f, g). Consistently, the higher signal of green fluorescence corresponding to γ-H2AX is observed in the tumour tissues of mice treated with NMPNs as compared with that of mice treated with cisplatin or PNPs (Supplementary Fig. 27). Considering that NMPNs-mediated ROS generation plays a crucial role in the destruction of the very structure of DNA required for NER, we further

evaluated the ROS level in tumour tissues after the treatments with PNPs, cisplatin or NMPNs. Among which, only NMPNs can significantly enhance the ROS level in tumour tissues (Supplementary Fig. 28). These results indicate that NMPNs can efficiently inhibit tumour growth in the orthotopic liver cancer mouse model by incurring irreparable DNA damage. Moreover, NMPNs show no obvious toxicity to normal tissues around the tumour in the liver and other major organs, suggesting the excellent biosafety of NMPNs (Fig. 7h and Supplementary Figs. 29–31). NMPNs significantly improve the survival rate of orthotopic liver cancer mice, as a result of high therapeutic efficacy and good biocompatibility (Fig. 7i). Overall, NMPNs can efficiently induce DNA platination and oxidative cleavage of the formed Pt-DNA adducts, which disable the downstream of NER-mediated DNA repair, enabling effective treatment of cisplatin-resistant tumours (Fig. 7j).

## Discussion

The resistance to Pt compounds (e.g., cisplatin and carboplatin) almost inevitably occurs in cancer patients, which greatly limits the therapeutic efficacy[6,7]. Pt-based nanodrugs have been exploited to combat cisplatin-resistant tumour[8,9,47,56,57]. Unfortunately, tumour cells frequently upgrade DDR system to handle the DNA damage after repeated treatment with these Pt-based DNA-damaging agents[16,17]. Notably, NER, as one of the most pivotal processes of DDRs, can eliminate the Pt drug-induced toxic Pt-DNA adducts in tumour cells and thus compromising the therapeutic efficacy of Pt-based agents[20,21]. Small molecule inhibitors of NER protein may be used as a countermeasure to overcome the Pt drug resistance of tumours. However, most currently available inhibitors are inefficient and do not substantially benefit the overall survival of patients[23,24], necessitating the exploration of potential strategies to win this arms race.

Herein, by targeting the structural vulnerability of NER process[25,58,59], we design and synthesize NMPNs and demonstrate their application for concurrently inducing DNA platination and oxidative cleavage to combat Pt resistance of cancer. We demonstrate that NMPNs can readily uptake by tumour cells and translocate to the nucleus by selectively exposing TAT peptides in acidic tumour microenvironment. Also, they release Pt ions to bind to DNA strands for efficient induction of Pt-DNA adducts, as well as act as a biomimetic nuclease to induce oxidative single-stranded DNA cleavage near the Pt-DNA binding sites by generating in situ ROS, which subsequently facilitates the hydrolysis of the remaining single-stranded DNA and thus impairing the DNA conformation required for NER. Intriguingly, the recruitment of downstream effectors of NER to DNA lesion sites, including XPA and XPF, is disrupted, allowing excessive accumulation of the cytotoxic Pt-DNA adducts in cisplatin-resistant tumour cells. Consequently, NMPNs remarkably induced the DNA damage and thus apoptosis of cisplatin-resistant tumour cells. Impressively, in contrast to cisplatin with compromised anti-tumour efficacy due to NER-mediated DNA repair, NMPNs can induce irreparable Pt-DNA adducts, and thus effectively suppress tumour growth and prolong the survival of orthotopic liver tumour-bearing mice. Our study provides an innovative Pt nanozyme-based "DNA platination and cleavage" strategy that not only effectively induces the formation of Pt-DNA adducts but also disables the downstream of NER-mediated repair by cleaving DNA, which successfully conquers Pt resistance of tumours.

## Methods

### Materials

Platinum(II) acetylacetonate (97%), oleylamine (80–90%), lithium triethylborohydride, N-Hydroxysuccinimide sodium salt (NHS), N-(3-Dimethylaminopropyl)-N′-ethylcarbodiimide hydrochloride (EDC), cysteamine (CA), tetrahydrofuran (THF), N,N-dimethylformamide (DMF), 2-(N-Morpholino) ethanesulfonic acid (MES), triethylamine (TEA), tetramethylbenzidine (TMB), 5,5-dimethyl-1-pyrroline-N-oxide (DMPO) and acryloyl chloride (AC), Rhodamine B Isothiocyanate (RITC), chlorpromazine, amiloride and methyl-β-cyclodextrin (MβCD) were purchased from Aladdin. N-acetyl cysteine (NAC), Oleic acid (technical grade, 90%), and tetramethylbenzidine were purchased from Sigma-Aldrich. 5-tertbutoxycarbonyl-5methyl-1-pyrroline N-oxide (BMPO) was purchased from APExBIO. Cisplatin injection was purchased from Haosen (Jiangsu) Pharmaceutical Company. mPEG$_{5K}$-OH (Mw = 5000), SH-PEG$_{2K}$-COOH (Mw = 2000), SH-PEG$_{2K}$-RITC (Mw = 2000) and RITC-PEG$_{2K}$-OH (Mw = 5000) were purchased from Punsore Biotechnology Company (China). TAT peptide (GRKKRRQRRR) was purchased from Chinese Peptide Company. Cell counting kit-8 was purchased from Shenzhen Sunview Technology Co., Ltd.

### Characterization

Transmission electron microscopy (TEM) analysis was performed on a transmission electron microscope (Hitachi HT7700, Japan). High-resolution transmission electron microscopy (HRTEM) images were obtained with a FEI Tecnai F20 (FEI, USA) at a voltage of 200 kV. The hydrodynamic size and zeta potential of PNPs and NMPNs were measured by Zetasizer Nano ZS90 (Malvern Instruments, UK) at room temperature. The X-ray photoelectron spectroscopy (XPS) results were obtained via a Thermo Scientific ESCALAB 250 Xi XPS system. X-Ray diffraction (XRD) patterns were performed by using an X-ray diffractometer (PANalytical B.V. X-pert Powder, Netherlands). Inductively coupled plasma mass spectrometry (ICP-MS, PerkinElmer Nex-ION 300X, USA) was utilized to measure the concentration of Pt$_{195}$ under the following test conditions: plasma gas flow rate was 17 L/min. The atomized gas flow rate was 1 L/min. The radio frequency (RF) power was 1300 W. The measured mode was KED mode (He gas flow rate: 3 mL/min). The integration time was 1000 ms. UV-vis absorption spectra were measured by a UV-Vis-NIR Spectrophotometer UV-3600 (Shimadzu, Japan). In vivo bioluminescence images were received by a VISQUE InVivo Elite imaging system. Catalase (CAT)-like activity assays of NMPNs was measured by an oxygen electrode on Multi-Parameter Analyzer (JPSJ-606L, Leici China).

### Synthesis of PtNCs

Pt nanoclusters (PtNCs) of ~2.5 nm in diameter were synthesized via the thermal decomposition method. In a typical process, 0.4 g of Platinum(II) acetylacetonate and 450 μL of oleic acid were mixed with 35 mL of oleylamine in a flask. The mixture was first heated to 70 °C and stirred vigorously for 1 h under vacuum. Then, the reaction mixture was heated to 170 °C with a constant heating rate of 5 °C/min under Ar atmosphere, and 1 mL of lithium triethylborohydride was injected after the temperature reached 170 °C. Next, the mixture was aged for 3 min. After the reaction, the mixture was rapidly cooled down to room temperature and precipitated with acetone. Finally, the Pt nanoclusters were separated by centrifugation and dispersed in chloroform.

### Synthesis of SH-PEG$_{2K}$-COOH modified PtNCs and fluorescein labelling of SH-PEG$_{2K}$-COOH modified PtNCs

PtNCs (5 mg) dissolved in chloroform were added into the solution of 5 mL of chloroform containing SH-PEG$_{2K}$-COOH (3 mg/mL) dropwise. Then, the solvent was evaporated by a rotary evaporator to remove the chloroform. 1 mL of deionized water was added to obtain SH-PEG$_{2K}$-COOH modified PtNCs. To prepare fluorescein labelled SH-PEG$_{2K}$-COOH modified PtNCs, 1 mL of chloroform containing 5 mg of PtNCs was added to the 5 mL chloroform containing 7.5 mg of SH-PEG$_{2K}$-COOH and 7.5 mg of SH-PEG-FITC. Then, the solvent was evaporated by a rotary evaporator to remove the chloroform. After which, 1 mL of deionized water was added to obtain SH-PEG$_{2K}$-COOH modified PtNCs with fluorescein labelling.

### Synthesis of PEG ligand equipped with acid-labile β-thiopropionate linker (mPEG$_{5K}$-AC-CA)

One gram of mPEG$_{5K}$-OH (Mw = 5000) was dissolved in 15 mL of Ar-purged tetrahydrofuran (THF) under ice-water bath. Then, 460 μL of TEA and 200 μL of acryloyl chloride were slowly injected into the mixture. The mixture was stirred for 16 h at 30 °C. Then, 10 mL of N,N-dimethylformamide (DMF) containing 0.3 g cysteamine was added. The mixture was then stirred at 30 °C for 24 h. The resulting product was dialyzed against distilled water for 2 days and then freeze-dried.

### Synthesis of PEG ligand equipped with acid-labile β-thiopropionate linker and RITC (RITC-mPEG$_{5K}$-AC-CA)

The RITC-PEG-OH (Mw = 5000, 80 mg, 8 μmol) was dissolved in 2 mL of THF and then purged with Ar under ice-water bath. Then, 200 μL of TEA and 200 μL of acryloyl chloride were added. The solution was then stirred at room temperature for 16 h. After which, 10 mL of DMF containing 0.3 g cysteamine was added. The mixture was then stirred for 24 h at 30 °C and dialyzed with dialysis bag (Mw = 3500) for 2 days to

remove the redundant reagents and the product was obtained after freeze-drying.

## Preparation of Pt NCs@mPEG$_{5K}$-AC-CA (PNPs) and nuclease-mimetic Pt nanozymes (NMPNs) with nucleus-targeting capability

To prepared PNPs, 2 mg SH-PEG$_{2K}$-COOH modified PtNCs, 5 mg of EDC and 5 mg of NHS were dissolved in 3 mL of MES buffer and then stirred at 30 °C for 15 min. After which, 3 mg of mPEG$_{5K}$-AC-CA was added into the mixture and stirred for 24 h at room temperature. The excess EDC was removed by washing with distilled water for several times. To prepare NMPNs, 2 mg of SH-PEG$_{2K}$-COOH modified PtNCs, 5 mg of EDC and 5 mg of NHS were dissolved in 3 mL of MES buffer and then stirred at a temperature of 30°C for 15 min. After which, 0.5 mg of TAT peptide was added and stirred for 6 h before the addition of 2.5 mg of mPEG$_{5K}$-AC-CA. The mixture was stirred at room temperature for another 16 h. The fluorescein labelled NMPNs and PNPs were prepared in the same way by using fluorescein labelled PEG-COOH modified PtNCs.

## Measurement of the released Pt ions from NMPNs

The Pt ion release behaviour of NMPNs after the discharge of mPEG$_{5K}$-AC-CA at different time points (0.5, 2, 4, 8 h) was analyzed by using ICP-MS. First, NMPNs were treated with PBS (pH 6.5) for 6 h. Then, PBS (pH 6.5) was removed by ultrafiltration and the NMPNs were collected and dispersed in PBS (pH 7.4). After which, 1 mL of PBS (pH 7.4) containing 500 μg of the obtained NMPNs was placed in a dialysis bag and dialyzed against 10 mL of PBS (pH 7.4) at 37 °C. The amount of the released Pt ions was quantified by using ICP-MS at 0.5, 2, 4, 8 h, respectively.

## The ROS generation capacity of NMPNs

The ROS generated by NMPNs were determined using TMB oxidation experiments. Typically, 20 μL of NMPNs (1 mg/mL) were added to HAc/NaAc buffer (pH 7.4, 0.2 M, 1.96 mL) with/without 10 μL of H$_2$O$_2$ (0.5 M). Then the mixture was purged with Ar before adding 20 μL TMB (50 mM). After which, UV-Vis absorption spectra were measured at different reaction time points. Furthermore, to verify the type of generated ROS, 20 μL of NMPNs (1.5 mg/mL) were added to HAc/NaAc buffer (pH 7.4, 0.2 M, 1.96 mL), and 100 μL of mannite (50 mM) was added into the mixture to identify •OH. To identify or •O$_2^-$, 20 μL of NMPNs (1.5 mg/mL) were added to HAc/NaAc buffer (pH 7.4, 0.2 M, 1.96 mL) containing 10 μL of H$_2$O$_2$ (0.5 M), and 100 μL of SOD (4000 unit/ml) was added into the aforementioned mixture to identify •O$_2^-$. Then, oxidation of TMB was measured at different reaction time points by UV-Vis detection at 652 nm.

## Kinetic assay for NMPNs with TMB as substrate

Kinetic measurement for NMPNs with TMB as substrate was carried out at room temperature in 0.2 M HAc/NaAc solution (pH 7.4). In a typical measurement, 20 μL of TMB (final concentration 0.1, 0.2, 0.3, 0.4, 0.5, 0.6, 0.8, 1.0, 1.25 mM) was added into 2 mL of HAc/NaAc buffer (pH 7.4, 0.2 M) containing 20 μL of NMPNs (1 mg/mL) and UV–Vis absorption spectra were detected at a 5-s interval for a total of 2.5 min. The apparent kinetic parameters were calculated according to the Michaelis-Menten curve by GraphPad Prism 8.0.2 (GraphPad Software).

## Kinetic assay for NMPNs with TMB and H$_2$O$_2$ as substrates

Kinetic measurements for NMPNs with TMB and H$_2$O$_2$ as substrates were carried out at room temperature in 0.2 M HAc/NaAc solution (pH 7.4) and detected by the oxidation of TMB in the presence of H$_2$O$_2$. Typically, the kinetic measurements of NMPNs using TMB as the substrate were performed by adding 20 μL of TMB (final concentration 0.1, 0.2, 0.3, 0.4, 0.5, 0.6, 0.7, 0.8, 0.9, 1.0 mM) into 2 mL of HAc/NaAc buffer (pH 7.4, 0.2 M) containing 20 μL of NMPNs (1 mg/mL) and 10 μL of H$_2$O$_2$ (final concentration 2.5 mM) and UV–Vis absorption spectra were detected at a 5-s interval for a total of 2.5 min. The kinetic measurements of NMPNs using H$_2$O$_2$ as the substrate were performed by adding 10 μL of H$_2$O$_2$ (final concentration 0.03125, 0.0625, 0.125, 0.25, 0.5, 0.75, 1.0, 1.25, 1.5, 1.75 mM) into 2 mL of HAc/NaAc buffer (pH 7.4, 0.2 M) containing 20 μL of NMPNs (1 mg/mL) and 20 μL of TMB (final concentration 0.5 mM) and UV–Vis absorption spectra were detected at a 5-s interval for a total of 2.5 min. The apparent kinetic parameters were calculated according to the Michaelis-Menten curve by GraphPad Prism 8.0.2 (GraphPad Software).

## NMPNs-induced ROS generation measured by ESR

The generation of ROS was measured by the electron spin resonance (ESR) spectroscopy using DMPO and BMPO as probes. In a typical test, 4 μL of NMPNs (1 mg/mL) were added into PBS (pH 7.4) containing BMPO (5 μL, 50 mM), and the final volume of the mixed solution is 200 μL. The spin trap BMPO was used to confirm the formation of •O$_2^-$. In another test, 4 μL of NMPNs (1 mg/mL) were added into PBS (pH 7.4) containing 5 μL H$_2$O$_2$ (1 M) and DMPO (50 μL, 500 mM), and the final volume of the mixed solution is 200 μL. Then, the spin trapped DMPO was recorded to confirm the generation of •OH.

## Study of DNA cleavage after NMPNs treatment

NMPNs were treated with PBS (pH 6.5) for 6 h. Then, PBS (pH 6.5) was removed by ultrafiltration and the NMPNs were collected. Three hundred nanograms of plasmid DNA (pet-32a+) were incubated with different amounts of the obtained NMPNs (2.5, 5, 10, 20 g/L) in PBS (pH 7.4) at 25 °C for 12 h. These samples were mixed with 0.20 volume of the desired 6× loading buffer and slowly added into the 1% agarose gel slots in an electrophoresis tank containing 1× TBE electrophoresis buffer respectively, which were then applied with a voltage of 90 V until the DNA samples migrate a sufficient distance through the gel. Lastly, the gel was stained for 30 min by using SYBR dyes at room temperature and an image of the gel was captured under UV transillumination.

## Study of kinetic of NMPNs-induced DNA cleavage

300 ng of Plasmid DNA were incubated with NMPNs with different concentrations (0.42, 0.84, 0.168, 0.336 μM) in PBS (pH 7.4) for 3, 9, 12 h, respectively. Then, the integrity of DNA was investigated by using gel electrophoresis. The band intensities were quantified using Image J software (version 1.8.0) to calculate the percentage of cleaved DNA. The percentage of cleaved DNA vs. time followed pseudo-first-order kinetic profiles. Then, GraphPad Prism Software Version 8.0 was used for the non-linear curve fitting.

## Cell culture

L02 cells (CL-0111) and Huh7 (CL-0120) cells were obtained from iCell Bioscience Inc. (Shanghai, China). Huh7 cells were obtained from human liver cancer tissue and identified by a Short Tandem Repeat (STR) method. L02 cells and Huh7 cells tested negative for mycoplasma contamination. Huh7 cells were cultured in Dulbecco's Modified Eagle medium (DMEM) supplement with 10% foetal bovine serum and 1% penicillin/streptomycin in an atmosphere of 95% humidified air and 5% CO$_2$ at 37 °C.

## The analysis of locations of Pt-DNA binding in the DNA

Cisplatin-resistant Huh7 cells were cultured in 6-well plates with 2 mL culture medium. After adherent to the wall, NMPNs (20 μg/mL, pH 6.5) were added. After 24 h incubation, cells were collected and quickly resuspended in 500-600 μL of ice-cold PBS. Then, 2.5 μL of cell resuspension was spotted onto a pre-cleaned glass slide and mixed with 7.5 μL of spreading buffer (0.5% SDS in 200 mM Tris-HCl (pH 7.4), 50 mM EDTA). After 10 min, the slides were tilted to 15° to spread DNA

fibres along the length. Then, air-dry the DNA spreads and fixed in 3:1 methanol/acetic acid for 20 min at −20 °C. After washing with PBS three times, slides were blocked with 1% BSA in PBS for 30 min at room temperature and incubated with anti-cisplatin modified DNA antibodies (GeneTex, GTX17412, dilution 1:100) to detect Pt-DNA adduct. After 1 h of incubation, slides were washed with PBS three times and stained with Alexan Fluor 488-conjugated AffiniPure Rabbit anti-Rat IgG (H + L) (Boster, BA1129, dilution 1:200) for 2 h at room temperature in the dark. After washing with PBS three times, slides were incubated with DAPI solution for 15 min in the dark. Finally, the cells were detected by using CLSM.

### DFT calculation of catalytic reaction on metallic and oxidized Pt of PtNCs

The metallic Pt bulk structure was first optimized using the cubic unit cell from the database of Materials Project [https://legacy.materialsproject.org/materials/mp-126/]. Then the Pt bulk structure was cleaved to (1 1 1) slab and modelled with the (4 × 6) periodically repeated supercell, consisting of two atomic layers, with a vacuum space of 15 Å. The oxidized Pt bulk structure was gained from Materials Project [https://materialsproject.org/materials/mp-1604?chemsys=Pt-O], which then was cleaved to (2 1 0) slab and modelled with the (2 × 1) periodically repeated supercell, consisting about 5-Å-thick atomic layers, with a vacuum space of 15 Å.

The calculations were performed using the DFT method with a standard solid-state pseudopotential (SSSP) efficiency as implemented in the Quantum-Espresso package. The kinetic energy cutoff for wave functions was set as 60 Ry, and the charge density cutoff for SSSP efficiency was set as 480 Ry, as SSSP efficiency recommends. The energy and force convergence criterion for all geometry optimizations were set as $1.0 \times 10^{-4}$ arb. units and $1.0 \times 10^{-3}$ arb. units, respectively. 2 × 2 × 1 Monkhorst-Pack k-point mesh samplings were used for the metallic and oxidized Pt slab.

### DFT calculation of ROS coordinating with Pt ion and causing oxidative DNA cleavage

The DNA model was obtained from the X-ray diffraction data of self-assembly DNA short fragments in Protein Data Bank (PDB ID: 6WQG). The DNA sequence of one strand from 5′-terminal to 3′-terminal is AGTCA, and the other strand is TGACT. The DFT calculation was operated by Gaussian 09 software using B3LYP method[60]. The basis sets of C, O, H, and P elements are 6-31G*, and the basis set of Pt element is Lanl2TZ from Basis Set Exchange software. The configurations during the DFT calculation are observed using GaussView 6.0 software.

### Endocytic pathway of NMPNs

To study the cellular uptake mechanisms of NMPNs by cisplatin-resistant Huh7 cells, three specific endocytic inhibitors were used: (1) chlorpromazine, an inhibitor to probe clathrin-mediated endocytosis; (2) amiloride, an inhibitor of micropinocytosis; (3) methyl-β-cyclodextrin (MβCD), an inhibitor of caveolin-mediated endocytosis. Specifically, cisplatin-resistant Huh7 cells were pre-incubated in serum-free RPMI DMEM medium with chlorpromazine (40 μM, 30 min), amiloride (100 μM, 30 min) or MβCD (5 mM, 30 min), respectively. The medium was then changed to fresh serum-free medium containing the inhibitors plus NMPNs (20 μg/mL) and further incubated for 6 h at 37 °C. The cells were collected for quantitative analysis. The gate strategies were shown in Supplementary Fig. 32a.

### Cell uptake and subcellular localization

Confocal laser scanning microscope (CLSM) imaging: Huh7 cells were cultured on confocal dishes (8 × 10³ cells/well) with DMEM containing 10% foetal bovine serum. After 24 h incubation, the medium was replaced with 2 mL of DMEM containing fluorescein labelled NMPNs or

PNPs with a concentration of 20 μg/mL at different pH conditions (6.5 or 7.4). For the analysis of endosomal escape and nucleus-targeting capability of NMPNs, after incubation for 6 and 12 h, the cells were washed with PBS twice and stained with Lysotracker Red for 40 min. Then, the cells were gently washed with PBS for 3 times and treated with 4% paraformaldehyde fix solution for 15 min. After being washed with PBS for 3 times, the cells were stained with DAPI for 6 min and observed by using CLSM (Olympus FV1200, Japan).

### Electron microscopy analysis

Huh7 cells were incubated with PNPs or NMPNs (20 μg/mL) at different pH conditions (6.5 or 7.4) for 12 h. Then, the cells were washed twice with PBS and detached by incubating with trypsin for 5 min. The cell suspension was centrifuged and washed with PBS for 3 times. After removal of the incubation medium, the cells were fixed by glutaraldehyde at room temperature, and rinsed with PBS and dehydrated through a graded ethanol series, then cleared with propylene oxide. Finally, the cell samples were cut and transferred to the copper grid for observations under a Hitachi H-7650 TEM at an accelerating voltage of 80 kV.

### The concentration of NMPNs in cisplatin-resistant Huh7 cells in the presence or absence of NAC

Cisplatin-resistant Huh7 cells were treated with NMPNs (20 μg/mL) in the presence or absence of NAC for 24 h at pH 6.5, respectively. Then the cells were mixed with aqua regia and incubated at 60 °C for 72 h. After dilution and filtration, the Pt concentrations were quantified by ICP-MS analysis.

### The concentration of NMPNs in the nucleus of cisplatin-resistant Huh7 cells

Cisplatin-resistant Huh7 cells were treated with NMPNs or PNPs (20 μg/mL) for 24 h at pH 7.4 or pH 6.5, respectively. The cell nucleus was extracted and then mixed with aqua regia and incubated at 60 °C for 72 h. After dilution and filtration, the Pt concentrations were quantified by ICP-MS analysis.

### In vitro cytotoxicity

Cisplatin-resistant Huh7 cells were seeded into 96-well plates (1 × 10⁴ cells/well) for 24 h and treated with cisplatin, PNPs or NMPNs with a concentration of 20 μg/mL for 6 h at pH 6.5. Then, the incubation medium containing cisplatin, PNPs or NMPNs were removed and replaced with fresh medium. After 6, 12 and 18 h, the cells were incubated with cell counting kit-8 (CCK-8) solution for each well and incubated for another 3 h, the absorbance of each well was measured at 450 nm by a microplate reader.

Cisplatin-resistant Huh7 cells were seeded into 96-well plates (1 × 10⁴ cells/well) for 24 h and treated with NMPNs (20 μg/mL) in the presence or absence of NAC for 6 h at pH 6.5. Then, the cells were incubated with CCK-8 solution for each well and incubated for another 3 h, the absorbance of each well was measured at 450 nm by a microplate reader.

L02 cells were seeded into 96-well plates (1 × 10⁴ cells/well) for 24 h and treated with NMPNs (20 μg/mL) for 6 h at pH 7.4. Then, the cells were incubated with CCK-8 solution for each well and incubated for another 3 h, the absorbance of each well was measured at 450 nm by a microplate reader.

### Flow cytometry analysis

Cisplatin-resistant Huh7 cells were cultured in 12-well plates (5 × 10⁴ cells/well). Cisplatin, PNPs or NMPNs diluted in 1 mL of DMEM (20 μg/mL, pH 6.5) were added and incubated for 24 h and washed with PBS for 3 times. After which, cells were collected and resuspended with binding buffer and stained with FITC-Annexin V and propidium iodide (PI) for 15 min at room temperature. After which, the binding

buffer was added again for the analysis by measuring fluorescence intensity on a flow cytometer (Beckman cytoflex, USA). The gate strategies were shown in Supplementary Fig. 32b.

## Immunofluorescence detection of γ-H2AX

The cisplatin-resistant Huh7 cells were treated with cisplatin, NMPNs, NMPNs+NAC, PNPs or PNPs+NAC for 24 h at pH 6.5. Then, cells were fixed by 4% paraformaldehyde for 15 min at room temperature and permeabilized with 0.5% Triton X-100 for another 20 min, and then blocked with 5% bovine serum albumin in PBS for 1 h at room temperature. After which, the cells were incubated with primary antibodies at room temperature for 1 h: anti-γ-H2AX (dilution 1:100, ab81299) from Abcam (Cambridge, UK). Then, the cells were incubated with Alexa Fluor 488-conjugated goat anti-rabbit IgG (H + L) from Boster Biological Technology Co., Ltd. for 1 h at room temperature. In addition, the DAPI solution was added for 15 min in the dark for nucleus staining. Finally, the cells were detected by using CLSM.

## Western blot analysis of chromatin-bound XPF

The cisplatin-resistant Huh7 cells were cultured in 12-well plates (5 × 10⁴ cells/well), and treated with 20 μg/mL cisplatin, 20 μg/mL PNPs, 20 μg/mL NMPNs, 20 μg/mL NMPNs plus 60 μM N-acetyl cysteine (NAC) or 20 μg/mL PNPs plus 60 μM NAC at pH 6.5 respectively. After 24 h of incubation, cells were washed with PBS for 3 times and treated with RIPA buffer containing protease inhibitors to obtain whole-cell protein extracts. Then, Subcellular Protein Fractionation Kit (Thermo Scientific) was exploited to prepare chromatin-bound protein[61]. The concentration of proteins was measured through the BCA assay. After which, protein samples with the same concentration were separated by 4–12% SDS-PAGE gels and further transferred to PVDF membranes. Then, the membranes were incubated in TBST containing 5% skimmed milk at room temperature for 1 h and anti-γ-H2AX (dilution 1:100, ab81299) from Abcam (Cambridge, UK), anti-XPF (1:1000, OM201253) from Omnimabs (California, USA) or anti-Actin (1:1000, AA128), XPA (1:1000, AF5336) and β-actin (1:1000, AF2811) from Beyotime Biotechnology (Shanghai, China) at 4 °C overnight. Finally, membranes were incubated with secondary antibodies at room temperature for 1.5 h. Then, the signals of immunoreactive bands were detected via a chemiluminescence system (Amersham Imager 680) and further quantified by using Image J.

## Immunofluorescence detection of γ-H2AX and anti-cisplatin modified DNA

The Huh7 cells were treated with cisplatin, PNPs, NMPNs or NMPNs plus NAC for 24 h at pH 6.5. The control group was set for no treatment. Then, cells were fixed by 4% paraformaldehyde for 15 min at room temperature and permeabilized with 0.5% Triton X-100 for another 20 min, and then blocked with 5% bovine serum albumin in PBS for 1 h at room temperature. After which, the cells were incubated with primary antibodies at room temperature for 1 h: anti-γ-H2AX (dilution 1:100, ab81299) from Abcam (Cambridge, UK) or anti-cisplatin modified DNA (dilution 1:100, GTX17412) from GeneTex (California, USA). Then, the cells were incubated with Alexa Fluor 555-conjugated goat anti-mouse IgG (H + L), Alexa Fluor 555-conjugated goat anti-rabbit IgG (H + L) or Alexa Fluor 488-conjugated goat anti-rabbit IgG (H + L) from Beyotime Biotechnology (Shanghai, China) for 1 h at room temperature. In addition, the DAPI solution was added for 15 min in the dark for nucleus staining. Finally, the cells were detected by using CLSM.

## Immunofluorescence detection of XPA or XPF and anti-cisplatin modified DNA

The Huh7 cells were treated with PNPs, NMPNs or NMPNs plus NAC for 24 h at pH 6.5. The control group was set for no treatment. Then, cells were fixed by 4% paraformaldehyde for 15 min at

room temperature and permeabilized with 0.5% Triton X-100 for another 20 min, and then blocked with 5% bovine serum albumin in PBS for 1 h at room temperature. After which, the cells were incubated with primary antibodies at room temperature for 1 h: anti-XPA (1:100, AF5336) from Beyotime Biotechnology (Shanghai, China) or anti-XPF (1:100, OM201253) from Omnimabs (California, USA) or anti-cisplatin modified DNA (dilution 1:100, GTX17412) from GeneTex (California, USA). Then, the cells were incubated with Alexan Fluor 488-conjugated rabbit anti-rat IgG (H + L) (dilution 1:200, BA1129) or Alexa Fluor 488-conjugated goat anti-rabbit IgG (H + L) (dilution 1:200, BA1127) or Alexa Fluor 555-conjugated goat anti-mouse IgG (H + L) (dilution 1:200, BA1126) from Boster Biological Technology Co., Ltd. for 1 h at room temperature. In addition, the DAPI solution was added for 15 min in the dark for nucleus staining. Finally, the cells were detected by using CLSM.

## ROS generation in vitro

The cisplatin-resistant Huh7 cells were treated with PNPs or NMPNs (20 μg/mL) at pH 6.5 for 24 h. Then, the cells were washed with PBS for 3 times and incubated with DCFH-DA for 30 min and subsequently incubated with DAPI solution. The level of intracellular ROS was analyzed by using CLSM.

## Single cell gel electrophoresis assays

The Huh7 cells were treated with cisplatin or NMPNs (20 μg/mL) at pH 6.5 for 24 h. Then, the cells were collected and resuspended into PBS with a concentration of 1 × 10⁴ cells/mL. Twenty microlitres of cell suspension was mixed with 75 μL low melting agarose solution and then dropped into the first gel of cooled normal melting agarose solution which was spread on a glass slide. Another coverslip was covered on the gel and moved after the gel was cooled. Then, glass slides were placed into the cooled lysis buffer for 2 h at 4 °C, which were further washed with DI water for several times. After which, the obtained glass slides were put into an electrophoresis chamber filled with the alkaline electrophoretic solution for 1 h, and electrophoresis was carried out for 30 min at 25 V. Next, these glass slides were immersed in cold Tris-HCL (pH 7.5) solution at 4 °C for 30 min and dyed with EB overnight in the dark, and images were taken with a fluorescence microscope. The images were processed by Image J and the tail moment was calculated by CASPLAB soft.

## XPA knockdown by small interfering RNA (siRNA)

To silence the gene expression of *XPA*, *XPA*-specific siRNA (si*XPA*) and control siRNA (siNC) were obtained from Genepharma (Shanghai, China). Cisplatin-resistant Huh7 cells were seeded into 6-well plates (2 × 10⁵ cells per well) overnight and then transfected with siRNA (si*XPA* or siNC) mixed with Lipofectamine 2000 transfection reagent, according to the recommended protocols by the manufacturer. The medium (Opti-MEM) was replaced at 6 h post-transfection. After 24 h, the cells were treated with cisplatin, PNPs, and NMPNs. SiRNA sequences of *XPA*-Homo (catalogue numbers: A01003): (1) sense (5′−3′)-GACCUGUUAUGGAAUUUGATT, anti-sense (5′−3′)-UCAAAUUCCA UAACAGGUCTT; (2) sense (5′−3′)- GGAGACGAUUGUUCAUCAATT, anti-sense (5′−3′)- UUGAUGAACAAUCGUCUCCTT.

## The orthotopic xenograft mouse model establishment and drug treatment

All animal use and studies were performed according to the relevant ethical regulations. The procedures conducted on animals were approved by the Institutional Animal Care and Use Committee (IACUC) of Zhejiang University (IACUC number: IACUC-s19-026, project number: 19NGYX087Nu). The maximal tumour size/burden permitted by the IACUC is 20 mm in diameter; no mice met this criterion. Following IACUC guidelines from Zhejiang University, weight loss of more than

20%, body conditioning score (BCS) of 2 or less, or mice exhibiting signs of hunched posture, impaired locomotion or respiratory distress are criteria followed for prompt euthanasia by $CO_2$ gas. BABL/c athymic nude mice (female, 2–3 weeks, not specifically transgenic) were obtained from Shanghai SLAC Laboratory Animal Co., Ltd. All mice were housed in a specific pathogen-free environment at $21 \pm 1\,°C$ and $60 \pm 5\%$ humidity, with a 12 h light/12 h dark cycle. The luciferase-expressing Huh7 cells (Huh7-Luc cells) were injected subcutaneously into the female BABL/c athymic nude mice. After the stabilization of tumour traits (after 2–3 weeks), the tumour blocks were collected and cut into small pieces. The small-sized tumour pieces were transplanted into the liver of nude mice to build the orthotopic xenograft model. On day 18, the mice were randomly sorted into four groups ($n = 5$ for each group) and respectively received the treatment with saline, cisplatin, PNPs or NMPNs (2.5 mg/kg body weight) on days 18, 21 and 24. To evaluate the therapeutic efficacy, all the mice were injected with D-Fluorescein disodium salt and the tumour size was detected by using a small animal imaging system (VISQUE In Vivo Elite, Korea) on days 18, 25 and 31. In addition, the body weight of all mice was recorded during the entire process. The mice were euthanized until the end of the experiment, and then the tumour and organs such as the heart, liver, spleen, lungs, and kidneys were harvested for pathological studies. The tumour tissues and organs were fixed in 10% formalin, embedded in paraffin, sectioned, and then stained by H&E or TUNEL.

### Biodistribution of NMPNs via ICP-MS analysis

Mice were intraperitoneally administered with NMPNs three times a week (2.5 mg/kg body weight) for two weeks. Then, the mice were sacrificed and the hearts, livers, spleens, lungs, kidneys, and tumours were harvested. The quantifiable amounts of hearts, livers, spleens, lungs, kidneys, and tumours were mixed with aqua regia and incubated at $60\,°C$ for 72 h. After dilution and filtration, the Pt concentrations were quantified by ICP-MS analysis.

### Hematoxylin and eosin (H&E) and TUNEL staining

Mice were treated with saline, cisplatin, PNPs or NMPNs (2.5 mg/kg body weight) three times a week, respectively. After 14 days, livers, spleens, lungs, kidneys, and tumour tissues were harvested, and fixed in 10% formalin, embedded in paraffin, sectioned, and then stained by H&E or TUNEL for further analysis.

### Evaluation of ROS level in tumour tissues

Mice were treated with saline, cisplatin, PNPs or NMPNs (2.5 mg/kg body weight) three times a week, respectively. After 14 days, the tumour tissues were collected and the ROS levels were analyzed through DCFH-DA staining by using CLSM.

### Immunofluorescence detection of γ-H2AX of tumour tissue

Mice were treated with saline, cisplatin, PNPs or NMPNs (2.5 mg/kg body weight) three times a week. After 14 days, tumour tissues were collected for further analysis of γ-H2AX expression by immunohistochemical staining.

### Statistical analysis

Data were presented as mean ± SEM ($n \geq 3$) and data analysis was performed using the Bruker WinEPR Acquisition Software (version 4.40), Microsoft Excel version 16.16.23, Image J (version 1.8.0) and GraphPad Prism Software Version 8.0 (GraphPad Prism, San Diego, California, USA). Statistical parameters including statistical analysis, statistical significance, and $n$ value were indicated in the figure legends. For statistical comparison, we performed two-tailed unpaired Student's $t$-test and one-way ANOVA to determine the significance. A value of $p < 0.05$ was considered significant (represented as *$p < 0.05$, **$p < 0.01$, ***$p < 0.001$ or not significant (n.s.)).

### Reporting summary

Further information on research design is available in the Nature Portfolio Reporting Summary linked to this article.

## Data availability

The public dataset used for this study is deposited in the RCSB PDB database under accession code 6WQG [Simmons, C.R., MacCulloch, T., Krepl, M. et al. The influence of Holliday junction sequence and dynamics on DNA crystal self-assembly. *Nat. Commun.* **13**, 3112 (2022).]. The remaining data are available in the Article, Supplementary Information and Source Data files. Source data are provided with this paper.

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

## Acknowledgements

We acknowledge financial support by the National Key Research and Development Program of China (2022YFB3203801 to D.L., 2022YFB3203804 to F.L., and 2022YFB3203800 to D.L.), Leading Talent of "Ten Thousand Plan"-National High-Level Talents Special Support Plan (to D.L.), National Natural Science Foundation of China (32071374 to F.L.), Program of Shanghai Academic Research Leader under the Science and Technology Innovation Action Plan (21XD1422100 to D.L.), Explorer Program of Science and Technology Commission of Shanghai Municipality (22TS1400700 to D.L.), One Belt and One Road International Cooperation Project from Key Research and Development Program of Zhejiang Province (2019C04024 to D.L.), Zhejiang Provincial Natural Science Foundation of China (LGF19C100002 to F.L.), Innovative Research Team of High-Level Local Universities in Shanghai (SHSMU-ZDCX20210900 to D.L.) and CAS Interdisciplinary Innovation Team (JCTD-2020-08). We thank Mr. Shengfei Yang for the contribution of cartoons in Fig. 7 (by using Figdraw).

## Author contributions

D.L. and F.L. conceived and designed the study. D.L., F.L., H.S., and X.H. developed the study. F.L., H.S., J.R., and X.H. synthesized and characterized the materials; J.R., H.S., and C.F. performed the cell and animal experiments; B.Z. performed the DFT calculation; D.L., F.L., H.S., J.R., B.Z., X.H., C.F., J.L., and H.G. analyzed the data and wrote the manuscript; D.L. and F.L. provided project supervision. All the authors discussed the results and approved the final version of the manuscript.

## Competing interests

The authors declare no competing interests.
