## [Peer Review File · Nature Communications]

Reviewers' Comments:

Reviewer #1:

Remarks to the Author:

This study reported tumour nucleus-targeted Pt nanozymes that possess a nuclease-mimetic activity to overcome cancer drug resistance. The authors demonstrated that, different from conventional Pt-based therapeutic strategies, the rationally designed Pt nanozymes can induce nanozyme-catalyzed cleavage of the formed Pt-DNA adducts and thus impair DNA bending structure, prerequisite for the NER process, which eventually overcome NER-associated tumour drug resistance. It is a clever and intriguing approach to exploit the structural vulnerability of Pt-DNA adducts to combat Pt-resistant tumors. The manuscript is well conceived and written, and the results are presented clearly with comprehensive discussion. The current work highlights the potential benefits of applying enzymatic activities to the use of Pt nanomedicines, which is important for the further development of advanced DNA damaging chemotherapy, and could have clinical benefit for cancer patients who have developed Pt drug resistance. Some minor comments:

1. The authors did a comprehensive work on clarifying that the NMPNs-generated ROS can inhibit the recruitment of NER-related proteins by disrupting the required DNA structure. However, to fully understand the roles of NMPNs in the suppression of cisplatin-resistant cells in vitro, a control group of the combination of NAC and NMPNs is needed.
2. NMPNs are expected to act at tumor area, it is necessary to provide the information about the concentration of NMPNs in tumor tissues. Also, did these NMPNs show unwanted toxicity to normal cells and tissues around tumors in the liver?
3. Considering that ROS can inhibit tumors insensitive to Pt by disrupting the NER pathway, it is necessary to assess the ROS level in tumor tissues after the treatment of NMPNs. More discussion is also needed.
4. The authors provided TEM images of PtNC@oleic acid and NMPNs, it is better to provide TEM images of PtNC@PEG and PtNC@TAT as well, and the corresponding DLS sizes and zeta potential.
5. The colloidal stability of NMPNs should be evaluated by analyzing their time-dependent size and surface charge.
6. In addition to the results regarding the endosomal escape and nucleus targeting of NMPNs, the description of the cellular uptake pathway of NMPNs are required.
7. Please unify the font size in the Figure 5.
8. Please specify the fluorescence labeling of NMPNs and PNPs in the figure legends.

Reviewer #2:

Remarks to the Author:

Platinum (Pt) resistance compromises the clinical usage of Platinum-based chemotherapeutics. In this manuscript, by exploiting the structural vulnerability of nucleotide excision repair (NER) process and the mechanism of Platinum (Pt) resistance, the authors developed a nuclease-mimetic Pt nanozyme (NMPNs) which can target to the cancer cell nucleus and induce concurrent DNA platination and oxidative cleavage to overcome Pt drug resistance. When NMPNs accumulated in tumor tissue, the acid-responsive layer on the surface was detached and the nucleus-targeting peptide TAT was exposed for the intracellular nucleus targeted delivery. The existence of Pt 0, Pt 2+, and Pt 4+ on the surface endowed NMPNs with potent oxidase and peroxidase-like activities to generate ROS for the cleavage of double-stranded DNA. Besides, NMPNs can efficiently suppress the recruitment of NER associated factors to inhibit NER process, thus leading to the excessive formation of Pt-DNA adducts and dramatically inducing the apoptosis of Pt-resistant tumor cells. Overall, this is a promising strategy to promote the clinical application of Platinum-based chemotherapeutics. The major conclusions are fully supported by the experimental data. I would suggest its publication on Nature communication after addressing the following comments.

1. The quality of Figure 2b is not high enough to distinguish lysosome and nanoparticle signals. The authors can supply the separated single fluorescence channels or improve the resolution.
2. The Figure 4a should also contain the PNPs group in the main text.
3. Please provide the quantitative analysis of Figure 4i.
4. Please describe how the experiment in Figure 3b was carried out in detail.
5. The stability of NMPNs in physiological condition should be evaluated.

6. There is another important aspect from the future translational aspect: how was the production capacity for making this new type of nanoparticle? Is this line of production capable for integration with the industrial scale production?

Reviewer #3:

Remarks to the Author:

Review of Li et al., "A nuclease-mimetic platinum nanoenzyme induces concurrent DNA platination and oxidative cleavage to overcome drug resistance"

In this manuscript the authors construct nuclease-mimetic Pt-nanoparticles designed to induce concurrent DNA platination and oxidative cleavage of the platinated lesion to prevent recognition and repair by nucleotide excision repair, with the intent of utilizing these particles for cancer therapy. The idea is clever and worth pursuing. However, many of the claims in the paper are based on correlation, key controls are missing, and the data is over-interpreted. The relative importance of NER in suppressing the cytotoxicity response to platinum treatments, compared to other repair pathways such as HR and the Fanconi pathway, remain subjects of active controversy. The authors do not definitively prove that the enhanced effects of their NMPNs are truly the result of NER disruption by the oxidative damage of the particle, and other parts of their data suggest that the oxidative effects may be more important for inducing other types of damage or modulating the platinum-induced damage in a manner that is independent of any effects on NER. Several of the Figure captions are so brief that it is difficult to know what exactly is being shown, and many parts of the work have not been quantified.

Major comments:

1- The cellular images shown in Figure 2B and C are so poorly illuminated that it is not possible to actually see what is claimed in the Results section about pH-dependent nuclear uptake, on page 8-9, lines 166-184, hence the reviewer is unable to determine if the claims are substantiated by the data.

2- Claims in Figure 2D and Supplemental Figure 12 that we are actually seeing Pt nanoparticles in the nucleus would be better if supported by some type of independent analysis directly showing the presence of platinum. Perhaps some type of EXAFS study or nuclear isolation and flame spectrometry could be done. Alternatively, the authors could use their anti-Pt:DNA adduct antibody as in Figure 4A. None of this data is quantified, which would also help make the findings more convincing.

3-Figure 3:

A- Please describe what is actually being measured in Figure 3A.

B-Why does the DNA fragmentation effect have such a sharp dose-dependence? There is essentially no DNA damage until the highest nanoparticle dose is reached, and these experiments required 12 hours of incubation. What are the kinetics of the DNA damage? Is there evidence that this relatively high concentration of nanoparticles is ever achieved in the nuclei of the treated cells?

C- Is the model of specific damage events shown panels C, D, and E simply a model postulated by the authors, or do they have actual experimental data demonstrating the specific chemical intermediates shown in the model are actually being formed?

D-The ultimate claim that these nanoparticles are causing ssDNA hydrolysis and a double strand break near the Pt site needs to be definitively demonstrated experimentally.

E- When is this double strand break actually being formed? Isn't it more likely to form during S-phase, at which time both HR and NHEJ are likely to outcompete NER as repair mechanisms?

4-Figure 4:

A- In panel B, why is the DCF fluorescence so localized to one pole of the nucleus?

B- In panel C, in their direct testing of the mechanism that oxidation is important following DNA platination, the authors show that NAC co-treatment reduces gH2AX formation, but the key controls showing equivalent NMPNs uptake in the presence of NAC are missing, and the presence of equivalent amounts of the Pt-DNA adduct are also missing. This Pt-DNA missing data, which is shown in panel F suggests that the NAC treatment has major effects on DNA platination itself, not just oxidative cleavage, and makes it impossible to use the results to validate the direct importance of the ROS-dependent cleavage effects. All of the data needs to be quantified.

C- What effect does NAC have on the PNP effects?

D- In panels D and E, the authors show primary data for XPA, which they do not quantify, and quantified data for XPF, which they do not show primary data for. Please show IF images for all of the relevant XP proteins and quantify all of the data for both NMPNs and PNPs.

E- In panel I, how reproducible is the effect? How many times was the experiment done? Are the differences statistically significant?

F- In panel J, how exactly is viability being measured? What is CCK-8 solution?

5- This is a critical point - what direct proof do the authors have that lack of NER is the primary mechanism responsible for the enhanced cytotoxicity? To make this claim, the authors need to compare the effects of the NMPNs on isogenic cells that are NER-proficient versus NER-deficient, such as XP knock-outs, and show that the cytotoxicity, compared to PNPs is only enhanced in NER-proficient cells.

6- Figure 5:

A- Images in Panel e are so dim that they are impossible to see, so the authors claims cannot be verified.

B- In panel f, please show gH2AX immunofluorescence, not just a Western blot

7- The models in Figure 4K and 5J are not adequately supported by the data, as I mention in comment #5.

In summary, the authors clearly appear to have created a novel nanoparticle with potential utility, but the mechanism that underlies the cellular effects is not adequately validated. I think the results are important even if the NER-focused mechanism is wrong, but additional experimental data is required.

Responses to REVIEWER COMMENTS

Reviewer #1 - nanoparticles and cancer (Remarks to the Author):

This study reported tumour nucleus-targeted Pt nanozymes that possess a nuclease-mimetic activity to overcome cancer drug resistance. The authors demonstrated that, different from conventional Pt-based therapeutic strategies, the rationally designed Pt nanozymes can induce nanozyme-catalyzed cleavage of the formed Pt-DNA adducts and thus impair DNA bending structure, prerequisite for the NER process, which eventually overcome NER-associated tumour drug resistance. It is a clever and intriguing approach to exploit the structural vulnerability of Pt-DNA adducts to combat Pt-resistant tumors. The manuscript is well conceived and written, and the results are presented clearly with comprehensive discussion. The current work highlights the potential benefits of applying enzymatic activities to the use of Pt nanomedicines, which is important for the further development of advanced DNA damaging chemotherapy, and could have clinical benefit for cancer patients who have developed Pt drug resistance. Some minor comments:

Response: *Thank you very much for your encouraging comments. We have made point-to-point responses to your suggestions and revised the manuscript accordingly to incorporate additional data. The revised sentences are marked with yellow background. We believe that your comments have significantly improved the quality of our manuscript.*

1. The authors did a comprehensive work on clarifying that the NMPNs-generated ROS can inhibit the recruitment of NER-related proteins by disrupting the required DNA structure. However, to fully understand the roles of NMPNs in the suppression of cisplatin-resistant cells in vitro, a control group of the combination of NAC and NMPNs is needed.

Response: *Thank you for your valuable comment. According to your suggestions, we analyzed the viability of cisplatin-resistant Huh7 cells after treatment with saline, NMPNs, NAC, or NMPNs+NAC. The results show that NMPNs treatment can dramatically suppress the growth of cisplatin-resistant Huh7 cells. However, the inhibitory effect of NMPNs on cisplatin-resistant Huh7 cell growth can be significantly compromised in the presence of a ROS scavenger NAC. Consistently, the cotreatment of NMPNs and NAC can suppress DNA double stranded breaks and decrease the accumulation of Pt-DNA adducts (Fig. 3c,d). These results demonstrate that ROS generated by NMPNs for DNA structure disruption is indispensable to the inhibitory effects of NMPNs on cisplatin-resistant cells.*

Our modification to the manuscript: *The results were added as Supplementary Fig.*

23 in the revised supporting information. In addition, the following sentences and methods were added on pages 12 and 30-31 in the revised manuscript, respectively.

- Figure S23

Supplementary Figure 23. The inhibition effect of saline, NMPNs, NAC, or NMPNs+NAC on cisplatin-resistant Huh7 cells growth. n = 5 independent experiments, data are presented as means \pm S.E.M. Source data are provided as a Source Data file.

- Page 12

“Compared with cisplatin, NMPNs cause more severe DNA damage (Fig. 5d), effectively inducing apoptosis to inhibit the proliferation of cisplatin-resistant Huh7 cells (Fig. 5e-g). However, the inhibitory effect of NMPNs on cisplatin-resistant Huh7 cell growth is significantly compromised in the presence of NAC (Supplementary Fig. 23).”

- Page 30-31

Method

In vitro cytotoxicity. Cisplatin-resistant Huh7 cells were seeded into 96-well plates (1×10^4 cells/well) for 24 h and treated with cisplatin, PNP or NMPNs with a concentration of 20 μ g/mL for 6 h at pH 6.5. Then, the incubation medium containing cisplatin, PNP or NMPNs were removed and replaced with fresh medium. After 12, 18 and 24 h, the cells were incubated with CCK-8 solution for each well and incubated for another 3 h, the absorbance of each well was measured at 450 nm by a microplate reader.

Cisplatin-resistant Huh7 cells were seeded into 96-well plates (1×10^4 cells/well) for 24 h and treated with NMPNs (20 μ g/mL) in the presence or absence of NAC for 6 h at pH 6.5. Then, the cells were incubated with CCK-8 solution for each well and

incubated for another 3 h, the absorbance of each well was measured at 450 nm by a microplate reader.

L02 cells were seeded into 96-well plates (1×10^4 cells/well) for 24 h and treated with NMPNs (20 $\mu\text{g}/\text{mL}$) for 6 h at pH 7.4. Then, the cells were incubated with CCK-8 solution for each well and incubated for another 3 h, the absorbance of each well was measured at 450 nm by a microplate reader.

2. NMPNs are expected to act at tumor area, it is necessary to provide the information about the concentration of NMPNs in tumor tissues. Also, did these NMPNs show unwanted toxicity to normal cells and tissues around tumors in the liver?

Response: Thank you for your kind suggestion. Accordingly, the biodistribution of NMPNs was studied by using inductively coupled plasma mass spectrometry (ICP-MS). The results show that NMPNs can effectively accumulate in tumour tissue. In addition, the effect of NMPNs has been evaluated on normal liver cell (e.g., L02 cell lines). The results demonstrated that NMPNs exert no inhibitory effect on the L02 cells growth under neutral conditions. Consistently, immunohistochemical analysis and TUNEL staining images demonstrated that NMPNs showed no obvious toxicity to normal liver tissues around tumours. These results indicate that NMPNs can effectively accumulate in tumour tissues and suppress tumour growth, and exert no evident toxicity to normal cells and tissues around tumours in the liver, which is likely due to cancer cell nucleus targeting capability of NMPNs as demonstrated in the manuscript.

Our modification to the manuscript: The results were added as Supplementary Figs. 25, 26, 29, 30 in the revised supporting information. In addition, the following sentences and methods were added on pages 12-13, 30-31, and 35-36 in the revised manuscript, respectively.

- Figure S25

Supplementary Figure 25. Cell viability of NMPNs on L02 cell lines in the neutral condition. $n = 4$ independent experiments, data are presented as means \pm S.E.M. Source

data are provided as a Source Data file.

- Figure S26

Supplementary Figure 26. Biodistribution of NMPNs in mice treated with NMPNs via ICP-MS analysis. n = 4 independent experiments, data are presented as means ± S.E.M. Source data are provided as a Source Data file.

- Figure S29

Supplementary Figure 29. Representative hematoxylin and eosin (H&E) staining images of normal tissues around tumour in the liver from saline or NMPNs-treated mice. n = 3 independent mouse livers. Scale bar: 500 µm.

- Figure S30

Supplementary Figure 30. Representative TUNEL staining images of normal tissues around tumour in the liver from saline or NMPNs-treated mice. n = 3 independent mouse livers. Scale bar: 100 μ m.

- Page 12

“Moreover, NMPNs show no obvious inhibitory effect on the normal cell growth, such as L02 cell, under neutral conditions (Supplementary Fig. 25).”

- Page 13

“NMPNs can efficiently accumulate in tumour tissues, likely due to their capability to expose TAT peptides under acidic tumour microenvironment (Supplementary Fig. 26)”. “Moreover, NMPNs show no obvious toxicity to normal tissues around the tumour in the liver and other major organs, suggesting the excellent biosafety of NMPNs (Fig. 6h and Supplementary Fig. 29-31).”

- Page 30-31

Method

In vitro cytotoxicity. Cisplatin-resistant Huh7 cells were seeded into 96-well plates (1×10^4 cells/well) for 24 h and treated with cisplatin, PNPs or NMPNs with a concentration of 20 μ g/mL for 6 h at pH 6.5. Then, the incubation medium containing cisplatin, PNPs or NMPNs were removed and replaced with fresh medium. After 12, 18 and 24 h, the cells were incubated with CCK-8 solution for each well and incubated for another 3 h, the absorbance of each well was measured at 450 nm by a microplate reader.

Cisplatin-resistant Huh7 cells were seeded into 96-well plates (1×10^4 cells/well) for 24 h and treated with NMPNs (20 $\mu\text{g}/\text{mL}$) in the presence or absence of NAC for 6 h at pH 6.5. Then, the cells were incubated with CCK-8 solution for each well and incubated for another 3 h, the absorbance of each well was measured at 450 nm by a microplate reader.

L02 cells were seeded into 96-well plates (1×10^4 cells/well) for 24 h and treated with NMPNs (20 $\mu\text{g}/\text{mL}$) for 6 h at pH 7.4. Then, the cells were incubated with CCK-8 solution for each well and incubated for another 3 h, the absorbance of each well was measured at 450 nm by a microplate reader.

- Page 35-36

Biodistribution of NMPNs via ICP-MS analysis. Mice were intraperitoneally administered with NMPNs three times a week (2.5 mg/kg body weight) for two weeks. Then, the mice were sacrificed and the hearts, livers, spleens, lungs, kidneys, and tumours were harvested. The quantifiable amounts of hearts, livers, spleens, lungs, kidneys, and tumours were mixed with aqua regia and incubated at 60°C for 72 h. After dilution and filtration, the Pt concentrations were quantified by ICP-MS analysis.

- Page 36

Hematoxylin and eosin (H&E) and TUNEL staining. Mice were treated with saline, cisplatin, PNPs or NMPNs (2.5 mg/kg body weight) three times a week, respectively. After 14 days, livers, spleens, lungs, kidneys, and tumour tissues were harvested, and fixed in 10% formalin, embedded in paraffin, sectioned, and then stained by H&E or TUNEL for further analysis.

3. Considering that ROS can inhibit tumors insensitive to Pt by disrupting the NER pathway, it is necessary to assess the ROS level in tumor tissues after the treatment of NMPNs. More discussion is also needed.

Response: *Thank you for your valuable comments. Per your suggestions, we have assessed the ROS level in tumour tissues after different treatments by using DCFH-DA probe. The results show that NMPNs can remarkably facilitate the ROS accumulation in tumour sites, which can impair the damage recognition factors-induced DNA bending prerequisite for NER.*

Our modification to the manuscript: *The results were added as Supplementary Fig. 28 in the revised supporting information. In addition, the following sentences and methods were added on pages 13 and 36 in the revised manuscript, respectively.*

- Figure S28

Supplementary Figure 28. The levels of ROS in tumour tissues after treatment with cisplatin, PNP or NMPN. n = 3 independent mice. Scale bar: 200 μ m.

- Page 13

“Considering that NMPNs-mediated ROS generation plays a crucial role in the destruction of the very structure of DNA required for NER, we further evaluated the ROS level in tumour tissues after the treatments with PNP, cisplatin or NMPN. Among which, only NMPNs can significantly enhance the ROS level in tumour tissues (Supplementary Fig. 28).”

- Page 36

Method

Evaluation of ROS level in tumour tissues. Mice were treated with saline, cisplatin, PNP or NMPN (2.5 mg/kg body weight) three times a week, respectively. After 14 days, the tumour tissues were collected and the ROS levels were analyzed through DCFH-DA staining by using CLSM.

4. The authors provided TEM images of PtNC@oleic acid and NMPN, it is better to provide TEM images of PtNC@PEG and PtNC@TAT as well, and the corresponding DLS sizes and zeta potential.

Response: Thank you for your valuable comment. According to your suggestion, the TEM images of PtNC@PEG and PtNC@TAT have been provided. In addition, the DLS sizes and zeta potentials of PtNC@TAT have been added in the revised manuscript. These results show that PtNC@PEG are well-dispersed in water, exhibiting negative charge (-24 mV) with a hydrodynamic diameter of ~11 nm as confirmed by DLS measurement. The DLS sizes and zeta potentials of PtNC@TAT had been previously

provided in the Supplementary Fig. 3 and 4 of the manuscript.

Our modification to the manuscript: The TEM results were added as Supplementary Fig. 3, and the DLS size and zeta potential of PtNC@PEG has been added into previous Supplementary Figs. 5 and 6 in the revised supporting information. In addition, the following sentences were added on page 6 in the revised manuscript.

- Figure S3

Supplementary Figure 3. a, b, TEM images of PtNC@PEG (a) and PtNC@TAT (b).

- Figure S5

Supplementary Figure 5. The hydrodynamic diameters of PtNC@PEG (pH 7.4), PtNC@TAT (pH 7.4), NMPNs (pH 7.4) and NMPNs (pH 6.5). n = 3 independent experiments, data are presented as mean values \pm S.E.M. Source data are provided as a Source Data file.

- Figure S6

Supplementary Figure 6. Zeta potentials of PtNC@PEG (pH 7.4), PtNC@TAT (pH 7.4), NMPNs (pH 7.4) and NMPNs (pH 6.5). n = 3 independent experiments, data are presented as mean values \pm S.E.M. Source data are provided as a Source Data file.

- Page 6

“PtNCs were further grafted with the heterobifunctional SH-PEG_{2K}-COOH to provide the carboxyl groups for 1-ethyl-3-(3-dimethylaminopropyl)-carbodiimide (EDC)-mediated coupling with the amino groups of the TAT peptides and then shielded with pH-responsive mPEG_{5K}-AC-CA (Supplementary Figs. 2, 3).”

“The successful modification of TAT peptide is verified by Fourier transform infrared (FT-IR) spectra (Supplementary Fig. 3), corresponding with the increased zeta potential and size relative to PtNC@PEG (Supplementary Fig. 5, 6).”

5. The colloidal stability of NMPNs should be evaluated by analyzing their time-dependent size and surface charge.

Response: Thank you very much for your valuable comments. Per your suggestion, the time-dependent sizes and zeta potentials of NMPNs were detected using a Zetasizer Nano ZS90 to confirm their colloidal stability. No obvious size or zeta potential changes were observed over a week, which demonstrates the high colloidal stability of NMPNs.

Our modification to the manuscript: The results were added as Supplementary Fig. 7 in the revised supporting information. In addition, the following sentences were added on page 6 in the revised manuscript.

- Figure S7

Supplementary Figure 7. a, b, The sizes (a) and zeta potentials (b) of NMPNs over a week. n = 3 independent experiments, data are presented as means \pm S.E.M. Source data are provided as a Source Data file.

- Page 6

“NMPNs are well-dispersed in water (Fig. 1c) with the high colloidal stability (Supplementary Fig. 7)”

6. In addition to the results regarding the endosomal escape and nucleus targeting of NMPNs, the description of the cellular uptake pathway of NMPNs are required.

Response: Thank you very much for your valuable comments. To investigate how NMPNs enter Huh7 cells, we quantified the cellular uptake of RITC-labelled NMPNs after adding methyl- β -cyclodextrin (M β CD, an inhibitor of caveolin-mediated endocytosis), chlorpromazine (an inhibitor of clathrin-mediated endocytosis) or amiloride (an inhibitor of macropinocytosis) <Ref. *Colloid. Surfaces B* 2018, 161, 10–17; *Nat. Commun.* 2016, 7, 11284>. In consistent with previously reported TAT peptide-modified nanoparticles <Ref. *Nat. Commun.* 2019, 10, 3646>, the cellular uptake reduced approximately 25 and 28% when Huh7 cells were pretreated with amiloride and chlorpromazine, respectively. Notably, a great reduction (about 45%) occurred when Huh7 cells were cultured with M β CD. These results demonstrate that NMPNs can be internalized into Huh7 cells by multiple pathways, among which the M β CD-mediated endocytosis played a major role in the cellular uptake of NMPNs.

Our modification to the manuscript: The results were added as Supplementary Fig. 15 in the revised supporting information. In addition, the following sentences and methods were added on pages 8-9 and 29 in the revised manuscript, respectively.

- Figure S15

Supplementary Figure 15. Flow cytometry analysis of internalization of NMPNs into Huh7 cells after pre-treatment with serum-free medium (vehicle), amiloride (an inhibitor of macropinocytosis), chlorpromazine (an inhibitor of clathrin-mediated endocytosis), or methyl-β-cyclodextrin (MβCD, an inhibitor of caveolin-mediated endocytosis). n = 3 independent experiments, data are presented as means ± S.E.M. Source data are provided as a Source Data file.

- Page 8-9

“Moreover, we further studied the cellular endocytosis mechanism of NMPNs in Huh7 cells. In accordance with previously reported TAT peptide-modified nanoparticles⁴⁷, the cellular uptake of NMPNs can be inhibited when Huh7 cells were pre-incubated with amiloride, chlorpromazine, and MβCD, respectively, indicating that NMPNs can be internalized into Huh7 cells through multiple pathways, among which the MβCD-mediated endocytosis plays a leading role in the cellular endocytosis of NMPNs (Supplementary Fig. 15).”

Reference:

47. Wei, Y., Tang, T. & Pang, H. Cellular internalization of bystander nanomaterial induced by TAT-nanoparticles and regulated by extracellular cysteine. *Nat. Commun.* **10**, 3646 (2019).

- Page 29

Method

Endocytic pathway of NMPNs. To study the cellular uptake mechanisms of NMPNs by cisplatin-resistant Huh7 cells, three specific endocytic inhibitors were used: (1) chlorpromazine, an inhibitor to probe clathrin-mediated endocytosis; (2) amiloride, an

inhibitor of micropinocytosis; (3) methyl- β -cyclodextrin (M β CD), an inhibitor of caveolin-mediated endocytosis. Specifically, cisplatin-resistant Huh7 cells were preincubated in serum-free RPMI DMEM medium with chlorpromazine (40 μ M, 30 min), amiloride (100 μ M, 30 min) or M β CD (5 mM, 30 min), respectively. The medium was then changed to fresh serum-free medium containing the inhibitors plus NMPNs (20 μ g/mL) and further incubated for 6 h at 37°C. The cells were collected for quantitative analysis.

7. Please unify the font size in the Figure 5.

Response: Thank you for your kind suggestion. We have unified the font size in the Fig. 5 (Now is Fig. 6 in the revised manuscript).

Our modification to the manuscript: We have unified the font size in Fig. 5 (Now is Fig. 6) in the revised manuscript.

Fig. 6 NMPNs suppress tumour in vivo and improve overall therapeutic outcome.

a, Schematic illustration of the establishment of orthotopic liver tumour mice model

and treatment process. **b**, Representative bioluminescence (BLI) images of each group after different treatments. **c**, Quantitative BLI signals of tumours after different treatments. n = 5 biologically independent animals. **d**, Representative images of liver collected from each treatment group on day 31. Black arrows indicate the tumour tissues. **e**, Representative TUNEL (green fluorescence) and DAPI (blue fluorescence) staining images of tumour tissues. Scale bar: 200 μm . **f,g**, Western blot analysis and quantitative analysis of $\gamma\text{-H2AX}$, XPA and XPF in the nucleus of tumour cells after different treatments. n = 3 independent experiments. **h**, The body weight changes of mice with different treatments. n = 5 biologically independent animals. **i**, The survival curves of orthotopic liver tumour mice after different treatments. n = 9 biologically independent animals. **j**, The schematic illustration of NMPNs to induce Pt-DNA adduct formation and oxidative cleavage for inhibiting the recruitment of XPA and XPF, which nullifies NER pathway and thus suppresses the tumour growth in vivo. All the data are presented as means \pm S.E.M. Statistical significance was analyzed by one-way ANOVA with multiple comparisons test (**g,h**) or two-tailed multiple *t*-tests with Bonferroni-Dunn correction (**c**). Source data are provided as a Source Data file.

8. Please specify the fluorescence labeling of NMPNs and PNP in the figure legends.

Response: *Thank you for your kind suggestion. We have specified the FITC labeling of NMPNs and PNP in the figure legends in the revised manuscript.*

Our modification to the manuscript: *We have specified the FITC labeling of NMPNs and PNP in the figure legends.*

Fig. 2 pH-dependent cell nucleus-targeting of NMPNs. **a**, CLSM images of intracellular localization of FITC-labelled NMPNs and FITC-labelled PNPs after incubated for 6 h under different pH conditions. Scale bar: 20 μm . Arrows indicate NMPNs in the cytoplasm after the escape from endosomes in cisplatin-resistant Huh7 cells. **b**, CLSM images of intracellular localization of FITC-labelled NMPNs and FITC-labelled PNPs after incubated for 12 h under different pH conditions. Scale bar: 40 μm . Asterisks indicate NMPNs in the nucleus of cisplatin-resistant Huh7 cells. **c**, Quantitative analysis of the colocalization between lysotracker and FITC-labelled NMPNs or PNPs. **d**, Quantitative analysis of the colocalization between DAPI and FITC-labelled NMPNs or PNPs. **e**, Bio-TEM images of cells after incubation for 12 h with NMPNs under different pH conditions. Arrowheads indicate NMPNs accumulated in the nucleus of Huh7 cells. Black asterisks indicate NMPNs in nucleus. Scale bar: 1 μm . **f**, Schematic diagram of the NMPNs to target the tumour cell nucleus. All the data are presented as means \pm S.E.M., $n = 3$ independent experiments. Source data are provided as a Source Data file.

Reviewer #2 - nanoparticles and cancer (Remarks to the Author):

Platinum (Pt) resistance compromises the clinical usage of Platinum-based chemotherapeutics. In this manuscript, by exploiting the structural vulnerability of nucleotide excision repair (NER) process and the mechanism of Platinum (Pt) resistance, the authors developed a nuclease-mimetic Pt nanozyme (NMPNs) which can target to the cancer cell nucleus and induce concurrent DNA platination and oxidative cleavage to overcome Pt drug resistance. When NMPNs accumulated in tumor tissue, the acid-responsive layer on the surface was detached and the nucleus-targeting peptide TAT was exposed for the intracellular nucleus targeted delivery. The existence of Pt⁰, Pt²⁺, and Pt⁴⁺ on the surface endowed NMPNs with potent oxidase and peroxidase-like activities to generate ROS for the cleavage of double-stranded DNA. Besides, NMPNs can efficiently suppress the recruitment of NER associated factors to inhibit NER process, thus leading to the excessive formation of Pt-DNA adducts and dramatically inducing the apoptosis of Pt-resistant tumor cells. Overall, this is a promising strategy to promote the clinical application of Platinum-based chemotherapeutics. The major conclusions are fully supported by the experimental data. I would suggest its publication on Nature communication after addressing the following comments.

Response: *Thank you very much for your encouraging comments. Based on your kind suggestions, we have made point-to-point responses and modified the manuscript.*

1. The quality of Figure 2b is not high enough to distinguish lysosome and nanoparticle signals. The authors can supply the separated single fluorescence channels or improve the resolution.

Response: *Thank you very much for your suggestion. Per your suggestion, we have provided the CLSM images with high quality and corresponding quantitative result in the revised manuscript.*

Our modification to the manuscript: *We have added the CLSM images with higher quality and the corresponding quantitative result as Fig. 2a and c in the revised manuscript, respectively. In addition, the following sentences were added on page 8 in the revised manuscript.*

• Figure 2

Fig. 2 pH-dependent cell nucleus-targeting of NMPNs. **a**, CLSM images of intracellular localization of FITC-labelled NMPNs and FITC-labelled PNPs after incubated for 6 h under different pH conditions. Scale bar: 20 μ m. Arrows indicate NMPNs in the cytoplasm after the escape from endosomes in cisplatin-resistant Huh7 cells. **b**, CLSM images of intracellular localization of FITC-labelled NMPNs and FITC-labelled PNPs after incubated for 12 h under different pH conditions. Scale bar: 40 μ m. Asterisks indicate NMPNs in the nucleus of cisplatin-resistant Huh7 cells. **c**, Quantitative analysis of the colocalization between lysotracker and FITC-labelled NMPNs or PNPs. **d**, Quantitative analysis of the colocalization between DAPI and FITC-labelled NMPNs or PNPs. **e**, Bio-TEM images of cells after incubation for 12 h with NMPNs under different pH conditions. Arrowheads indicate NMPNs accumulated in the nucleus of Huh7 cells. Black asterisks indicate NMPNs in nucleus. Scale bar: 1 μ m. **f**, Schematic diagram of the NMPNs to target the tumour cell nucleus. All the data are presented as means \pm S.E.M., n = 3 independent experiments. Source data are provided as a Source Data file.

• Page 8

“Under acidic conditions, the FITC signals diffuse across the cytoplasm of the cells, indicating the efficient cellular uptake and endosomal escape of NMPNs. In stark contrast, FITC signals of NMPNs are mostly confined in endosomes in a neutral environment. Moreover, FITC labelled PNPs are trapped in endosomes both in acidic and neutral environments (Fig. 2a,c).”

2. The Figure 4a should also contain the PNPs group in the main text.

Response: Thank you very much for your kind suggestion. Actually, we have provided the capacity of PNPs to induce Pt-DNA adducts in Fig. 5a. Compared with NMPNs treatment, PNPs cannot significantly enhance the production of Pt-DNA adducts in the nucleus of cisplatin-resistant cells, likely result from the lack of nucleus-targeting capability.

• Figure 5

Fig. 5 NMPNs promote cisplatin-resistant tumour cell apoptosis by inducing Pt-

DNA adducts without NER repairing. **a**, Immunofluorescence of the Pt-DNA adducts in cisplatin-resistant Huh7 cells after different treatments at pH 6.5. Scale bar: 40 μ m. **b**, Immunofluorescence of the Pt-DNA adducts in cisplatin-resistant Huh7 cells after treatments with NMPNs or NMPNs+NAC at different time points. Scale bar: 40 μ m. **c**, Quantitative analysis of Pt-DNA adducts in cisplatin-resistant Huh7 cells after treatments with NMPNs or NMPNs+NAC at different time points. **d**, Quantitative analysis of DNA damage of cisplatin-resistant Huh7 cells after treatment with cisplatin or NMPNs. Cisplatin: n = 46; NMPNs: n = 44. **e,f**, Flow cytometry analysis of cell apoptosis after different treatments at pH 6.5 (**e**) and corresponding quantitative results (**f**). **g**, The inhibition effect of cisplatin and NMPNs on cisplatin-resistant Huh7 cells growth at different incubation times. **h**, The cell viabilities of cisplatin-resistant Huh7 cells or siXPA-transfected cisplatin-resistant Huh7 cells after treatment with NMPNs, PNs or cisplatin. All the data are presented as means \pm S.E.M., n = 3 independent experiments. Statistical significance was analyzed by one-way ANOVA with multiple comparisons test. Source data are provided as a Source Data file.

3. Please provide the quantitative analysis of Figure 4i.

Response: *Thank you for your valuable suggestion. Per your suggestion, we have added the quantitative results of flow cytometry analysis of cell apoptosis after different treatments at pH 6.5.*

Our modification to the manuscript: *The results were added as Fig. 5f in the revised manuscript. In addition, the following sentences were added on page 12 in the revised manuscript.*

• Figure 5

Fig. 5 NMPNs promote cisplatin-resistant tumour cell apoptosis by inducing Pt-DNA adducts without NER repairing. **a**, Immunofluorescence of the Pt-DNA adducts in cisplatin-resistant Huh7 cells after different treatments at pH 6.5. Scale bar: 40 μ m. **b**, Immunofluorescence of the Pt-DNA adducts in cisplatin-resistant Huh7 cells after treatments with NMPNs or NMPNs+NAC at different time points. Scale bar: 40 μ m. **c**, Quantitative analysis of Pt-DNA adducts in cisplatin-resistant Huh7 cells after treatments with NMPNs or NMPNs+NAC at different time points. **d**, Quantitative analysis of DNA damage of cisplatin-resistant Huh7 cells after treatment with cisplatin or NMPNs. Cisplatin: n = 46; NMPNs: n = 44. **e,f**, Flow cytometry analysis of cell apoptosis after different treatments at pH 6.5 (**e**) and corresponding quantitative results (**f**). **g**, The inhibition effect of cisplatin and NMPNs on cisplatin-resistant Huh7 cells growth at different incubation times. **h**, The cell viabilities of cisplatin-resistant Huh7 cells or siXPA-transfected cisplatin-resistant Huh7 cells after treatment with NMPNs, PNPs or cisplatin. All the data are presented as means \pm S.E.M., n = 3 independent

experiments. Statistical significance was analyzed by one-way ANOVA with multiple comparisons test. Source data are provided as a Source Data file.

- Page 12

“Compared with cisplatin, NMPNs cause more severe DNA damage (Fig. 5d), effectively inducing apoptosis to inhibit the proliferation of cisplatin-resistant Huh7 cells (Fig. 5e-g).”

4. Please describe how the experiment in Figure 3b was carried out in detail.

Response: *Thank you for your kind suggestion. Per your suggestion, we have added the detailed description of experimental method.*

Our modification to the manuscript: *The method of the experiment in Fig. 3b were added on page 27 in the revised manuscript.*

- Page 27

Method

Study of DNA cleavage after NMPNs treatment. NMPNs were treated with PBS (pH 6.5) for 6 h. Then, PBS (pH 6.5) was removed by ultrafiltration and the NMPNs were collected. 300 ng of plasmid DNA (pet-32a+) were incubated with different amounts of the obtained NMPNs (2.5, 5, 10, 20 g/L) in PBS (pH 7.4) at 25°C for 12 h. These samples were mixed with 0.20 volume of the desired 6× loading buffer and slowly added into the 1% agarose gel slots in an electrophoresis tank containing 1× TBE electrophoresis buffer respectively, which were then applied with a voltage of 90 V until the DNA samples migrate a sufficient distance through the gel. Lastly, the gel was stained for 30 min by using SYBR dyes at room temperature and an image of the gel was captured under UV transillumination.

5. The stability of NMPNs in physiological condition should be evaluated.

Response: *Thank you very much for your valuable comments. Per your suggestion, the size and zeta potential of NMPNs in PBS and DMEM were detected using a Zetasizer Nano ZS90. No obvious size or zeta potential changes was observed over a week, which demonstrates their high colloidal stability.*

Our modification to the manuscript: *The results were added as Supplementary Fig. 7 in the revised supporting information. In addition, the following sentences were added on page 6 in the revised manuscript.*

- Figure S7

Supplementary Figure 7. a, b, The sizes (a) and zeta potentials (b) of NMPNs over a week. n = 3 independent experiments, data are presented as means \pm S.E.M. Source data are provided as a Source Data file.

- Page 6

“NMPNs are well-dispersed in water (Fig. 1c) with high colloidal stability (Supplementary Fig. 7)”

6. There is another important aspect from the future translational aspect: how was the production capacity for making this new type of nanoparticle? Is this line of production capable for integration with the industrial scale production?

Response: Thank you very much for your valuable comments. Pt nanoparticles were synthesized via a modified heat-up method, which is widely used for ultra-large-scale syntheses of monodisperse nanocrystals <Ref. Nature materials, 2004, 3, 891>. In our study, the Pt nanoclusters can be synthesized at 100-mg scale in a single reaction at 170°C by using 100 ml solution containing 1 g of Platinum(II) acetylacetonate. The modification of pH responsible ligands and nucleus targeting peptide can be completed at room temperature. Considering the high yield rate and mild reaction conditions, we speculate that the expansion of reactor can readily realize the larger-scale synthesis of NMPNs.

Reviewer #3 - DNA repair, cancer and chemotherapy. (Remarks to the Author):

Review of Li et al., “A nuclease-mimetic platinum nanozyme induces concurrent DNA platination and oxidative cleavage to overcome drug resistance”

In this manuscript the authors construct nuclease-mimetic Pt-nanoparticles designed to induce concurrent DNA platination and oxidative cleavage of the platinated lesion to prevent recognition and repair by nucleotide excision repair, with the intent of utilizing these particles for cancer therapy. The idea is clever and worth pursuing. However, many of the claims in the paper are based on correlation, key controls are missing, and the data is over-interpreted. The relative importance of NER in suppressing the cytotoxicity response to platinum treatments, compared to other repair pathways such as HR and the Fanconi pathway, remain subjects of active controversy. The authors do not definitively prove that the enhanced effects of their NMPNs are truly the result of NER disruption by the oxidative damage of the particle, and other parts of their data suggest that the oxidative effects may be more important for inducing other types of damage or modulating the platinum-induced damage in a manner that is independent of any effects on NER. Several of the Figure captions are so brief that it is difficult to know what exactly is being shown, and many parts of the work have not been quantified.

Response: *Thank you very much for your encouraging comments. Based on your kind suggestions, we have made point-to-point responses and modified the manuscript. We believe that your comments have significantly improved the quality of our manuscript.*

Major comments:

1- The cellular images shown in Figure 2B and C are so poorly illuminated that it is not possible to actually see what is claimed in the Results section about pH-dependent nuclear uptake, on page 8-9, lines 166-184, hence the reviewer is unable to determine if the claims are substantiated by the data.

Response: *Thank you very much for your comments. Per your suggestion, we provide the CLSM images with high quality to clarify the endosomal escape and nucleus-targeting capacity of NMPNs in cisplatin-resistant Huh7 cells at different pH conditions. In addition, the corresponding quantitative analyses have been added. As shown in Fig. 2b, the FITC signals diffuse across the cytoplasm of the cisplatin-resistant Huh7 cells, demonstrating the efficient cellular uptake and endosomal escape of NMPNs. However, FITC signals of NMPNs are mostly confined in endosomes in a neutral environment. Moreover, FITC-labelled PNPs are trapped in endosomes both in acidic and neutral environments. These results demonstrate the capability of NMPNs to escape the endosomes. As shown in Fig. 2b, NMPNs can efficiently accumulate in*

the cell nucleus in an acidic microenvironment. In contrast, PNPs can barely reach the nucleus, regardless of the pH conditions. These results clearly demonstrate that NMPNs can target and accumulate in the nucleus of cisplatin-resistant Huh7 cells.

Our modification to the manuscript: We have provided CLSM images with high quality in Fig. 2a (previous Fig. 2b) and Fig. 2b (previous Fig. 2c) and the corresponding quantitative analyses (Fig. 2c,d). In addition, the following sentences were added on page 8 in the revised manuscript.

- Figure 2

Fig. 2 pH-dependent cell nucleus-targeting of NMPNs. **a**, CLSM images of intracellular localization of FITC-labelled NMPNs and FITC-labelled PNPs after incubated for 6 h under different pH conditions. Scale bar: 20 μ m. Arrows indicate NMPNs in the cytoplasm after the escape from endosomes in cisplatin-resistant Huh7 cells. **b**, CLSM images of intracellular localization of FITC-labelled NMPNs and FITC-labelled PNPs after incubated for 12 h under different pH conditions. Scale bar: 40 μ m. Asterisks indicate NMPNs in the nucleus of cisplatin-resistant Huh7 cells. **c**, Quantitative analysis of the colocalization between lysotracker and FITC-labelled NMPNs or PNPs. **d**, Quantitative analysis of the colocalization between DAPI and FITC-labelled NMPNs or PNPs. **e**, Bio-TEM images of cells after incubation for 12 h

with NMPNs under different pH conditions. Arrowheads indicate NMPNs accumulated in the nucleus of Huh7 cells. Black asterisks indicate NMPNs in nucleus. Scale bar: 1 μm . **f**, Schematic diagram of the NMPNs to target the tumour cell nucleus. All the data are presented as means \pm S.E.M., n = 3 independent experiments. Source data are provided as a Source Data file.

- Page 8

“Under acidic conditions, the FITC signals diffuse across the cytoplasm of the cells, indicating the efficient cellular uptake and endosomal escape of NMPNs. In stark contrast, FITC signals of NMPNs are mostly confined in endosomes in a neutral environment. Moreover, FITC-labelled PNPs are trapped in endosomes both in acidic and neutral environments (Fig. 2a,c). These results indicate that the acid-induced exposure of TAT peptides can facilitate the endosomal escape of NMPNs⁴⁵, contributing to the efficient nucleus targeting. Indeed, in an acidic microenvironment, NMPNs can efficiently accumulate in the cell nucleus (Fig. 2b,d), which is beneficial from the surface TAT peptides that can promote the entry into the nucleus via the importin α/β pathway⁴⁶. In contrast, PNPs can barely reach the nucleus, regardless of the pH conditions (Fig. 2b,d).”

2- Claims in Figure 2D and Supplemental Figure 12 that we are actually seeing Pt nanoparticles in the nucleus would be better if supported by some type of independent analysis directly showing the presence of platinum. Perhaps some type of EXAFS study or nuclear isolation and flame spectrometry could be done. Alternatively, the authors could use their anti-Pt:DNA adduct antibody as in Figure 4A. None of this data is quantified, which would also help make the findings more convincing.

Response: *Thanks for your valuable comment. Per your suggestion, after treatment with NMPNs or PNPs, we isolated nucleus of cisplatin-resistant Huh7 cells and quantified the concentration of Pt ions in the nucleus by using ICP-MS. Consistent with the results of CLSM images and Bio-TEM analysis, the concentration of Pt ions in the nucleus significantly enhanced after treatment with NMPNs under the acidic condition as compared to PNPs treatment group (Supplementary Fig. 14). Moreover, to fully study the capability of NMPNs to target the tumour cell nucleus, we further analyzed the amount of Pt-DNA adducts after treatment with NMPNs or PNPs. The result demonstrated the enhanced accumulation of Pt-DNA adducts in the nucleus after NMPNs treatment as compared to that of PNPs treatment under the acidic conditions (Supplementary Fig. 17 a,b). These results can clearly show the accumulation of NMPNs in the nucleus of Huh7 cells under acidic conditions is more than that under neutral conditions, demonstrating the capability of NMPNs to target the nucleus of cisplatin-resistant Huh7 cells.*

Our modification to the manuscript: *The results were added as Supplementary Fig. 14 and 17 in the revised supporting information. In addition, the following sentences and method were added on pages 8-9 and 30 in the revised manuscript, respectively.*

- Figure S14

Supplementary Figure 14. The concentrations of Pt ions in the nucleus of cisplatin-resistant Huh7 cells after treatment with NMPNS or PNPs under different pH conditions. n = 3 independent experiments, data are presented as means ± S.E.M. Source data are provided as a Source Data file.

- Figure S17

Supplementary Figure 17. a, Immunofluorescence images of the Pt-DNA adducts in cisplatin-resistant Huh7 cells after different treatments at pH 6.5. Scale bar: 40 µm. **b,** Quantification analysis of Pt-DNA adducts in cisplatin-resistant Huh7 cells after different treatments at pH 6.5. n = 3 independent experiments, data are presented as means ± S.E.M. Source data are provided as a Source Data file.

- Page 8

“The Bio-TEM images and inductively coupled plasma mass spectrometry (ICP-MS) analysis of Pt ions in the nucleus also show the accumulation of NMPNs in the nucleus of Huh7 cells under acidic conditions (Fig. 2d and Supplementary Fig. 13, 14).”

- Page 9

“These results demonstrate that, different from PNPs, NMPNs implement the pH-responsive discharge of the “protective shield” under acidic conditions to exposure TAT peptides for facilitating the endosomal escape and subsequent nucleus targeting (Fig. 2f and Supplementary Fig. 16), which can readily initiate the formation Pt-DNA adducts (Supplementary Fig. 17a,b).”

- Page 30

Method

The concentration of NMPNs in the nucleus of cisplatin-resistant Huh7 cells. Cisplatin-resistant Huh7 cells were treated with NMPNs or PNPs (20 µg/mL) for 24 h at pH 7.4 or pH 6.5, respectively. The cell nucleus was extracted and then mixed with aqua regia and incubated at 60°C for 72 h. After dilution and filtration, the Pt concentrations were quantified by ICP-MS analysis.

3-Figure 3:

A- Please describe what is actually being measured in Figure 3A.

Response: *Thanks for your comment. In Fig. 3a, we have analyzed the Pt ions release behavior of NMPNs after the discharge of mPEG_{5K}-AC-CA at different time points (0.5, 2, 4, 8 h) under the neutral condition (pH 7.4). The result shows that NMPNs can quickly and persistently release Pt ions. To clarify what is being measured, we have provided detailed information of the Pt ions release measurement experiment in the revised “Methods” section.*

Our modification to the manuscript: *The following methods were added on page 25 in the revised manuscript.*

- Page 25

Method

Measurement of the released Pt ions from NMPNs. The Pt ion release behavior of NMPNs after the discharge of mPEG_{5K}-AC-CA at different time points (0.5, 2, 4, 8 h) was analyzed by using ICP-MS. First, NMPNs were treated with PBS (pH 6.5) for 6 h. Then, PBS (pH 6.5) was removed by ultrafiltration and the NMPNs were collected and dispersed in PBS (pH 7.4). After which, 1 mL of PBS (pH 7.4) containing 500 µg of the obtained NMPNs was placed in a dialysis bag and dialyzed against 10 mL of PBS (pH 7.4) at 37°C. The amount of the released Pt ions was quantified by using ICP-MS at 0.5, 2, 4, 8 h, respectively.

B-Why does the DNA fragmentation effect have such a sharp dose-dependence? There is essentially no DNA damage until the highest nanoparticle dose is reached, and these experiments required 12 hours of incubation. What are the kinetics of the DNA damage? Is there evidence that this relatively high concentration of nanoparticles is ever achieved in the nuclei of the treated cells?

Response: *Thanks for your valuable comment. The capability of NMPNs to induce DNA fragmentation is highly associated with the ROS generated by NMPNs. NMPNs generate ROS in a concentration dependent manner. We chose three gradient concentrations of NMPNs in the previous experiment (0.075, 0.374, 0.75 g/L), which might not fully clarify the changes in ROS generation and thus the DNA fragmentation mediated by NMPNs. Based on your suggestions, more gradient concentrations (0.25, 0.5, 1.0, 2.0 g/L) of NMPNs have been added to study the effect of NMPNs on the induction of DNA fragmentation. Meanwhile, we have increased the loading quantity to avoid errors caused by insufficient samples. As shown in the agarose gel electrophoresis of DNA treated with different concentrations of NMPNs, the breakage of DNA can be observed after treatment of NMPNs with the concentration from 0.25 g/L to 2.0 g/L. The results demonstrate that NMPNs can induce DNA fragmentation in a dose-dependent manner.*

Additionally, per your suggestion, we study the kinetics of the DNA cleavage after NMPNs treatment <Ref. Nature Biotechnology, 2019, 37, 945>. The detailed method are described in the Method section. The percentage of cleaved DNA vs. time followed pseudo-first-order kinetic profiles and can be well fitted by a single exponential function. Moreover, the kinetic of DNA cleavage were studied using different concentrations of NMPNs (0.42, 0.84, 1.68, 3.36 μM) and constant DNA amount (300 ng). The pseudo Michaelis-Menten kinetic parameters (V_{max} and K_M) were calculated to be $3.8 \times 10^{-7} \text{ M}^{-1}\text{s}^{-1}$ and $6.5 \times 10^{-5} \text{ M}$, respectively (Supplementary Fig. 18).

Finally, we have also measured the concentrations of Pt ions in the nuclei of Huh7 cells after treatment with NMPNs by using ICP-MS. The result shows that the Pt ion in the nuclei reaches up to $\sim 1.1 \mu\text{g}/10^7$ cells ($\sim 0.22 \text{ g/L}$) (Supplementary Fig. 14), which is comparable to the concentration of NMPNs (0.25 g/L) that can induce DNA cleavage in cell-free experiment, demonstrating that NMPNs can efficiently target and accumulate in the nucleus of cisplatin-resistant Huh7 cells to induce Pt-DNA adducts and DNA cleavage.

Our modification to the manuscript: *The results were added as Fig. 3b and Supplementary Fig. 14 and 18 in the revised manuscript. In addition, the following sentences and method were added on pages 8-9, 27-28, and 30 in the revised manuscript, respectively.*

- Figure S14

Supplementary Figure 14. The concentrations of Pt ions in the nucleus of cisplatin-resistant Huh7 cells after treatment with NMPNs or PNPs under different pH conditions. n = 3 independent experiments, data are presented as means ± S.E.M. Source data are provided as a Source Data file.

- Figure 3

Fig. 3 NMPNs induce Pt-DNA adducts formation and oxidative cleavage of DNA strand. **a**, The release of Pt ions from NMPNs at different time points under neutral conditions after the discharge of mPEG_{5K}-AC-CA. **b**, The agarose gel electrophoresis

of DNA treated with different concentrations of NMPNs (0.25, 0.5, 1.0, 2.0 g/L). **c**, Schematic illustration of Pt ion binding to DNA backbone. **d**, Schematic illustration of ROS coordinating with Pt ion and causing oxidative DNA cleavage. **e**, Comparison of hydrolysis and desorption process of complete DNA double-stranded fragments (upper) and DNA double-stranded fragments attacked by ROS (lower). The hydrolysis energy of DNA strand fragments attacked by ROS is -4.66 eV, lower than that of the complete double-stranded DNA (3.00 eV). Besides, the desorption energy of the unpaired base strand in ROS-attacked DNA is 3.05 eV, which is also lower than that of the bases with pairs in complete double-stranded DNA (9.22 eV). All the data are presented as means \pm S.E.M., $n = 3$ independent experiments. Source data are provided as a Source Data file.

- Figure S18

Supplementary Figure 18. Pseudo Michaelis-Menten kinetics of the cleavage of DNA (~300 ng) after treatment with NMPNs. The pseudo Michaelis-Menten kinetic parameters V_{max} and K_M were calculated to be $3.8 \times 10^{-7} \text{ M}^{-1}\text{s}^{-1}$ and $6.5 \times 10^{-5} \text{ M}$, respectively.

- Page 8

“The Bio-TEM images as well as inductively coupled plasma mass spectrometry (ICP-MS) analysis of Pt ions in the nucleus also show the accumulation of NMPNs in the nucleus of Huh7 cells under acidic conditions (Fig. 2e and Supplementary Fig. 13, 14).”

- Page 9

“Additionally, NMPNs can cause DNA fragmentation (Fig. 3b), probably resulting from the excessive NMPNs-catalyzed ROS generation.³⁴”

“Moreover, the NMPNs-induced DNA cleavage follows pseudo Michaelis-Menten kinetics (Supplementary Fig. 18).”

- Page 27

Method

Study of DNA cleavage after NMPNs treatment. NMPNs were treated with PBS (pH 6.5) for 6 h. Then, PBS (pH 6.5) was removed by ultrafiltration and the NMPNs were collected. 300 ng of plasmid DNA (pet-32a+) were incubated with different amounts of the obtained NMPNs (2.5, 5, 10, 20 g/L) in PBS (pH 7.4) at 25°C for 12 h. These samples were mixed with 0.20 volume of the desired 6× loading buffer and slowly added into the 1% agarose gel slots in an electrophoresis tank containing 1× TBE electrophoresis buffer respectively, which were then applied with a voltage of 90 V until the DNA samples migrate a sufficient distance through the gel. Lastly, the gel was stained for 30 min by using SYBR dyes at room temperature and an image of the gel was captured under UV transillumination.

Study of kinetic of NMPNs-induced DNA cleavage. 300 ng of Plasmid DNA were incubated with NMPNs with different concentrations (0.42, 0.84, 1.68, 3.36 μM) in PBS (pH 7.4) for 3, 9, 12 h, respectively. Then, the integrity of DNA was investigated by using gel electrophoresis. The band intensities were quantified using Image J software (version 1.8.0) to calculate the percentage of cleaved DNA. The percentage of cleaved DNA vs. time followed pseudo-first-order kinetic profiles. Then, GraphPad Prism Software Version 8.0 was used for the non-linear curve fitting.

- Page 30

The concentration of NMPNs in the nucleus of cisplatin-resistant Huh7 cells. Cisplatin-resistant Huh7 cells were treated with NMPNs or PNP (20 μg/mL) for 24 h at pH 7.4 and pH 6.5, respectively. Then the cells were mixed with aqua regia and incubated at 60°C for 72 h. After dilution and filtration, the Pt concentrations were quantified by ICP-MS analysis.

C- Is the model of specific damage events shown panels C, D, and E simply a model postulated by the authors, or do they have actual experimental data demonstrating the specific chemical intermediates shown in the model are actually being formed?

Response: *Thanks for your valuable comment. The configurations we constructed in panels C, D and E were based on the theoretical postulation in conjugation with the experimental evidence we collected. The models shown in Figs. 3c,d and e are postulated based on the properties of NMPNs (Figs. 1f,g and 3a) on their effects on DNA (Fig. 3b and Fig. 4a,c). It has been reported that Pt nanoparticles can release Pt ions (Fig. 3a), which can react with phosphate groups of DNA and form Pt-DNA adducts (Fig. 4a) < Ref. Arch Toxicol, 2011, 85, 799-812 >. Therefore, we first placed a Pt ion near the DNA backbone as the initial structure to describe the details of the interaction between Pt ions and DNA, and then obtained the Pt-H intermediate by DFT*

optimization (as shown in Fig 3c), which is similar to the intermediate products of cisplatin verified in the experiment <Ref. *J. Med. Chem.* 2007, 50, 2601-2604; *Phys. Chem. Chem. Phys.*, 2014, 16, 19290-19297; *Ref. Comput. Theor. Chem.*, 2016, 1094, 47-54>. The sugar radical generated by DNA binding to Pt ions is highly reactive and easily oxidized, causing the P-O bond breaks and DNA cleavages <Ref. *Chem. Rev.*, 2010, 110, 1018-1059>. Considering the capability of NMPNs to generate ROS (Fig. 1f,g), in Fig 3d, we took the Pt-DNA intermediate configuration with ROS added as the initial structure, designed for exploring the interaction process of ROS on Pt-DNA adducts, and got the configuration of DNA strand breaks by DFT calculation, which is consistent with previous reports. Besides, unrepaired DNA single-strand breaks have been reported to terminate gene transcription and create toxic DNA double-strand breaks during DNA replication <Ref. *Proc. Natl. Acad. Sci. USA*, 2001, 98, 8241-8246>. Therefore, in Fig 3e, representative DNA structural fragments were selected as templates to calculate hydrolysis energy, with the complete DNA compared with the DNA with SSBs. It is concluded that DNA with SSB that can be induced by NMPNs is easier to hydrolyze (Fig. 3b), which is consistent with the results of previous literature <Ref. *Proc. Natl. Acad. Sci. USA*, 2001, 98, 8241-8246>, and ultimately resulting in a double-strand break near the Pt site. Indeed, the confocal images of DNA fiber extracted from the nucleus of NMPNs-treated cisplatin-resistant Huh7 cells show that the Pt-DNA binding is mainly localized at the end of DNA fragmentation (Supplementary Fig. 19).

Our modification to the manuscript: The results were added as Fig. 3b and Supplementary Fig. 19 in the revised manuscript. In addition, the following sentences and method were added on pages 10 and 28 in the revised manuscript, respectively.

• Figure 3

Fig. 3 NMPNs induce Pt-DNA adducts formation and oxidative cleavage of DNA strand. **a**, The release of Pt ions from NMPNs at different time points under neutral conditions after the discharge of mPEG₅K-AC-CA. **b**, The agarose gel electrophoresis of DNA treated with different concentrations of NMPNs (0.25, 0.5, 1.0, 2.0 g/L). **c**, Schematic illustration of Pt ion binding to DNA backbone. **d**, Schematic illustration of ROS coordinating with Pt ion and causing oxidative DNA cleavage. **e**, Comparison of hydrolysis and desorption process of complete DNA double-stranded fragments (upper) and DNA double-stranded fragments attacked by ROS (lower). The hydrolysis energy of DNA strand fragments attacked by ROS is -4.66 eV, lower than that of the complete double-stranded DNA (3.00 eV). Besides, the desorption energy of the unpaired base strand in ROS-attacked DNA is 3.05 eV, which is also lower than that of the bases with pairs in complete double-stranded DNA (9.22 eV). All the data are presented as means ± S.E.M., n = 3 independent experiments. Source data are provided as a Source Data file.

- Figure S19

DAPI / Pt-DNA adducts

Supplementary Figure 19. The analysis of Pt-DNA binding sites in the DNA extracted from cisplatin-resistant Huh7 cells after treatment with NMPNs. The blue fluorescence and the green fluorescence correspond to DNA fibers and Pt-DNA binding sites, respectively. Arrows indicate the DNA fragmentations. Asterisks indicate the Pt-DNA binding sites at the end of DNA fragmentations. n = 3 independent experiments. Scale bar: 100 μm .

- Page 10

“In line with the DFT results, we found that NMPNs can induce the formation of Pt-DNA adducts and DNA breakage in the cisplatin-resistant Huh7 cells (Fig. 4a,c). Moreover, we extracted the DNA fragmentations from NMPNs-treated cisplatin-resistant Huh7 cells and analyzed the location of Pt-DNA binding in the DNA fragmentations. Intriguingly, the result shows that the Pt-DNA binding sites mainly localize at the end of DNA fragmentation (Supplementary Fig. 19).”

- Page 28

Method

The analysis of locations of Pt-DNA binding in the DNA. Cisplatin-resistant Huh7 cells were cultured in 6-well plates with 2 mL culture medium. NMPNs (20 $\mu\text{g}/\text{mL}$, pH 6.5) were added and incubated for 24 h, cells were subsequently collected and quickly resuspended in 500-600 μL of ice-cold PBS. Then, 2.5 μL of cell resuspension was spotted onto a pre-cleaned glass slide and mixed with 7.5 μL of spreading buffer (0.5% SDS in 200 mM Tris-HCl (pH 7.4), 50 mM EDTA). After 10 min, the slides were tilted to 15° to spread DNA fibers along the length. Then, air-dry the DNA spreads and fixed

in 3:1 methanol/acetic acid for 20 min at -20°C. After washing with PBS three times, slides were blocked with 1% BSA in PBS for 30 min at room temperature and incubated with anti-cisplatin modified DNA antibodies (GeneTex, GTX17412, dilution 1:100) to detect Pt-DNA adduct. After 1 h of incubation, slides were washed with PBS three times and stained with Alexan Fluor 488-conjugated AffiniPure Rabbit anti-Rat IgG (H+L) (Boster, BA1129, dilution 1:200) for 2 h at room temperature in the dark. After washing with PBS three times, slides were incubated with DAPI solution for 15 min in the dark. Finally, the cells were detected by using CLSM.

D-The ultimate claim that these nanoparticles are causing ssDNA hydrolysis and a double strand break near the Pt site needs to be definitively demonstrated experimentally.

Response: *Thanks for your valuable comment. As shown in the image of agarose gel electrophoresis, NMPNs can efficiently induced the hydrolysis of DNA (Fig. 3b). Notably, NMPNs induce the double strand breaks of DNA in the nucleus of cisplatin-resistant Huh7 cells as indicated by the increased level of γ -H2AX (Fig. 4c) <Ref. Nucleic Acids Res., 2008, 36, 5678>. Moreover, NMPNs can also significantly enhance the level of Pt-DNA adducts in the in the nucleus of cisplatin-resistant Huh7 cells, demonstrating the binding of Pt ions to DNA (Fig. 5a). In light of these results, we extracted DNA fragmentations from the nucleus of cisplatin-resistant Huh7 cells after NMPNs treatment, and analyzed the Pt-DNA binding sites by using confocal laser scanning microscopy. The result shows that the green signals corresponding to the Pt-DNA binding mainly localize at the end of DNA fibers (Supplementary Fig. 19). These results demonstrate that NMPNs can induce the double strand break of DNA near the Pt-DNA binding sites.*

Our modification to the manuscript: *The results were added as Supplementary Fig. 19 in the revised manuscript. In addition, the following sentences and method were added on pages 10 and 28 in the revised manuscript, respectively.*

• Figure 3

Fig. 3 NMPNs induce Pt-DNA adducts formation and oxidative cleavage of DNA strand. **a**, The release of Pt ions from NMPNs at different time points under neutral conditions after the discharge of mPEG_{5K}-AC-CA. **b**, The agarose gel electrophoresis of DNA treated with different concentrations of NMPNs (0.25, 0.5, 1.0, 2.0 g/L). **c**, Schematic illustration of Pt ion binding to DNA backbone. **d**, Schematic illustration of ROS coordinating with Pt ion and causing oxidative DNA cleavage. **e**, Comparison of hydrolysis and desorption process of complete DNA double-stranded fragments (upper) and DNA double-stranded fragments attacked by ROS (lower). The hydrolysis energy of DNA strand fragments attacked by ROS is -4.66 eV, lower than that of the complete double-stranded DNA (3.00 eV). Besides, the desorption energy of the unpaired base strand in ROS-attacked DNA is 3.05 eV, which is also lower than that of the bases with pairs in complete double-stranded DNA (9.22 eV). All the data are presented as means ± S.E.M., n = 3 independent experiments. Source data are provided as a Source Data file.

• Figure 4

Fig. 4 NMPNs circumvent NER pathway by concurrent of DNA platination and oxidative cleavage. **a**, The level of Pt-DNA adducts in the nucleus of cisplatin-resistant Huh7 cells after treatment with NMPNs at pH 6.5. Scale bar: 40 μ m. **b**, ROS levels in the nucleus of cisplatin-resistant Huh7 cells after treatment with PNP or NMPNs at pH 6.5. Scale bar: 40 μ m. **c**, Immunofluorescence of the γ -H2AX in cisplatin-resistant

Huh7 cells after different treatments at pH 6.5. Scale bar: 40 μm . **d**, Immunofluorescence of the XPA and Pt-DNA adducts in cisplatin-resistant Huh7 cells after different treatments at pH 6.5. The green fluorescence and the red fluorescence correspond to XPA and Pt-DNA adducts, respectively. Scale bar: 40 μm . **e**, Quantitative analysis of the colocalization between XPA and Pt-DNA adducts. **f**, Immunofluorescence of the XPF and Pt-DNA adducts in cisplatin-resistant Huh7 cells after different treatments at pH 6.5. The green fluorescence and the red fluorescence correspond to Pt-DNA adducts and XPF, respectively. Scale bar: 40 μm . **g**, Quantitative analysis of the colocalization between XPF and Pt-DNA adducts. **h**, Western blot analysis of XPF expression in the nucleus of cisplatin-resistant Huh7 cells after different treatments. **i**, Quantitative analysis of XPF expression in cisplatin-resistant Huh7 cells after different treatments at pH 6.5. **j**, The schematic illustration of the mechanism underlying NMPNs to induce DNA platination and oxidative cleavage to combat Pt resistance in tumour cells. In comparison to Pt compounds and PNPs that cannot effectively generate ROS in the nucleus, NMPNs can readily accumulate in the nucleus by acidity-induced exposure of TAT peptides, and induce in situ ROS generation to induce DNA oxidative cleavage, thus destroying the DNA conformation required for NER. It hampers the recruitment of XPA and XPF and thus inhibiting NER pathway. All the data are presented as means \pm S.E.M., $n = 3$ independent experiments. Statistical significance was analyzed by one-way ANOVA with multiple comparisons test. Source data are provided as a Source Data file.

• Figure 5

Fig. 5 NMPNs promote cisplatin-resistant tumour cell apoptosis by inducing Pt-DNA adducts without NER repairing. **a**, Immunofluorescence of the Pt-DNA adducts in cisplatin-resistant Huh7 cells after different treatments at pH 6.5. Scale bar: 40 μ m. **b**, Immunofluorescence of the Pt-DNA adducts in cisplatin-resistant Huh7 cells after treatments with NMPNs or NMPNs+NAC at different time points. Scale bar: 40 μ m. **c**, Quantitative analysis of Pt-DNA adducts in cisplatin-resistant Huh7 cells after treatments with NMPNs or NMPNs+NAC at different time points. **d**, Quantitative analysis of DNA damage of cisplatin-resistant Huh7 cells after treatment with cisplatin or NMPNs. Cisplatin: n = 46; NMPNs: n = 44. **e,f**, Flow cytometry analysis of cell apoptosis after different treatments at pH 6.5 (**e**) and corresponding quantitative results (**f**). **g**, The inhibition effect of cisplatin and NMPNs on cisplatin-resistant Huh7 cells growth at different incubation times. **h**, The cell viabilities of cisplatin-resistant Huh7 cells or siXPA-transfected cisplatin-resistant Huh7 cells after treatment with NMPNs, PNPs or cisplatin. All the data are presented as means \pm S.E.M., n = 3 independent

experiments. Statistical significance was analyzed by one-way ANOVA with multiple comparisons test. Source data are provided as a Source Data file.

- Figure S19

Supplementary Figure 19. The analysis of Pt-DNA binding sites in the DNA extracted from cisplatin-resistant Huh7 cells after treatment with NMPNs. The blue fluorescence and the green fluorescence correspond to DNA fibers and Pt-DNA binding sites, respectively. Arrows indicate the DNA fragmentations. Asterisks indicate the Pt-DNA binding sites at the end of DNA fragmentations. n = 3 independent experiments. Scale bar: 100 μm .

- Page 10

“In line with the DFT results, we found that NMPNs can induce the formation of Pt-DNA adducts and DNA breakage in the cisplatin-resistant Huh7 cells (Fig. 4a,c). Moreover, we extracted the DNA fragmentations from NMPNs-treated cisplatin-resistant Huh7 cells and analyzed the location of Pt-DNA binding in the DNA fragmentations. Intriguingly, the result shows that the Pt-DNA binding sites mainly localize at the end of DNA fragmentation (Supplementary Fig. 19).”

- Page 28

Method

The analysis of locations of Pt-DNA binding in the DNA. Cisplatin-resistant Huh7 cells were cultured in 6-well plates with 2 mL culture medium. NMPNs (20 $\mu\text{g}/\text{mL}$, pH 6.5) were added and incubated for 24 h, cells were subsequently collected and quickly resuspended in 500-600 μL of ice-cold PBS. Then, 2.5 μL of cell resuspension was spotted onto a pre-cleaned glass slide and mixed with 7.5 μL of spreading buffer (0.5%

SDS in 200 mM Tris-HCl (pH 7.4), 50 mM EDTA). After 10 min, the slides were tilted to 15° to spread DNA fibers along the length. Then, air-dry the DNA spreads and fixed in 3:1 methanol/acetic acid for 20 min at -20°C. After washing with PBS three times, slides were blocked with 1% BSA in PBS for 30 min at room temperature and incubated with anti-cisplatin modified DNA antibodies (GeneTex, GTX17412, dilution 1:100) to detect Pt-DNA adduct. After 1 h of incubation, slides were washed with PBS three times and stained with Alexan Fluor 488-conjugated AffiniPure Rabbit anti-Rat IgG (H+L) (Boster, BA1129, dilution 1:200) for 2 h at room temperature in the dark. After washing with PBS three times, slides were incubated with DAPI solution for 15 min in the dark. Finally, the cells were detected by using CLSM.

E- When is this double strand break actually being formed? Isn't it more likely to form during S-phase, at which time both HR and NHEJ are likely to outcompete NER as repair mechanisms?

Response: *Thank you for your valuable comments. As the reviewer mentioned, both HR and NHEJ are likely to outcompete NER as DSBs repair mechanisms during S-phase <Molecular Cell, 2012, 47, 320-329>. However, in this study, NMPNs can release Pt ions that interact with DNA to form Pt-DNA adducts, which is cell cycle phase nonspecific <Ref. Brit. J. Cancer, 2007, 96, 231-240; Nanomedicine, 2010, 5, 51-64>. Moreover, the accumulated NMPNs in the nucleus can efficiently generate ROS, which can induce the break of DNA strains around the Pt-DNA binding site, which can lead to the hydrolysis of the other DNA strain, ultimately resulting in the DSBs (Fig. 4b). Therefore, the DSB induced by NMPNs can be formed in any phase of cell cycles. Notably, the repair of DSBs in S phase mediated by HR and NHEJ can be hindered by the Pt ions that bind to the end of DNA via steric effect (Supplementary Fig. 19) <Ref. Mol. Cancer Res., 2005, 3, 277-285; DNA and Chromosomes, 2012, 287, 24263-24272>. Notably, NER is the dominant way to remove the Pt-DNA adducts <Ref. Nature, 2009, 461, 1071-1078; Sci. Rep., 2017, 7, 11785>. Consequently, the NMPNs induced the formation of Pt-DNA adducts and the subsequent oxidative cleavage of Pt-DNA adducts can impair the structure basis required for NER, thus inhibiting the DNA repair and suppressing the growth of cisplatin-resistant cancer cells.*

Our modification to the manuscript: *The results were added as Supplementary Fig. 19 in the revised manuscript. In addition, the following method were added on page 28 in the revised manuscript.*

• Figure 4

Fig. 4 NMPNs circumvent NER pathway by concurrent of DNA platination and oxidative cleavage. **a**, The level of Pt-DNA adducts in the nucleus of cisplatin-resistant Huh7 cells after treatment with NMPNs at pH 6.5. Scale bar: 40 μm. **b**, ROS levels in the nucleus of cisplatin-resistant Huh7 cells after treatment with PNPs or NMPNs at pH 6.5. Scale bar: 40 μm. **c**, Immunofluorescence of the γ-H2AX in cisplatin-resistant

Huh7 cells after different treatments at pH 6.5. Scale bar: 40 μ m. **d**, Immunofluorescence of the XPA and Pt-DNA adducts in cisplatin-resistant Huh7 cells after different treatments at pH 6.5. The green fluorescence and the red fluorescence correspond to XPA and Pt-DNA adducts, respectively. Scale bar: 40 μ m. **e**, Quantitative analysis of the colocalization between XPA and Pt-DNA adducts. **f**, Immunofluorescence of the XPF and Pt-DNA adducts in cisplatin-resistant Huh7 cells after different treatments at pH 6.5. The green fluorescence and the red fluorescence correspond to Pt-DNA adducts and XPF, respectively. Scale bar: 40 μ m. **g**, Quantitative analysis of the colocalization between XPF and Pt-DNA adducts. **h**, Western blot analysis of XPF expression in the nucleus of cisplatin-resistant Huh7 cells after different treatments. **i**, Quantitative analysis of XPF expression in cisplatin-resistant Huh7 cells after different treatments at pH 6.5. **j**, The schematic illustration of the mechanism underlying NMPNs to induce DNA platination and oxidative cleavage to combat Pt resistance in tumour cells. In comparison to Pt compounds and PNPs that cannot effectively generate ROS in the nucleus, NMPNs can readily accumulate in the nucleus by acidity-induced exposure of TAT peptides, and induce in situ ROS generation to induce DNA oxidative cleavage, thus destroying the DNA conformation required for NER. It hampers the recruitment of XPA and XPF and thus inhibiting NER pathway. All the data are presented as means \pm S.E.M., n = 3 independent experiments. Statistical significance was analyzed by one-way ANOVA with multiple comparisons test. Source data are provided as a Source Data file.

- Figure S19

Supplementary Figure 19. The analysis of Pt-DNA binding sites in the DNA extracted

from cisplatin-resistant Huh7 cells after treatment with NMPNs. The blue fluorescence and the green fluorescence correspond to DNA fibers and Pt-DNA binding sites, respectively. Arrows indicate the DNA fragmentations. Asterisks indicate the Pt-DNA binding sites at the end of DNA fragmentations. n = 3 independent experiments. Scale bar: 100 μm .

- Page 28

Method

The analysis of locations of Pt-DNA binding in the DNA. Cisplatin-resistant Huh7 cells were cultured in 6-well plates with 2 mL culture medium. NMPNs (20 $\mu\text{g}/\text{mL}$, pH 6.5) were added and incubated for 24 h, cells were subsequently collected and quickly resuspended in 500-600 μL of ice-cold PBS. Then, 2.5 μL of cell resuspension was spotted onto a pre-cleaned glass slide and mixed with 7.5 μL of spreading buffer (0.5% SDS in 200 mM Tris-HCl (pH 7.4), 50 mM EDTA). After 10 min, the slides were tilted to 15° to spread DNA fibers along the length. Then, air-dry the DNA spreads and fixed in 3:1 methanol/acetic acid for 20 min at -20°C. After washing with PBS three times, slides were blocked with 1% BSA in PBS for 30 min at room temperature and incubated with anti-cisplatin modified DNA antibodies (GeneTex, GTX17412, dilution 1:100) to detect Pt-DNA adduct. After 1 h of incubation, slides were washed with PBS three times and stained with Alexan Fluor 488-conjugated AffiniPure Rabbit anti-Rat IgG (H+L) (Boster, BA1129, dilution 1:200) for 2 h at room temperature in the dark. After washing with PBS three times, slides were incubated with DAPI solution for 15 min in the dark. Finally, the cells were detected by using CLSM.

4-Figure 4:

A- In panel B, why is the DCF fluorescence so localized to one pole of the nucleus?

Response: *Thank you for your comments. Previous studies reported that the lysosome location itself tends to be distributed on one side of the cell nucleus <Ref. Chem. Sci., 2020, 11, 596; Sensor. Actuat. B-Chem., 2020, 15, 128302; Sensor. Actuat. B: Chem., 2021, 15, 130397; Adv. Funct. Mater., 2020, 30, 1909999>. Once NMPNs escape from the lysosomes, they also tend to enter the nucleus on the side of the nucleus that is adjacent to the lysosome via the importin α/β pathway, and accumulate at one pole of the nucleus (Fig. 2b). Consequently, NMPNs generate ROS in situ and the higher level of ROS (as indicated by the DCF fluorescence signals) was observed at one pole of the nucleus (Fig. 4b).*

Our modification to the manuscript: *The following sentences were added on page 10 in the revised manuscript.*

- Page 10

“As mentioned above, NMPNs accumulated in the nucleus can effectively induce the formation of Pt-DNA adducts by releasing Pt ions (Fig. 4a), and facilitate ROS generation in the nucleus (Fig. 4b) for inducing intensive oxidative cleavage of DNA strands (Fig. 4c), mimicking a biomimetic nuclease.”

B- In panel C, in their direct testing of the mechanism that oxidation is important following DNA platination, the authors show that NAC co-treatment reduces gH2AX formation, but the key controls showing equivalent NMPNs uptake in the presence of NAC are missing, and the presence of equivalent amounts of the Pt-DNA adduct are also missing. This Pt-DNA missing data, which is shown in panel F suggests that the NAC treatment has major effects on DNA platination itself, not just oxidative cleavage, and makes it impossible to use the results to validate the direct importance of the ROS-dependent cleavage effects. All of the data needs to be quantified.

Response: *Thank you for your valuable suggestions. Per your suggestion, we studied the NMPNs uptake in the presence or absence of NAC. The result shows no obvious difference in the intracellular concentration of Pt ions after treatment with NMPNs or NMPNs+NAC, demonstrating that NAC exerts no effects on the cellular uptake of NMPNs. Additionally, we assessed the amount of Pt-DNA adducts in cisplatin-resistant Huh7 cells after treatments with NMPNs or NMPNs+NAC at different time points. As shown in Fig. 5b,c, there is no significant difference in the amount of Pt-DNA adducts between NMPNs-treated and NMPNs+NAC-treated cisplatin-resistant Huh7 cells at 6 h and 12 h, indicating that NAC shows no effects on the priming the formation of Pt-DNA adducts. This is because NAC shows no effect on the cellular uptake of NMPNs, which enables the comparable release of Pt ions. However, 18 h or 24 h later, we found the remarkable enhancement of Pt-DNA adducts in NMPNs-treated cisplatin-resistant Huh7 cells as compared to that of NMPNs+NAC-treated cisplatin-resistant Huh7 cells. It is understood that NMPNs can simultaneously release Pt ions to form Pt-DNA adducts and generate ROS to induce the cleavage of Pt-DNA adducts near the Pt-DNA binding sites, which can impair the DNA structure required for NER and thus inhibiting the NER-mediated DNA repairing. In contrast, the cotreatment with NAC can scavenge ROS generated by NMPNs, avoiding the oxidative cleavage of Pt-DNA adducts, which enables the NER-mediated DNA repairing. These results demonstrate that NMPNs can behave as a biomimetic nuclease to rupture the very structure of DNA required for NER by generating in situ ROS, which can induce DNA oxidative cleavage and facilitate the accumulation of Pt-DNA adducts.*

Our modification to the manuscript: *The results were added as Supplementary Fig. 22 and Fig. 5b,c in the revised supporting information. In addition, the following sentences and methods were added on pages 11-12 and 30 in the revised manuscript,*

respectively.

- Figure S22

Supplementary Figure 22. The concentrations of Pt ions in the cisplatin-resistant Huh7 cells after treatment with NMPNs or NMPNs+NAC. $n = 5$ independent experiments, data are presented as means \pm S.E.M. Source data are provided as a Source Data file.

• Figure 5

Fig. 5 NMPNs promote cisplatin-resistant tumour cell apoptosis by inducing Pt-DNA adducts without NER repairing. **a**, Immunofluorescence of the Pt-DNA adducts in cisplatin-resistant Huh7 cells after different treatments at pH 6.5. Scale bar: 40 μ m. **b**, Immunofluorescence of the Pt-DNA adducts in cisplatin-resistant Huh7 cells after treatments with NMPNs or NMPNs+NAC at different time points. Scale bar: 40 μ m. **c**, Quantitative analysis of Pt-DNA adducts in cisplatin-resistant Huh7 cells after treatments with NMPNs or NMPNs+NAC at different time points. **d**, Quantitative analysis of DNA damage of cisplatin-resistant Huh7 cells after treatment with cisplatin or NMPNs. Cisplatin: n = 46; NMPNs: n = 44. **e,f**, Flow cytometry analysis of cell apoptosis after different treatments at pH 6.5 (**e**) and corresponding quantitative results (**f**). **g**, The inhibition effect of cisplatin and NMPNs on cisplatin-resistant Huh7 cells growth at different incubation times. **h**, The cell viabilities of cisplatin-resistant Huh7 cells or siXPA-transfected cisplatin-resistant Huh7 cells after treatment with NMPNs, PNPs or cisplatin. All the data are presented as means \pm S.E.M., n = 3 independent

experiments. Statistical significance was analyzed by one-way ANOVA with multiple comparisons test. Source data are provided as a Source Data file.

- Page 11

“The cotreatment with ROS scavenger NAC shows no interference with the cellular uptake of NMPNs (Supplementary Fig. 22).”

- Page 11-12

“To fully understand the capability of NMPNs to facilitate the buildup of Pt-DNA adducts, we further investigated the Pt-DNA adducts formation and accumulation in cisplatin-resistant Huh7 cells after incubation with NMPNs in the presence or absence of NAC. After 6-hour or 12-hour incubation, we found that both NMPNs and NMPNs+NAC treatment can initiate the formation of Pt-DNA adducts by releasing Pt ions in the cisplatin-resistant Huh7 cells. However, after 18-hour or 24-hour incubation, only NMPNs can facilitate the accumulation of Pt-DNA adducts, demonstrating that NMPNs can not only release Pt ions to induce the formation of Pt-DNA adducts, but also effectively generate in situ ROS to destroy the DNA conformation required for NER²⁶, promoting the accumulation of cytotoxic Pt-DNA adducts. (Fig. 5b,c).”

- Page 30

Method

The concentration of NMPNs in cisplatin-resistant Huh7 cells in the presence or absence of NAC. Cisplatin-resistant Huh7 cells were treated with NMPNs (20 µg/mL) in the presence or absence of NAC for 24 h at pH 6.5, respectively. Then the cells were mixed with aqua regia and incubated at 60°C for 72 h. After dilution and filtration, the Pt concentrations were quantified by ICP-MS analysis.

C- What effect does NAC have on the PNP effects?

Response: *Thank you for your kind comment. Based on your suggestion, we studied the PNPs effect on the induction of DNA damage in cisplatin-resistant Huh7 cells in the presence or absence of NAC. The representative images of each group and corresponding quantitative analysis demonstrate that the cotreatment with NAC can compromise the effect of PNPs on the DNA damage induction in cisplatin-resistant Huh7 cells.*

Our modification to the manuscript: *The results were added as Supplementary Fig. 20 in the revised supporting information. In addition, the following sentences were added on pages 10 and 31-32 in the revised manuscript, respectively.*

- Figure S20

Supplementary Figure 20. a, Immunofluorescence images of the γ -H2AX in cisplatin-resistant Huh7 cells after treatments with PNPs with or without NAC at pH 6.5. Scale bar: 40 μ m. **b**, Quantification analysis of Pt-DNA adducts in cisplatin-resistant Huh7 cells after treatments with PNPs with or without NAC at pH 6.5. $n = 4$ independent experiments, data are presented as means \pm S.E.M. Source data are provided as a Source Data file.

- Page 10

“The cotreatment with NAC can compromise the effect of NMPNs and PNPs on the induction of oxidative cleavage of DNA (Fig. 4c and Supplementary Fig. 20).”

- Page 31-32

Method

Immunofluorescence detection of γ -H2AX. The cisplatin-resistant Huh7 cells were treated with cisplatin, NMPNs, NMPNs+NAC, PNPs or PNPs+NAC for 24 h at pH 6.5. Then, cells were fixed by 4% paraformaldehyde for 15 min at room temperature and permeabilized with 0.5% Triton X-100 for another 20 min, and then blocked with 5% bovine serum albumin in PBS for 1 h at room temperature. After which, the cells were incubated with primary antibodies at room temperature for 1 h: anti- γ -H2AX (dilution 1:100, ab81299) from Abcam (Cambridge, UK). Then, the cells were incubated with Alexa Fluor 488-conjugated goat anti rabbit IgG (H+L) from Boster Biological Technology Co., Ltd. for 1 h at room temperature. In addition, the DAPI solution was added for 15 min in the dark for nucleus staining. Finally, the cells were detected by using CLSM.

D- In panels D and E, the authors show primary data for XPA, which they do not quantify, and quantified data for XPF, which they do not show primary data for. Please

show IF images for all of the relevant XP proteins and quantify all of the data for both NMPNs and PNPs.

Response: *Thank you for your valuable comments. Per your suggestions, we have added the quantitative results for the IF images of XPA. In addition, the image of Western blot results and their corresponding quantitative analysis are also presented in Fig. 5h,i. The IF images and their corresponding quantitative analysis of XPF protein in the cisplatin-resistant Huh7 cells after treatment with NMPNs or NMPNs+NAC have also been provided. Moreover, we compared the effects of cisplatin, PNPs and NMPNs on the XPF recruitment into the nucleus of cisplatin-resistant Huh7 cells. The results showed that, among these different treatments, only NMPNs can significantly inhibit the recruitment of XPF into the nucleus and thus facilitating the accumulation of Pt-DNA adducts. These results demonstrate that NMPNs can induce concurrent DNA platination and oxidative cleavage, which lower the recruitment of XPA and XPF proteins into the nucleus of cisplatin-resistant Huh7 cells.*

Our modification to the manuscript: *The results were added as Fig. 4c-i and Supplementary Fig. 21 in the revised supporting information. In addition, the following sentences and method were added on pages 10-11 and 33 in the revised manuscript, respectively.*

• Figure 4

Fig. 4 NMPNs circumvent NER pathway by concurrent of DNA platination and oxidative cleavage. **a**, The level of Pt-DNA adducts in the nucleus of cisplatin-resistant Huh7 cells after treatment with NMPNs at pH 6.5. Scale bar: 40 μm. **b**, ROS levels in the nucleus of cisplatin-resistant Huh7 cells after treatment with PNPs or NMPNs at pH 6.5. Scale bar: 40 μm. **c**, Immunofluorescence of the γ-H2AX in cisplatin-resistant

Huh7 cells after different treatments at pH 6.5. Scale bar: 40 μm . **d**, Immunofluorescence of the XPA and Pt-DNA adducts in cisplatin-resistant Huh7 cells after different treatments at pH 6.5. The green fluorescence and the red fluorescence correspond to XPA and Pt-DNA adducts, respectively. Scale bar: 40 μm . **e**, Quantitative analysis of the colocalization between XPA and Pt-DNA adducts. **f**, Immunofluorescence of the XPF and Pt-DNA adducts in cisplatin-resistant Huh7 cells after different treatments at pH 6.5. The green fluorescence and the red fluorescence correspond to Pt-DNA adducts and XPF, respectively. Scale bar: 40 μm . **g**, Quantitative analysis of the colocalization between XPF and Pt-DNA adducts. **h**, Western blot analysis of XPF expression in the nucleus of cisplatin-resistant Huh7 cells after different treatments. **i**, Quantitative analysis of XPF expression in cisplatin-resistant Huh7 cells after different treatments at pH 6.5. **j**, The schematic illustration of the mechanism underlying NMPNs to induce DNA platination and oxidative cleavage to combat Pt resistance in tumour cells. In comparison to Pt compounds and PNPs that cannot effectively generate ROS in the nucleus, NMPNs can readily accumulate in the nucleus by acidity-induced exposure of TAT peptides, and induce in situ ROS generation to induce DNA oxidative cleavage, thus destroying the DNA conformation required for NER. It hampers the recruitment of XPA and XPF and thus inhibiting NER pathway. All the data are presented as means \pm S.E.M., n = 3 independent experiments. Statistical significance was analyzed by one-way ANOVA with multiple comparisons test. Source data are provided as a Source Data file.

- Figure S21

Supplementary Figure 21. a, Immunofluorescence images of the XPA and Pt-DNA adducts in cisplatin-resistant Huh7 cells after treatment with PNPs or NMPNs at pH 6.5 for 24 h. The green fluorescence and the red fluorescence correspond to Pt-DNA adducts and XPA, respectively. Scale bar: 40 μ m. **b** Quantification analysis of the colocalization between XPA and Pt-DNA adducts. **c**, Immunofluorescence images of the XPF and Pt-DNA adducts in cisplatin-resistant Huh7 cells after treatment with PNPs or NMPNs at pH 6.5 for 24 h. The green fluorescence and the red fluorescence correspond to Pt-DNA adducts and XPF, respectively. Scale bar: 40 μ m. **d** Quantification analysis of the colocalization between XPF and Pt-DNA adducts. Scale bar: 40 μ m. $n = 3$ independent experiments, data are presented as means \pm S.E.M. Source data are provided as a Source Data file.

- Page 10-11

“Intriguingly, as compared to PNPs, nearly no XPA colocalizes with Pt-DNA adducts after NMPNs treatment, indicating the recruitment of XPA is suppressed (Fig. 4d,e and Supplementary Fig. 21a,b). XPF, a pivotal NRE factor for making incisions near DNA lesion sites⁵², is also significantly downregulated after the treatment with NMPNs as

indicated by the decreased red fluorescence signals corresponding to XPF (Fig. 4f,g) and XPF protein expression level in the nucleus of cisplatin-resistant Huh7 cells (Fig. 4h,i). In comparison, PNPs and cisplatin show no impact on the NER process (Fig. 4h,i and Supplementary Fig. 21c,d).”

- Page 33

Method

Immunofluorescence detection of XPA or XPF and anti-cisplatin modified DNA.

The Huh7 cells were treated with PNPs, NMPNs or NMPNs plus NAC for 24 h at pH 6.5. The control group was set for no treatment. Then, cells were fixed by 4% paraformaldehyde for 15 min at room temperature and permeabilized with 0.5% Triton X-100 for another 20 min, and then blocked with 5% bovine serum albumin in PBS for 1 h at room temperature. After which, the cells were incubated with primary antibodies at room temperature for 1 h: anti-XPA (1:100, AF5336) from Beyotime Biotechnology (Shanghai, China) or anti-XPF (1:100, OM201253) from Omnimabs (California, USA) or anti-cisplatin modified DNA (dilution 1:100, GTX17412) from GeneTex (California, USA). Then, the cells were incubated with Alexa Fluor 488-conjugated rabbit anti-rat IgG (H+L) or Alexa Fluor 488-conjugated goat anti rabbit IgG (H+L) or Alexa Fluor 555-conjugated goat anti mouse IgG (H+L) from Boster Biological Technology Co., Ltd. for 1 h at room temperature. In addition, the DAPI solution was added for 15 min in the dark for nucleus staining. Finally, the cells were detected by using CLSM.

E- In panel I, how reproducible is the effect? How many times was the experiment done? Are the differences statistically significant?

Response: *Thank you for your valuable suggestion. The flow cytometry analysis of cell apoptosis was independently repeated three times, all showing that NMPNs can effectively induce the apoptosis of cisplatin-resistant Huh7 cells. In addition, we have added the quantitative results to clarify the statistical significance.*

Our modification to the manuscript: *The results were added as Fig. 5f in the revised manuscript. In addition, the following sentences were added on pages 12, 23, and 29 in the revised manuscript, respectively.*

• Figure 5f

Fig. 5 NMPNs promote cisplatin-resistant tumour cell apoptosis by inducing Pt-DNA adducts without NER repairing. **a**, Immunofluorescence of the Pt-DNA adducts in cisplatin-resistant Huh7 cells after different treatments at pH 6.5. Scale bar: 40 μ m. **b**, Immunofluorescence of the Pt-DNA adducts in cisplatin-resistant Huh7 cells after treatments with NMPNs or NMPNs+NAC at different time points. Scale bar: 40 μ m. **c**, Quantitative analysis of Pt-DNA adducts in cisplatin-resistant Huh7 cells after treatments with NMPNs or NMPNs+NAC at different time points. **d**, Quantitative analysis of DNA damage of cisplatin-resistant Huh7 cells after treatment with cisplatin or NMPNs. Cisplatin: n = 46; NMPNs: n = 44. **e,f**, Flow cytometry analysis of cell apoptosis after different treatments at pH 6.5 (**e**) and corresponding quantitative results (**f**). **g**, The inhibition effect of cisplatin and NMPNs on cisplatin-resistant Huh7 cells growth at different incubation times. **h**, The cell viabilities of cisplatin-resistant Huh7 cells or siXPA-transfected cisplatin-resistant Huh7 cells after treatment with NMPNs, PNPs or cisplatin. All the data are presented as means \pm S.E.M., n = 3 independent

experiments. Statistical significance was analyzed by one-way ANOVA with multiple comparisons test. Source data are provided as a Source Data file.

- Page 12

“Compared with cisplatin, NMPNs cause more severe DNA damage (Fig. 5d), effectively inducing apoptosis to inhibit the proliferation of cisplatin-resistant Huh7 cells (Fig. 5e-g)”

F- In panel J, how exactly is viability being measured? What is CCK-8 solution?

Response: *Thank you for your comments. We have clarified the CCK-8 measuring method of cell viability in the “Methods” section. The CCK-8 solution is a solution that contains WST-8, which is purchased from Shenzhen Sunview Technology Co., Ltd. In the presence of an electron-coupled reagent, WST-8 is reduced by some dehydrogenases within the mitochondria to produce the orange-yellow formazan.*

Our modification to the manuscript: *The method of the experiment to assess in vitro cytotoxicity of NMPNs was added on page 30-31 in the revised manuscript. In addition, the information of CCK-8 solution has been added to the Materials section on page 22 in the manuscript.*

- Page 22

Method

Materials. Platinum(II) acetylacetonate (97%), oleylamine (80-90%), lithium triethylborohydride, N-Hydroxysuccinimide sodium salt (NHS), N-(3-Dimethylaminopropyl)-N'-ethylcarbodiimide hydrochloride (EDC), cysteamine (CA), tetrahydrofuran (THF), N,N-dimethylformamide (DMF), 2-(N-Morpholino) ethanesulfonic acid (MES), triethylamine (TEA), tetramethylbenzidine (TMB), 5,5-dimethyl-1-pyrroline-N-oxide (DMPO) and acryloyl chloride (AC), Rhodamine B Isothiocyanate (RITC), chlorpromazine, amiloride and methyl- β -cyclodextrin (M β CD) were purchased from Aladdin. N-acetyl cysteine (NAC), Oleic acid (technical grade, 90%), and tetramethylbenzidine were purchased from Sigma-Aldrich. 5-tertbutoxycarbonyl-5methyl-1-pyrroline N-oxide (BMPO) was purchased from APEX BIO. Cisplatin injection was purchased from Haosen (Jiangsu) Pharmaceutical Company. mPEG_{5K}-OH (Mw = 5000), SH-PEG_{2K}-COOH (Mw = 2000), SH-PEG_{2K}-RITC (Mw = 2000) and RITC-PEG_{2K}-OH (Mw = 5000) were purchased from Punsore Biotechnology Company (China). TAT peptide (GRKKRRQRRR) was purchased from Chinese Peptide Company. Cell counting kit-8 was purchased from Shenzhen Sunview Technology Co., Ltd.

- Page 30-31

In vitro cytotoxicity. Cisplatin-resistant Huh7 cells were seeded into 96-well plates (1

$\times 10^4$ cells/well) for 24 h and treated with cisplatin, PNPs or NMPNs with a concentration of 20 $\mu\text{g/mL}$ for 6 h at pH 6.5. Then, the incubation medium containing cisplatin, PNPs or NMPNs were removed and replaced with fresh medium. After 6, 12 and 18 h, the cells were incubated with cell counting kit-8 (CCK-8) solution for each well and incubated for another 3 h, the absorbance of each well was measured at 450 nm by a microplate reader.

Cisplatin-resistant Huh7 cells were seeded into 96-well plates (1×10^4 cells/well) for 24 h and treated with NMPNs (20 $\mu\text{g/mL}$) in the presence or absence of NAC for 6 h at pH 6.5. Then, the cells were incubated with CCK-8 solution for each well and incubated for another 3 h, the absorbance of each well was measured at 450 nm by a microplate reader.

L02 cells were seeded into 96-well plates (1×10^4 cells/well) for 24 h and treated with NMPNs (20 $\mu\text{g/mL}$) for 6 h at pH 7.4. Then, the cells were incubated with CCK-8 solution for each well and incubated for another 3 h, the absorbance of each well was measured at 450 nm by a microplate reader.

5- This is a critical point - what direct proof do the authors have that lack of NER is the primary mechanism responsible for the enhanced cytotoxicity? To make this claim, the authors need to compare the effects of the NMPNs on isogenic cells that are NER-proficient versus NER-deficient, such as XP knock-outs, and show that the cytotoxicity, compared to PNPs is only enhanced in NER-proficient cells.

Response: *Thank you for your valuable comment. We carried out XPA siRNA transfection in cisplatin-resistant Huh7 cells by using two double-stranded siRNA sequences of XPA-Homo: (1) sense (5'-3')-GACCUGUUAUGGAAUUUGATT, anti-sense (5'-3')-UCAAAUCCAUACAGGUCTT; (2) sense (5'-3')-GGAGACGAUUGUUCAUCAATT, anti-sense (5'-3')-UUGAUGAACAAUCGUCUCCTT, resulting in the XPA knock-out (Supplementary Fig. 24). To investigate whether NER plays a primary role in the enhanced cytotoxicity of NMPNs against cisplatin-resistant Huh7 cells, we compared the inhibitory effect of NMPNs, PNPs, and cisplatin on the growth of cisplatin-resistant Huh7 cells. The results show that both PNPs and cisplatin exert enhanced cytotoxicity against NER-deficient Huh7 cells as compared to that of NER-proficient cisplatin-resistant Huh7 cells, indicating that NER deficiency can sensitize cisplatin-resistant Huh7 cells to PNPs and cisplatin. However, there is no obvious difference in the cytotoxicities of NMPNs against NER-proficient and NER-deficient Huh7 cells. In NER-deficient Huh7 cells, the XPA expression level is too low to be recruited to the Pt-DNA adducts for repairing <Ref. Nat. Commun., 2020, 11, 4124>. Similarly, in NER-proficient Huh7 cells that expresses high level of XPA, NMPNs can inhibit the recruitment of XPA to the Pt-DNA adducts by destroying the NER-required DNA bending structure. These*

results demonstrate that the enhanced cytotoxicity of NMPNs to cisplatin-resistant Huh7 cells is highly dependent on their capacity to impair NER pathway by inducing oxidative cleavage of Pt-DNA adducts.

Our modification to the manuscript: The results were added as Fig. 5h in the revised supporting information. In addition, the following sentences and methods were added on pages 12 and 34-35 in the revised manuscript, respectively.

• Figure 5

Fig. 5 NMPNs promote cisplatin-resistant tumour cell apoptosis by inducing Pt-DNA adducts without NER repairing. **a**, Immunofluorescence of the Pt-DNA adducts in cisplatin-resistant Huh7 cells after different treatments at pH 6.5. Scale bar: 40 μ m. **b**, Immunofluorescence of the Pt-DNA adducts in cisplatin-resistant Huh7 cells after treatments with NMPNs or NMPNs+NAC at different time points. Scale bar: 40 μ m. **c**, Quantitative analysis of Pt-DNA adducts in cisplatin-resistant Huh7 cells after treatments with NMPNs or NMPNs+NAC at different time points. **d**, Quantitative analysis of DNA damage of cisplatin-resistant Huh7 cells after treatment with cisplatin

or NMPNs. Cisplatin: n = 46; NMPNs: n = 44. **e,f**, Flow cytometry analysis of cell apoptosis after different treatments at pH 6.5 (**e**) and corresponding quantitative results (**f**). **g**, The inhibition effect of cisplatin and NMPNs on cisplatin-resistant Huh7 cells growth at different incubation times. **h**, The cell viabilities of cisplatin-resistant Huh7 cells or siXPA-transfected cisplatin-resistant Huh7 cells after treatment with NMPNs, PNPs or cisplatin. All the data are presented as means \pm S.E.M., n = 3 independent experiments. Statistical significance was analyzed by one-way ANOVA with multiple comparisons test. Source data are provided as a Source Data file.

- Figure S24

Supplementary Figure 24. Western blot analysis of XPA expression in cisplatin-resistant Huh7 cells, siNC-transfected cisplatin-resistant Huh7 cells and siXPA-transfected cisplatin-resistant Huh7 cells, respectively. Source data are provided as a Source Data file.

- Page 12

“To further investigate the role of NER in NMPNs-mediated therapeutic effect on cisplatin-resistant Huh7 cells, we checked the cytotoxicity of NMPNs, PNPs, or cisplatin to cisplatin-resistant Huh 7 cells (which are proficient in NER) and siXPA-transfected cisplatin-resistant Huh 7 cells (which are deficient in NER) (Supplementary Fig. 24). The results show that both PNPs and cisplatin exert enhanced cytotoxicity against NER-deficient Huh7 cells as compared to that of NER-proficient cisplatin-resistant Huh7 cells, indicating that NER deficiency can sensitize cisplatin-resistant Huh7 cells to PNPs and cisplatin. However, there is no obvious difference between the cytotoxicity of NMPNs against NER-proficient and NER-deficient Huh7 cells. In NER-deficient Huh7 cells, the XPA expression level is too low to be recruited to the Pt-DNA adducts for repairing⁵⁴. While in NER-proficient Huh7 cells that express a high level of XPA, NMPNs can inhibit the recruitment of XPA to the Pt-DNA adducts by destroying the NER-required DNA bending structure (Fig. 5h).”

Reference:

54. Kong, Y.W. et al. Enhancing chemotherapy response through augmented synthetic lethality by co-targeting nucleotide excision repair and cell-cycle checkpoints. *Nat.*

- Page 34-35

Method

XPA knockdown by small interfering RNA (siRNA). To silence the gene expression of XPA, XPA-specific siRNA (siXPA) and control siRNA (siNC) were obtained from Genepharma (Shanghai, China). Cisplatin-resistant Huh7 cells were seeded into 6-well plates (2×10^5 cells per well) overnight and then transfected with siRNA (siXPA or siNC) mixed with Lipofectamine 2000 transfection reagent, according to the recommended protocols by the manufacturer. The medium (Opti-MEM) was replaced at 6 h post-transfection. After 24 h, the cells were treated with cisplatin, PNPs, and NMPNs. SiRNA sequences of XPA-Homo: (1) sense (5'-3')-GACCUGUUAUGGAAUUUGATT, anti-sense (5'-3')-UCAAAUUCCAUAACAGGUCTT; (2) sense (5'-3')-GGAGACGAUUGUUCAUCAATT, anti-sense (5'-3')-UUGAUGAACAAUCGUCUCCTT.

6- Figure 5:

A- Images in Panel e are so dim that they are impossible to see, so the authors claims cannot be verified.

Response: *Thank you for your comment and we're sorry to make you confused. We have provided representative images with high quality of each group in the revised manuscript.*

Our modification to the manuscript: *The TUNEL (green fluorescence) and DAPI (blue fluorescence) staining images with high quality were added as Fig. 5e (Now is Fig. 6e) in the revised manuscript.*

• Figure 6

Fig. 6 NMPNs suppress tumour in vivo and improve overall therapeutic outcome.

a, Schematic illustration of the establishment of orthotopic liver tumour mice model and treatment process. **b**, Representative bioluminescence (BLI) images of each group after different treatments. **c**, Quantitative BLI signals of tumours after different treatments. $n = 5$ biologically independent animals. **d**, Representative images of liver collected from each treatment group on day 31. Black arrows indicate the tumour tissues. **e**, Representative TUNEL (green fluorescence) and DAPI (blue fluorescence) staining images of tumour tissues. Scale bar: 200 μ m. **f,g**, Western blot analysis and quantitative analysis of γ -H2AX, XPA and XPF in the nucleus of tumours cells after different treatments. $n = 3$ independent experiments. **h**, The body weight changes of mice with different treatments. $n = 5$ biologically independent animals. **i**, The survival curves of orthotopic liver tumour mice after different treatments. $n = 9$ biologically independent animals. **j**, The schematic illustration of NMPNs to induce Pt-DNA adduct formation and oxidative cleavage for inhibiting the recruitment of XPA and XPF, which

nullifies NER pathway and thus suppresses the tumour growth in vivo. All the data are presented as means \pm S.E.M. Statistical significance was analyzed by one-way ANOVA with multiple comparisons test (g,h) or two-tailed multiple *t*-tests with Bonferroni-Dunn correction (c). Source data are provided as a Source Data file.

B- In panel f, please show γ H2AX immunofluorescence, not just a Western blot

Response: Thank you for your kind comment. Based on your suggestion, γ -H2AX immunofluorescence is performed and representative images are shown in Supplementary Fig. 27, whose results are consistent with Western blot analysis.

Our modification to the manuscript: The results were added as Supplementary Fig. 27 in the revised supporting information. In addition, the following sentences and methods were added on pages 13 and 36 in the revised manuscript, respectively.

- Figure S27

Supplementary Figure 27. Immunofluorescence images of γ -H2AX in the tumour tissues after different treatments. n = 3 independent mice. Scale bar: 200 μ m.

- Page 13

“Western blot analysis of tumour tissue reveals that NMPNs can decrease the expression of XPA and XPF in the nucleus to inhibit the DNA repair mediated by NER, thus inducing more severe DNA damage (Fig. 6f,g). Consistently, the higher signal of green fluorescence corresponding to γ -H2AX is observed in the tumour tissues of mice treated with NMPNs as compared with that of mice treated with cisplatin or PNPs (Supplementary Fig. 27).”

Method

Immunofluorescence detection of γ -H2AX of tumour tissue. Mice were treated with saline, cisplatin, PNPs or NMPNs (2.5 mg/kg body weight) three times a week. After 14 days, tumour tissues were collected for further analysis of γ -H2AX expression by immunohistochemical staining.

7- The models in Figure 4K and 5J are not adequately supported by the data, as I mention in comment #5.

Response: *Thank you for your valuable comment. We carried out XPA siRNA transfection in cisplatin-resistant Huh7 cells by using two double-stranded siRNA sequences of XPA-Homo: (1) sense (5'-3')-GACCUGUUAUGGAAUUUGATT, anti-sense (5'-3')-UCAAAUCCAUACAGGUCTT; (2) sense (5'-3')-GGAGACGAUUGUUCAUCAATT, anti-sense (5'-3')-UUGAUGAACAAUCGUCUCCTT, resulting in the XPA knock-out (Supplementary Fig. 24). To investigate whether NER plays a primary role in the enhanced cytotoxicity of NMPNs against cisplatin-resistant Huh7 cells, we compared the inhibitory effect of NMPNs, PNPs, and cisplatin on the growth of cisplatin-resistant Huh7 cells. These results show that both PNPs and cisplatin exert the enhanced cytotoxicity against NER-deficient Huh7 cells as compared to that against NER-proficient cisplatin-resistant Huh7 cells, indicating that NER deficiency can sensitize cisplatin-resistant Huh7 cells to PNPs and cisplatin. However, there is no obvious difference in the cytotoxicity of NMPNs against NER-proficient and NER-deficient Huh7 cells. In NER-deficient Huh7 cells, the XPA expression level is too low to be recruited to the Pt-DNA adducts for repairing <Ref. Nat. Commun., 2020, 11, 4124>. Similarly, in NER-proficient Huh7 cells that expresses high level of XPA, NMPNs can inhibit the recruitment of XPA to the Pt-DNA adducts by destroying the NER-required DNA bending structure. Moreover, we extracted DNA fragmentations from the nucleus of cisplatin-resistant Huh7 cells after NMPNs treatment, and analyzed the Pt-DNA binding sites by using confocal laser scanning microscopy. The result shows that the green signals corresponding to the Pt-DNA binding mainly localize at the end of DNA fibers (Supplementary Fig. 19). These results demonstrate that the enhanced cytotoxicity of NMPNs to cisplatin-resistant Huh7 cells is highly dependent on their capacity to impair the NER pathway by inducing oxidative cleavage of Pt-DNA adducts.*

Our modification to the manuscript: *The results were added as Supplementary Fig. 19, 24 and Fig. 5h in the revised manuscript. In addition, the following sentences and methods were added on pages 10, 12, 28, and 34-35 in the revised manuscript, respectively.*

- Figure S24

Supplementary Figure 24. Western blot analysis of XPA expression in cisplatin-resistant Huh7 cells, siNC-transfected cisplatin-resistant Huh7 cells and siXPA-transfected cisplatin-resistant Huh7 cells, respectively. Source data are provided as a Source Data file.

- Figure S19

DAPI / Pt-DNA adducts

Supplementary Figure 19. The analysis of Pt-DNA binding sites in the DNA extracted from cisplatin-resistant Huh7 cells after treatment with NMPNs. The blue fluorescence and the green fluorescence correspond to DNA fibers and Pt-DNA binding sites, respectively. Arrows indicate the DNA fragmentations. Asterisks indicate the Pt-DNA binding sites at the end of DNA fragmentations. n = 3 independent experiments. Scale bar: 100 μm .

• Figure 5

Fig. 5 NMPNs promote cisplatin-resistant tumour cell apoptosis by inducing Pt-DNA adducts without NER repairing. **a**, Immunofluorescence of the Pt-DNA adducts in cisplatin-resistant Huh7 cells after different treatments at pH 6.5. Scale bar: 40 μ m. **b**, Immunofluorescence of the Pt-DNA adducts in cisplatin-resistant Huh7 cells after treatments with NMPNs or NMPNs+NAC at different time points. Scale bar: 40 μ m. **c**, Quantitative analysis of Pt-DNA adducts in cisplatin-resistant Huh7 cells after treatments with NMPNs or NMPNs+NAC at different time points. **d**, Quantitative analysis of DNA damage of cisplatin-resistant Huh7 cells after treatment with cisplatin or NMPNs. Cisplatin: n = 46; NMPNs: n = 44. **e,f**, Flow cytometry analysis of cell apoptosis after different treatments at pH 6.5 (**e**) and corresponding quantitative results (**f**). **g**, The inhibition effect of cisplatin and NMPNs on cisplatin-resistant Huh7 cells growth at different incubation times. **h**, The cell viabilities of cisplatin-resistant Huh7 cells or siXPA-transfected cisplatin-resistant Huh7 cells after treatment with NMPNs, PNPs or cisplatin. All the data are presented as means \pm S.E.M., n = 3 independent

experiments. Statistical significance was analyzed by one-way ANOVA with multiple comparisons test. Source data are provided as a Source Data file.

- Page 10

“In line with the DFT results, we found that NMPNs can induce the formation of Pt-DNA adducts and DNA breakage in the cisplatin-resistant Huh7 cells (Fig. 4a,c). Moreover, we extracted the DNA fragmentations from NMPNs-treated cisplatin-resistant Huh7 cells and analyzed the location of Pt-DNA binding in the DNA fragmentations. Intriguingly, the result shows that the Pt-DNA binding sites mainly localize at the end of DNA fragmentation (Supplementary Fig. 19).”

- Page 12

“To further investigate the role of NER in NMPNs-mediated therapeutic effect on cisplatin-resistant Huh7 cells, we checked the cytotoxicity of NMPNs, PNPs, or cisplatin to cisplatin-resistant Huh 7 cells (which are proficient in NER) and siXPA-transfected cisplatin-resistant Huh 7 cells (which are deficient in NER) (Supplementary Fig. 24). The results show that both PNPs and cisplatin exert enhanced cytotoxicity against NER-deficient Huh7 cells as compared to that of NER-proficient cisplatin-resistant Huh7 cells, indicating that NER deficiency can sensitize cisplatin-resistant Huh7 cells to PNPs and cisplatin. However, there is no obvious difference between the cytotoxicity of NMPNs against NER-proficient and NER-deficient Huh7 cells. In NER-deficient Huh7 cells, the XPA expression level is too low to be recruited to the Pt-DNA adducts for repairing⁵⁴. While in NER-proficient Huh7 cells that express a high level of XPA, NMPNs can inhibit the recruitment of XPA to the Pt-DNA adducts by destroying the NER-required DNA bending structure (Fig. 5h).”

Reference:

54. Kong, Y.W. et al. Enhancing chemotherapy response through augmented synthetic lethality by co-targeting nucleotide excision repair and cell-cycle checkpoints. *Nat. Commun.* **11**, 4124 (2020).

- Page 28

Method

The analysis of locations of Pt-DNA binding in the DNA. Cisplatin-resistant Huh7 cells were cultured in 6-well plates with 2 mL culture medium. NMPNs (20 µg/mL, pH 6.5) were added and incubated for 24 h, cells were subsequently collected and quickly resuspended in 500-600 µL of ice-cold PBS. Then, 2.5 µL of cell resuspension was spotted onto a pre-cleaned glass slide and mixed with 7.5 µL of spreading buffer (0.5% SDS in 200 mM Tris-HCl (pH 7.4), 50 mM EDTA). After 10 min, the slides were tilted to 15° to spread DNA fibers along the length. Then, air-dry the DNA spreads and fixed in 3:1 methanol/acetic acid for 20 min at -20°C. After washing with PBS three times,

slides were blocked with 1% BSA in PBS for 30 min at room temperature and incubated with anti-cisplatin modified DNA antibodies (GeneTex, GTX17412, dilution 1:100) to detect Pt-DNA adduct. After 1 h of incubation, slides were washed with PBS three times and stained with Alexan Fluor 488-conjugated AffiniPure Rabbit anti-Rat IgG (H+L) (Boster, BA1129, dilution 1:200) for 2 h at room temperature in the dark. After washing with PBS three times, slides were incubated with DAPI solution for 15 min in the dark. Finally, the cells were detected by using CLSM.

- Page 34-35

XPA knockdown by small interfering RNA (siRNA). To silence the gene expression of *XPA*, *XPA*-specific siRNA (si*XPA*) and control siRNA (siNC) were obtained from Genepharma (Shanghai, China). Cisplatin-resistant Huh7 cells were seeded into 6-well plates (2×10^5 cells per well) overnight and then transfected with siRNA (si*XPA* or siNC) mixed with Lipofectamine 2000 transfection reagent, according to the recommended protocols by the manufacturer. The medium (Opti-MEM) was replaced at 6 h post-transfection. After 24 h, the cells were treated with cisplatin, PNPs, and NMPNs. SiRNA sequences of *XPA*-Homo: (1) sense (5'-3')-GACCUGUUAUGGAAUUUGATT, anti-sense (5'-3')-UCAAAUUGCAUAACAGGUUCT; (2) sense (5'-3')-GGAGACGAUUGUUCAUCAATT, anti-sense (5'-3')-UUGAUGAACAAUCGUCUCCT.

In summary, the authors clearly appear to have created a novel nanoparticle with potential utility, but the mechanism that underlies the cellular effects is not adequately validated. I think the results are important even if the NER-focused mechanism is wrong, but additional experimental data is required.

Response: *Thank you very much for your encouraging comments. Based on your suggestions, we have provided several lines of evidence showing NMPNs induce concurrent DNA platination and oxidative cleavage to inhibit NER-mediated DNA repairing for combating Pt-based drug resistance in cancer treatment. (1) Compared to cisplatin and PNPs, NMPNs significantly facilitate the formation of Pt-DNA adducts and induce the DNA cleavage near the Pt-DNA binding sites, hindering the recruitment of XPA and XPF to the damaged DNA for NER-mediated DNA repairing. (2) In contrast to cisplatin and PNPs, NMPNs show no enhanced cytotoxicity to NER deficient cancer cells as compared to that of NER sufficient cancer cells, suggesting that the therapeutic effects of NMPNs is mainly dependent on the inhibition of NER pathway. We are sure that these revisions have made a significant improvement in the quality of the manuscript. Thank you very much again for your diligent review of our paper.*

Reviewers' Comments:

Reviewer #1:

Remarks to the Author:

The authors have adequately addressed the comments raised by this reviewer.

Reviewer #2:

Remarks to the Author:

The authors have addressed most, if not all, issues raised in the previous round. I have no other comments. Congratulations for the authors to generate a great piece of work.

Reviewer #3:

Remarks to the Author:

The authors have adequately addressed my comments.

Dear Joanne,

We are delighted to receive your email and have addressed the editorial requests as quickly as possible to publish our manuscript (NCOMMS-22-09377B) in Nature Communications before the end of 2022. We deeply appreciate your detailed and constructive comments to comply with the policies and formatting requirements. In the “Editorial Quests” file, we have carefully provided point-by-point responses to all the guidance and suggestions. Thank you very much for your kind help, we are looking forward to hearing from you soon.

REVIEWERS’ COMMENTS

Reviewer #1 (Remarks to the Author):

The authors have adequately addressed the comments raised by this reviewer.

Response: Thank you very much for your kind supports.

Reviewer #2 (Remarks to the Author):

The authors have addressed most, if not all, issues raised in the previous round. I have no other comments. Congratulations for the authors to generate a great piece of work.

Response: Thank you very much for your kind supports.

Reviewer #3 (Remarks to the Author):

The authors have adequately addressed my comments.

Response: Thank you very much for your kind supports.